# Sampling with Riemannian Hamiltonian Monte Carlo in a Constrained Space

**Yunbum Kook**
Georgia Tech
yb.kook@gatech.edu

**Yin Tat Lee**
Microsoft Research,
University of Washington
yintat@uw.edu

**Ruoqi Shen**
University of Washington
shenr3@cs.washington.edu

**Santosh S. Vempala**
Georgia Tech
vempala@gatech.edu

## Abstract

We demonstrate for the first time that ill-conditioned, non-smooth, constrained distributions in very high dimension, upwards of 100,000, can be sampled efficiently *in practice*. Our algorithm incorporates constraints into the Riemannian version of Hamiltonian Monte Carlo and maintains sparsity. This allows us to achieve a mixing rate independent of condition numbers.

On benchmark data sets from systems biology and linear programming, our algorithm outperforms existing packages by orders of magnitude. In particular, we achieve a 1,000-fold speed-up for sampling from the largest published human metabolic network (RECON3D). Our package has been incorporated into the COBRA toolbox.

## 1 Introduction

**Sampling is Fundamental.** Sampling algorithms arise naturally in models of statistical physics, e.g., Ising, Potts models for magnetism, Gibbs model for gases, etc. These models directly suggest Markov chain algorithms for sampling the corresponding configurations. In the Ising model where the vertices of a graph are assigned a spin, i.e., $\pm 1$, in each step, we pick a vertex at random and flip its spin with some probability. The probability is chosen so that the distribution of the vector of all spins approaches a target distribution where the probability exponentially decays with the number of agreements in spin for pairs corresponding to edges of the graph. In the Gibbs model, particles move randomly with collisions and their motion is often modeled as reflecting Brownian motion. Sampling with Markov chains is today the primary algorithmic approach for high-dimensional sampling. For some fundamental problems, sampling with Markov chains is the only known efficient approach or the only approach to have guarantees of efficiency. Two notable examples are sampling perfect matchings of a bipartite graph and sampling points from a convex body. These are the core subroutines for estimating the permanent of a nonnegative matrix and estimating the volume of a convex body, respectively. The solution space for these problems scales exponentially with the dimension. In spite of this, polynomial-time algorithms have been discovered for both problems. The current best permanent algorithm scales as $n^7$ (time) [2, 23], while the current best volume algorithm scales as $n^3$ (number of membership tests) [24]. For the latter, the first polynomial-time algorithm had a complexity of $n^{27}$ [16], and the current best complexity is the result of many breakthrough discoveries, including general-purpose algorithms and analysis tools.

36th Conference on Neural Information Processing Systems (NeurIPS 2022).

**Sampling is Ubiquitous.** The need for efficient high-dimensional sampling arises in many fields. A notable setting is *metabolic networks* in systems biology. A constraint-based model of a metabolic network consists of $m$ metabolites and $n$ reactions, and a set of equalities and inequalities that define a set of feasible steady state reaction rates (fluxes):

$$\Omega = \left\{ v \in \mathbb{R}^n \,|\, Sv = 0,\, l \le v \le u,\, c^T v = \alpha \right\},$$

where $S$ is a stoichiometric matrix with coefficients for each metabolite and reaction. The linear equalities ensure that the fluxes into and out of every node are balanced. The inequalities arise from thermodynamical and environmental constraints. Sampling constraint-based models is a powerful tool for evaluating the metabolic capabilities of biochemical networks [33, 46]. While the most common distribution used is uniform over the feasible region, researchers have also argued for sampling from the Gaussian density restricted to the feasible region; the latter has the advantage that the feasible set does not have to be bounded. A previous approach to sampling, using hit-and-run with rounding [20], has been incorporated into the COBRA package [21] for metabolic systems analysis (Bioinformatics).

A second example of mathematical interest is the problem of computing the volume of the Birkhoff polytope. For a given dimension $n$, the Birkhoff polytope is the set of all doubly stochastic $n \times n$ matrices (or the convex hull of all permutation matrices). This object plays a prominent role in algebraic geometry, probability, and other fields. Computing its volume has been pursued using algebraic representations; however exact computations become intractable even for $n = 11$, requiring years of computation time. Hit-and-run has been used to show that sampling-based volume computation can go to higher dimension [11], with small error of estimation. However, with existing sampling implementations, going beyond $n = 20$ seems prohibitively expensive.

A third example is from machine learning, a field that is increasingly turning to *sampling* models of data according to their performance in some objective. One such commonly used criterion is the logistic regression function. The popularity of logistic regression has led to sampling being incorporated into widely used packages such as STAN [44], PyMC3 [41], and Pyro [3]. However, those packages in general do not run on the constraint-based models we are interested in.

**Problem Description.** In this paper, we consider the problem of sampling from distributions whose densities are of the form

$$e^{-f(x)} \text{ subject to } Ax = b, x \in K \tag{1.1}$$

where $f$ is a convex function and $K$ is a convex body. We assume that a self-concordant barrier $\phi$ for $K$ is given. Note that any convex body has a self-concordant barrier [32] and there are explicit barriers for convex bodies that come up in practical applications [37], so this is a mild assumption. We introduce an efficient algorithm for the problem when $K$ is a product of convex bodies $K_i$, each with small dimension. Many practical instances can be written in this form. As a special case, the algorithm can handle $K$ in the form of $\{x \in \mathbb{R}^n : l_i \le x_i \le u_i \text{ for all } i \in [n]\}$ with $l_i \in \mathbb{R} \cup \{-\infty\}$ and $u_i \in \mathbb{R} \cup \{+\infty\}$, which is the common model structure in systems biology. Moreover, any generalized linear model $\exp(-\sum f_i(a_i^\top x - b_i))$, e.g., the logistic model, can be rewritten in the form

$$\exp(-\sum t_i) \text{ subject to } Ax = b + s, (s, t) \in K \tag{1.2}$$

where $K = \Pi K_i$ and each $K_i = \{(s_i, t_i) : f_i(s_i) \le t_i\}$ is a two-dimensional convex body.

**The Challenges of Practical Sampling.** High dimensional sampling has been widely studied in both the theoretical computer science and the statistics communities. Many popular samplers are first-order methods, such as MALA [40], basic HMC [36, 14] and NUTS [22], which update the Markov chain based on the gradient information of $f$. The runtime of such methods can depend on the condition number of the function $f$ [15, 30, 7, 8, 42]. However, the condition number of real-world applications can be very large. For example, RECON1 [27], a reconstruction of the human metabolic network, can have condition number as large as $10^6$ due to the dramatically different orders of different chemicals' concentrations. Motivated by sampling from ill-conditioned distributions, another class of samplers use higher-order information such as Hessian of $f$ to take into account the local structure of the problems [43, 9]. However, such samplers cannot handle non-smooth distributions, such as hinge-loss, lasso, or uniform densities over polytopes.

For non-smooth distributions, the best polytime methods are based on discretizations of Brownian motion, e.g., the Ball walk [25] (and its affine-invariant cousin, the Dikin walk [26]), which takes a

random step in a ball of a fixed size around the current point. Hit-and-Run [34] builds on these by avoiding an explicit step size and going to a random point along a random line through the current point. Both approaches hit the same bottleneck — in a polytope that contains a unit ball, the step size should be $O(1/\sqrt{n})$ to avoid stepping out of the body with large probability. This leads to quadratic bounds (in dimension) on the number of steps to "mix".

Due to the reduction mentioned in (1.2), non-smooth distributions can be translated to the form in (1.1) with constraint $K$. Both the first and higher-order sampler and the polytime non-smooth samplers have their limitations in handling distributions with non-smooth objective function or constraint $K$. Given the limitations of all previous samplers, a natural question we want to ask is the following.

**Question.** *Can we develop a practically efficient sampler that can handle the constrained problem in* (1.1) *and preserve sparsity[1] with mixing time independent of the condition number?*

In some applications, smoothness and condition number can be controlled with tailor-made models. Our goal here is to propose a general solver that can sample from any non-smooth distributions as given. For traditional samplers such as the Ball walk and Hit-and-Run, as mentioned earlier, the step size needs to be small so that the process does not step out. An approach that gets around this bottleneck is Hamiltonian Monte Carlo (HMC), where the next step is given by a point along a Hamiltonian-preserving curve according to a suitably chosen Hamiltonian. It has two advantages. First, the steps are no longer straight lines in Euclidean space, and we no longer have the concern of "stepping out". Second, the process is *symplectic* (so measure-preserving), and hence the filtering step is easy to compute. It was shown in [31] that significantly longer steps can be taken and the process with a convergence analysis in the setting of Hessian manifolds, leading to subquadratic convergence for uniformly sampling polytopes.

To make this practical, however, is a formidable challenge. There are two high-level difficulties. One is that many real-world instances are *highly skewed* (far from isotropic) and hence it is important to use the local geometry of the density function. This means efficiently computing or maintaining second-order information such as a Hessian of the logarithm of the density. This can be done in the Riemannian HMC (RHMC) framework [17, 31], but the computation of the next step requires solving the Hamiltonian ODE to high accuracy, which in turn needs the computation of leverage scores, a procedure that takes at least matrix-multiplication time in the worst case. Another important difficulty is maintaining hard linear constraints. Existing high-dimensional packages do not allow for constraints (they must be somehow incorporated into the target density), and RHMC is usually considered with a full-dimensional feasible region such as a full-dimensional polytope. This can also be done in the presence of linear equalities by working in the affine subspace defined by the equalities, but this has the effect of *losing any sparsity* inherent in the problem and turning all coefficient matrices and objective coefficients into dense objects, thereby potentially incurring a quadratic blow-up.

**Our Solution: Constrained Riemannian Hamiltonian Monte Carlo (CRHMC).** We develop a constrained version of RHMC, maintaining both *sparsity* and *constraints*. Our refinement of RHMC ensures that the process satisfies the given constraints throughout, without incurring a significant overhead in time or sparsity. It works even if the resulting feasible region is poorly conditioned. Since many instances in practice are ill-conditioned and have degeneracies, we believe this is a crucial aspect. Our algorithm outperforms existing packages by orders of magnitude.

In Section 2, we give the main ingredients of the algorithm and discuss how we overcome the challenges that prevent us from sampling efficiently in practice. Following that, in Section 3, we present empirical results on several benchmark datasets, showing that CRHMC successfully samples much larger models than previously known to be possible, and is significantly faster in terms of rate of convergence ("number of steps") and total sampling time. Our complete package is available on GitHub. We refer the reader to Appendix for theory, notations, and definitions.

---

[1]When $A$ is sparse, preserving the sparsity of $A$ can greatly enhance both the runtime and the space efficiency.

# 2 Algorithm: Constrained RHMC

In this section, we propose a constrained Riemannian Hamiltonian Monte Carlo (CRHMC[2]) algorithm to sample from a distributions of the form

$$e^{-f(x)} \text{ subject to } c(x) = 0 \text{ and } x \in K \text{ for some convex body } K,$$

where the constraint function $c : \mathbb{R}^n \to \mathbb{R}^m$ satisfies the property that the Jacobian $Dc(x)$ has full rank for all $x$ such that $c(x) = 0$. It is useful to keep in mind the case when $c(x) = 0$ is an affine subspace $Ax = b$, in which case $Dc(x) = A$, and the full-rank condition simply says that the rows of $A$ are independent.

We refer readers to [1, 4, 39] for preliminary versions of CRHMC called the constrained Hamiltonian Monte Carlo (CHMC). In particular, a framework in [4] can be extended to CRHMC when $K = \mathbb{R}^n$, and in fact they mention CRHMC as a possible variant. However, their algorithm for CRHMC requires eigenvalue decomposition and is not efficient for large problems, which takes $n^3$ time and $n^2$ space per MCMC step in practice. In this section, we propose an algorithm that overcomes those limitations and satisfies the additional constraint $K$ by using a local metric induced by the Hessian of self-concordant barriers, leading to $n^{1.5}$ time and $n$ space in practice.

## 2.1 Basics of CRHMC

To introduce our algorithm, we first recall the RHMC algorithm (Algorithm 1). In RHMC, we extend the space $x$ to the pair $(x, v)$, where $v$ denotes the *velocity*. Instead of sampling from $e^{-f(x)}$, RHMC samples from the distribution $e^{-H(x,v)}$, where $H(x, v)$ is the Hamiltonian, and then outputs $x$. To make sure the distribution is correct, we choose the Hamiltonian such that the marginal of $e^{-H(x,v)}$ along $v$ is proportional to $e^{-f(x)}$. One common choice of $H(x, v)$ is

$$H(x, v) = f(x) + \frac{1}{2}v^\top M(x)^{-1}v + \frac{1}{2}\log \det M(x), \tag{2.1}$$

where $M(x)$ is a position-dependent positive definite matrix defined on $\mathbb{R}^n$.

---

**Algorithm 1:** `Riemannian Hamiltonian Monte Carlo` (RHMC)

---

**Input:** Initial point $x^{(0)}$, step size $h$

**for** $k = 1, 2, \cdots$ **do**

    `// Step 1: resample v`

    Sample $v^{(k-\frac{1}{2})} \sim \mathcal{N}(0, M(x^{(k-1)}))$ and set $x^{(k-\frac{1}{2})} \leftarrow x^{(k-1)}$.

    `// Step 2: Hamiltonian dynamics`

    Solve the ODE

$$\frac{dx}{dt} = \frac{\partial H(x, v)}{\partial v}, \ \frac{dv}{dt} = -\frac{\partial H(x, v)}{\partial x} \tag{2.2}$$

    with $H$ defined in (2.1) and the initial point given by $(x^{(k-\frac{1}{2})}, v^{(k-\frac{1}{2})})$.

    Set $x^{(k)} \leftarrow x(h)$ and $v^{(k)} \leftarrow v(h)$.

**end**

**Output:** $x^{(k)}$

---

To extend RHMC to the constrained case, we need to make sure both Step 1 and Step 2 satisfy the constraints, so the Hamiltonian dynamics has to maintain $c(x) = 0$ throughout Step 2. Note that

$$\frac{d}{dt}c(x_t) = Dc(x_t) \cdot \frac{dx_t}{dt} = Dc(x_t) \cdot \frac{\partial H(x_t, v_t)}{\partial v_t}, \tag{2.3}$$

where $Dc(x)$ is the Jacobian of $c$ at $x$. With $H$ defined in (2.1), Condition (2.3) becomes $Dc(x)M(x)^{-1}v = 0$. However, for full rank $Dc(x)$, if $M(x)$ is invertible, then $\text{Range}(v) = \text{Range}(\mathcal{N}(0, M(x))) = \mathbb{R}^n$ immediately violates this condition due to

---

[2]pronounced "crumch".

$\dim(\mathrm{Null}(Dc(x)M^{-1}(x))) = n - m$. To get around this issue, we use a non-invertible matrix $M(x)$ with its pseudo-inverse $M(x)^\dagger$ to satisfy $Dc(x)M(x)^\dagger v = 0$ for any $v \in \mathrm{Range}(M(x))$. Since we want the step to be able to move in all directions satisfying $c(x) = 0$, we impose the following condition with $\mathrm{Range}(M(x)) = \mathrm{Range}(M(x)^\dagger)$ in mind:

$$\mathrm{Range}(M(x)) = \mathrm{Null}(Dc(x)) \text{ for all } x \in \mathbb{R}^n, \qquad (2.4)$$

which can be achieved by $M(x)$ proposed soon.

Under the condition (2.4), we sample $v$ from $\mathcal{N}(0, M(x))$ in Step 1, which is equivalent to sampling from $e^{-H(x,v)}$ subject to $v \in \mathrm{Range}(M(x)) = \mathrm{Null}(Dc(x))$. Also, the stationary distribution of CRHMC should be proportional to

$$e^{-H(x,v)} \text{ subject to } c(x) = 0 \text{ and } v \in \mathrm{Null}(Dc(x)).$$

Here, to maintain $v \in \mathrm{Null}(Dc(x))$ during Step 2 we add a Lagrangian term to $H$. Without the Lagrangian term, $v_t$ would escape from $\mathrm{Null}(Dc(x_t)) = \mathrm{Range}(M(x_t))$ in Step 2 as seen in the proof of Lemma 1, which contradicts $\mathrm{Range}(v_t) = \mathrm{Range}(\mathcal{N}(0, M(x_t))) = \mathrm{Range}(M(x_t))$. The constrained Hamiltonian we propose is (See its rigorous derivation in Lemma 1)

$$H(x,v) = \overline{H}(x,v) + \lambda(x,v)^\top c(x) \quad \text{with} \quad \overline{H}(x,v) = f(x) + \frac{1}{2}v^\top M(x)^\dagger v + \log \mathrm{pdet}(M(x)) \tag{2.5}$$

where $\lambda(x,v) = (Dc(x)Dc(x)^\top)^{-1}\left(D^2 c(x)[v, \frac{dx}{dt}] - Dc(x)\frac{\partial \overline{H}(x,v)}{\partial x}\right)$. Here, $\mathrm{pdet}$ denotes pseudo-determinant and $\lambda(x,v)$ is picked so that $v \in \mathrm{Null}(Dc(x))$. An algorithmic description of CRHMC is the same as Algorithm 1 with the constrained $H$ in place of the unconstrained $\overline{H}$. We show the convergence of CRHMC to the correct distribution $\exp(-f(x))$ in Appendix B.3.

**Choice of $M$ via Self-concordant Barriers.** The construction of the Hamiltonian (2.5) relies on having a family of positive semi-definite matrix $M(x)$ satisfying the condition (2.4) (i.e., $\mathrm{Range}(M(x)) = \mathrm{Null}(Dc(x))$). One natural choice is the orthogonal projection to $\mathrm{Null}(Dc(x))$:

$$Q(x) = I - Dc(x)^\top (Dc(x)Dc(x)^\top)^{-1} Dc(x), \qquad (2.6)$$

which is similar to the choice in [4].

For the problem we care about, there are additional constraints on $x$ other than $\{c(x) = 0\}$. In the standard HMC algorithm, we have $\frac{dx}{dt} \sim \mathcal{N}(0, M(x)^{-1})$. For example, for a simple constraint $K = [0, 1]$, to ensure every direction is moving towards/away from $x = 0$ multiplicatively, a natural choice of $M$ is $M(x) = \mathrm{diag}(x^{-2})$. For general convex body $K$, we can use a *self-concordant barrier*, a function defined on $K$ such that $\phi(x)$ is self-concordant and $\phi(x) \to +\infty$ as $x \to \partial K$. Using the barrier $\phi$, we can define the local metric based on $g(x) = \nabla^2 \phi(x)$. Intuitively, as the sampler approaches $\partial K$, the local metric stretches accordingly so that the Hamiltonian dynamics never passes the barrier, respecting $x \in K$ throughout.

In summary, we need $M(x)$ to have its range match the null space of $Dc(x)$ and agree with $g(x)$ in its range. We can verify that $M(x) = Q(x)^\top g(x)Q(x)$, where $Q(x)$ is the symmetric matrix defined in (2.6), satisfies these two constraints.

## 2.2 Efficient Computation of $\partial H/\partial x$ and $\partial H/\partial v$

With $M(x) = Q(x)^\top g(x)Q(x)$, we have all the pieces of the algorithm. However, using this naive algorithm to compute $\partial H/\partial x$ and $\partial H/\partial v$, we face several challenges.

1. The algorithm involves computing the pseudo-inverse and its derivatives, which takes $O(n^3)$ except for very special matrices.

2. The Lagrangian term in the constrained Hamiltonian dynamics requires additional computation such as the second-order derivative of $c(x)$.

3. A naive approach to computing leverage scores in $\partial H/\partial x$ results in a very dense matrix.

Those challenges make the algorithm hard to implement and inefficient, especially when the dimension is high. In the following paragraphs, we give an overview of how we overcome each of the challenges above. We defer a more detailed discussion of our approaches and the proofs to Appendix B.2.

**Avoiding Pseudo-inverse and Pseudo-determinant.** We are able to show equivalent formulas for $M(x)^\dagger$ and $\log \operatorname{pdet} M(x)$ that can take advantage of sparse linear system solvers. In particular, we show that $M(x)^\dagger = g(x)^{-\frac{1}{2}} \cdot (I - P(x)) \cdot g(x)^{-\frac{1}{2}}$, where

$$P(x) = g(x)^{-\frac{1}{2}} \cdot Dc(x)^\top (Dc(x) \cdot g(x)^{-1} \cdot Dc(x)^\top)^{-1} Dc(x) \cdot g(x)^{-\frac{1}{2}}. \tag{2.7}$$

As mentioned earlier, a majority of convex bodies appearing in practice are of the form $K = \prod_i K_i$, where $K_i$ are constant dimensional convex bodies. In this case, we will choose $g(x)$ to be a block diagonal matrix with each block of size $O(1)$. Hence, the bottleneck of applying $P(x)$ to a vector is simply solving a linear system of the form $(Dc \cdot g^{-1} \cdot Dc^\top)u = b$ for some $b$. The existing sparse linear system solvers can solve large classes of sparse linear system much faster than $O(n^3)$ time [13]. For $\log \operatorname{pdet} M(x)$, we show

$$\log \operatorname{pdet}(M(x)) = \log \det g(x) + \log \det \left(Dc(x) \cdot g(x)^{-1} \cdot Dc(x)^\top\right) - \log \det \left(Dc(x) \cdot Dc(x)^\top\right). \tag{2.8}$$

This simplification allows us to take advantage of sparse Cholesky decomposition. We prove (2.7) and (2.8) in Lemma 2 and Lemma 3 in Appendix B.2.1. The formulas (2.7) and (2.8) avoid the expensive pseudo-inverse and pseudo-determinant computations, and significantly improve the practical performance of our algorithm.

**Simplification for Subspace Constraints.** For the case $c(x) = Ax - b$, the Hamiltonian is now

$$H(x,v) = f(x) + \frac{1}{2} v^\top g^{-\frac{1}{2}} (I - P) g^{-\frac{1}{2}} v + \frac{1}{2} \left(\log \det g + \log \det Ag^{-1}A^\top - \log \det AA^\top\right) + \lambda^\top c,$$

where $P = g^{-\frac{1}{2}} A^\top (Ag^{-1}A^\top)^{-1} Ag^{-\frac{1}{2}}$. The key observation is that the algorithm only needs to know $x(h)$ in the HMC dynamics, and not $v(h)$. Thus, we can replace $H$ by any other that produces the same $x(h)$. We show in Lemma 4 (Appendix B.2.2) that the dynamics corresponding to $H$ above is equivalent to the dynamics that corresponds to a much simpler Hamiltonian:

$$H(x,v) = f(x) + \frac{1}{2} v^\top g^{-\frac{1}{2}} (I - P) g^{-\frac{1}{2}} v + \frac{1}{2} \left(\log \det g + \log \det Ag^{-1}A^\top\right).$$

Furthermore, we have

$$\frac{dx}{dt} = g^{-\frac{1}{2}} (I - P) g^{-\frac{1}{2}} v, \quad \frac{dv}{dt} = -\nabla f(x) + \frac{1}{2} Dg \left[\frac{dx}{dt}, \frac{dx}{dt}\right] - \frac{1}{2} \operatorname{Tr}(g^{-\frac{1}{2}} (I - P) g^{-\frac{1}{2}} Dg).$$

**Efficient Computation of Leverage Score.** Even after simplifying the Hamiltonian as above, we still have a term for the leverage scores, $\operatorname{Tr}(g^{-\frac{1}{2}} (I - P) g^{-\frac{1}{2}} Dg)$ in $\frac{dv}{dt}$ so that we need to compute the diagonal entries of $P = g^{-\frac{1}{2}} A^\top (Ag^{-1}A^\top)^{-1} Ag^{-\frac{1}{2}}$ to compute $\frac{dv}{dt}$. Since $(Ag^{-1}A^\top)^{-1}$ can be extremely dense even when $A$ is very sparse, a naive approach such as direct computation of the inverse can lead to a dense-matrix multiplication. To avoid dense-matrix multiplication, our approach is based on the fact that certain entries of $(Ag^{-1}A^\top)^{-1}$ can be computed as fast as computing sparse Cholesky decomposition of $Ag^{-1}A^\top$ [45, 5], which can be $O(n)$ time faster than computing $(Ag^{-1}A^\top)^{-1}$ in many settings. We first compute the Cholesky decomposition to obtain a sparse triangular matrix $L$ such that $LL^\top = Ag^{-1}A^\top$. Then, we show that only entries of $Ag^{-1}A^\top$ in $\operatorname{sp}(L) \cup \operatorname{sp}(L^\top)$ matter in computing $\operatorname{diag}(A^\top (Ag^{-1}A^\top)^{-1} A)$, where $\operatorname{sp}(L)$ is the sparsity pattern of $L$. We give the details of our approach in Appendix B.2.3.

## 2.3 Discretization

Explicit integrators such as leapfrog integrator, which are commonly used for Hamiltonian Monte Carlo, are no longer symplectic on general Riemannian manifolds (see Appendix C.1). Even though there have been some attempts [38] to make explicit integrators work in the Riemannian setting, its variants do not work for ill-conditioned problems.

Our algorithm uses the *implicit midpoint method* (Algorithm 3) to discretize the Hamiltonian process into steps of step size $h$ and run the process for $T$ iterations. This integrator is reversible and symplectic (so measure-preserving) [19], which allows us to use a Metropolis filter to ensure the

distribution is correct so that we no longer need to solve ODE to accuracy to maintain the correct stationary distribution. We write $H(x, v) = \overline{H}_1(x, v) + \overline{H}_2(x, v)$, where

$$\overline{H}_1(x, v) = f(x) + \frac{1}{2} \left( \log \det g(x) + \log \det Ag(x)^{-1}A^\top \right),$$

$$\overline{H}_2(x, v) = \frac{1}{2} v^\top g(x)^{-\frac{1}{2}} \left( I - P(x) \right) g(x)^{-\frac{1}{2}} v.$$

Starting from $(x_0, v_0)$, in the first step of the integrator, we run the process on the Hamiltonian $\overline{H}_1$ with step size $\frac{h}{2}$ to get $(x_{1/3}, v_{1/3})$. In the second step of the integrator, we run the process on $\overline{H}_2$ with step size $h$ by solving

$$x_{\frac{2}{3}} = x_{\frac{1}{3}} + h \frac{\partial \overline{H}_2}{\partial v} \left( \frac{x_{\frac{1}{3}} + x_{\frac{2}{3}}}{2}, \frac{v_{\frac{1}{3}} + v_{\frac{2}{3}}}{2} \right), \quad v_{\frac{2}{3}} = v_{\frac{1}{3}} - h \frac{\partial \overline{H}_2}{\partial x} \left( \frac{x_{\frac{1}{3}} + x_{\frac{2}{3}}}{2}, \frac{v_{\frac{1}{3}} + v_{\frac{2}{3}}}{2} \right),$$

iteratively using the Newton's method. This step involves computing the Cholesky decomposition of $(Ag^{-1}A^\top)^{-1}$ using the Cholesky decomposition of $Ag^{-1}A^\top$. In the third step, we run the process on the Hamiltonian $\overline{H}_1$ with step size $\frac{h}{2}$ again to get $(x_1, v_1)$.

We state the complete algorithm (Algorithm 2 and Algorithm 3) with details on the step size in Appendix C.1 and give the theoretical guarantees in Appendix C.2 (convergence of implicit midpoint method) and Appendix D (independence of condition number).

## 3 Experiments

In this section, we demonstrate the efficiency of our sampler using experiments on real-world datasets and compare our sampler with existing samplers. We demonstrate that CRHMC is able to sample larger models than previously known to be possible, and is significantly faster in terms of rate of convergence and sampling time in Section 3.2, along with convergence test in Section 3.4. We examine its behavior on benchmark instances such as simplices and Birkhoff polytopes in Section 3.3.

### 3.1 Experimental Setting

**Settings.** We performed experiments on the Standard DS12 v2 model from MS Azure cloud, which has a 2.1GHz Intel Xeon Platinum 8171M CPU and 28GB memory. In the experiments, we used our MATLAB and C++ implementation of CRHMC[3], which is available here and has been integrated into the COBRA toolbox.

We used twelve constraint-based metabolic models from molecular systems biology in the COBRA Toolbox v3.0 [21] and ten real-world LP examples randomly chosen from NETLIB LP test sets. A polytope from each model is defined by $\{x \in \mathbb{R}^n : Ax = b, l \le x \le u\}$ for $A \in \mathbb{R}^{m \times n}, b \in \mathbb{R}^m$, and $l, u \in \mathbb{R}^n$, which is input to CRHMC for uniform sampling. We describe in Appendix A how we preprocessed these dataset, along with full information about the datasets in Table 2.

**Comparison.** We used as a baseline the Coordinate Hit-and-Run (CHAR) implemented in two different languages. The former is Coordinate Hit-and-Run with Rounding (CHRR) written in MATLAB [11, 20] and the latter is the same algorithm (CDHR) with an R interface and a C++ library, VolEsti [6]. We refer readers to Appendix A for the details of these algorithms and our comparison setup. We note that popular sampling packages such as STAN and Pyro were not included in the experiments as they do not support constrained-based models. Even after transforming our dataset to their formats, the transformed dataset were too ill-conditioned for those algorithms to run. CHMC in [4] works only for manifolds implicitly defined by $\{c(x) = 0\}$ for continuously differentiable $c(x)$ with $Dc(x)$ full-rank everywhere, so we could not use it for comparison.

**Measurements.** To evaluate the quality of sampling methods, we measured two quantities, the *number of steps per effective sample* (i.e., mixing rate) and the *sampling time per effective sample*, $T_s$. The *effective sample size* (ESS)[4] can be thought of as the number of actual independent samples,

---

[3]Our package can be run to sample from general logconcave densities and has a feature for parallelization.
[4]We use the minimum of the ESS of each coordinate.

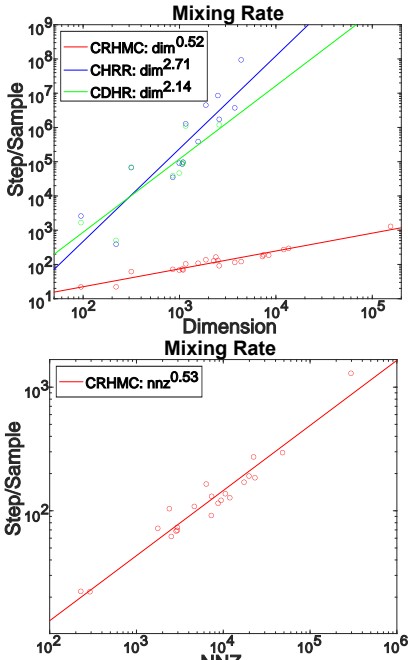

**Figure 3.1:** Mixing rate of CRHMC and the competitors. Mixing rate of CRHMC was sub-linear in dimension and the nnz of a preprocessed matrix A in a model, whereas the others needed quadratically many steps to converge to uniform distribution. In particular for our dataset, CRHMC mixed up to 6 orders of magnitude earlier than the others. Note that mixing rate of CHAR was very close to quadratic growth when using the full-dimensional scale (the first column in Table 2).

**Figure 3.2:** Sampling time of CRHMC and the competitors. The sampling time per effective sample of CRHMC was sub-quadratic in dimension and the nnz of a preprocessed matrix A in a model, while the others indicates at least a cubic dependency on dimension. In particular for our dataset, CRHMC was able to obtain a statistically independent sample up to 4 orders of magnitude faster than the others. This benefit of speed-up was actually straightforward from the figure, since CHRR could not obtain enough samples from instances with more than 5000 variables until it ran out of time.

taking into account correlation of samples from a target distribution. Thus the number of steps per effective sample is estimated by the total number of steps divided by the ESS, and the sampling time $T_s$ is estimated as the total sampling time until termination divided by the ESS.

Each algorithm attempted to draw 1000 uniform samples, with limits on running time set to 1 day (3 days for the largest instance *ken_18*) and memory usage to 6GB. If an algorithm passes either the time or the memory limit, we stop the algorithm and measure the quantities of interest based on samples drawn until that moment. After getting uniform samples, we thinned the samples twice to ensure independence of samples; first we computed the ESS of the samples, only kept ESS many samples, and repeated this again. We estimated the above quantities only if the ESS is more than 10 and an algorithm does not run into any error while running[5].

## 3.2 Mixing Rate and Sampling Time

**Sub-linear Mixing Rate.** We examined how the number of steps per effective sample grows with the number of nonzeros (nnz) of matrix $A$ (after preprocessing) and the number of variables (dimension in the plots). To this end, we counted the total number of steps taken until termination of algorithms and divided it by the effective sample size of drawn samples. Note that we thinned twice to ensure independence of samples used.

The mixing rate of CRHMC was sub-linear in both dimension and nnz, whereas previous implementations based on CHAR required at least $n^2$ steps per sample as seen in Figure 3.1. On the dataset,

---

[5]When running CDHR from the VolEsti package on some instances, we got an error message "R session aborted and R encountered a fatal error".

| Bio Model | Vars ($n$) | nnz | CRHMC | CHRR | CDHR | LP Model | Vars ($n$) | nnz | CRHMC | CHRR | CDHR |
|---|---|---|---|---|---|---|---|---|---|---|---|
| ecoli | 95 | 291 | 0.0098 | 0.0365 | 0.0022 | | | | | | |
| cardiac_mit | 220 | 228 | 0.0100 | 0.0059 | 0.0005 | israel | 316 | 2519 | 0.1186 | 1.2224 | 0.4426 |
| Aci_D21 | 851 | 1758 | 0.4257 | 0.6884 | 0.2974 | gfrd_pnc | 1160 | 2393 | 0.2199 | 40.988 | 18.468 |
| Aci_MR95 | 994 | 2859 | 0.9624 | 2.0668 | 0.5237 | 25fv47 | 1876 | 10566 | 0.8159 | 199.9 | - |
| Abi_49176 | 1069 | 2951 | 0.9608 | 1.9395 | 0.9622 | pilot_ja | 2267 | 11886 | 1.3490 | 5059* | - |
| Aci_20731 | 1090 | 2946 | 0.1540 | 2.3014 | 1.1086 | sctap2 | 2500 | 7334 | 0.6752 | 520.2 | - |
| Aci_PHEA | 1561 | 4640 | 0.3701 | 12.06 | - | ship08l | 4363 | 9434 | 0.6258 | 6512 | - |
| iAF1260 | 2382 | 6368 | 4.4355 | 3687.2 | - | cre_a | 7248 | 17368 | 2.2205 | 30455* | - |
| iJO1366 | 2583 | 7284 | 4.1608 | 70.5 | 35.556 | woodw | 8418 | 23158 | 2.0689 | 30307* | - |
| Recon1 | 3742 | 8717 | 0.7184 | 208.5 | - | 80bau3b | 12061 | 22341 | 11.881 | 47432* | - |
| Recon2 | 7440 | 19791 | 2.6116 | 10445* | - | ken_18 | 154699 | 295946 | 1616.3 | - | - |
| Recon3 | 13543 | 48187 | 31.114 | 29211* | - | | | | | | |

**Table 1:** Sampling time per effective sample of CHRR and CRHMC. We note that CRHMC is 1000 times faster than CHRR on the latest metabolic network (Recon3). Sampling time with asterisk (*) indicates that the effective sample size is less than 10.

mixing rate attained was up to 6 orders of magnitude faster for CRHMC compared to CHAR, implying that CRHMC converged to uniform distribution substantially faster than the other competitors. This gap in mixing rate increased super-linearly in dimension, enabling CRHMC to run on large instances of dimension up to 100000.

**Sub-quadratic Sampling Time.** We next examined the sampling time $T_s$ in terms of both the nnz of $A$ and the dimension of the instance. We computed the runtime of algorithms until their termination divided by the effective sample size of drawn samples, where we ignored the time it takes for preprocessing. Note that the sampling time $T_s$ is essentially multiplication of the mixing rate and the *per-step complexity* (i.e., how much time each step takes).

As shown in Figure 3.2 and Table 1, we found that the per-step complexity of CRHMC was small enough to make the sampling time sub-quadratic in both dimension and nnz, whereas CHAR had at least a cubic dependency on dimension, despite of a low per-step complexity. On our dataset, the sampling time of CRHMC was up to 4 orders of magnitude less than that of CHRR and CDHR. While CHRR can be used on dimension only up to a few thousands, increasing benefits of sampling time in higher dimension allows CRHMC to run on dimension up to 0.1 million.

### 3.3 CRHMC on Structured Instances

To see the behavior of CRHMC on very large instances, we ran the algorithm on three families of structured polytopes – hypercube, simplex, and Birkhoff polytope – up to dimension half-million. We attempted to draw 500 uniform samples with a 1 day time limit (except for 2 days for half-million-dimensional Birkhoff polytope). The definitions of these polytopes are shown in Appendix A.1.

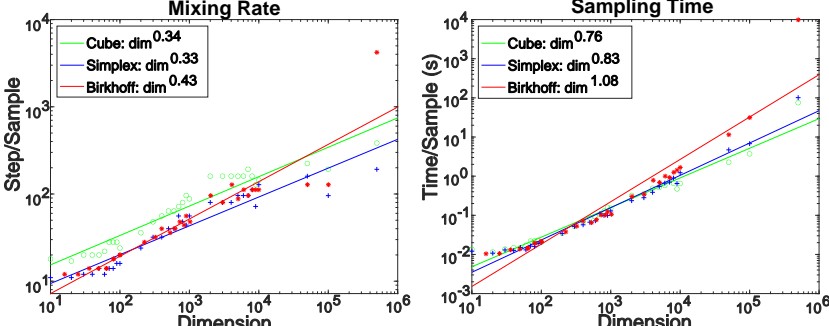

**Figure 3.3:** Mixing rate and sampling time on structured polytopes including hybercubes, simplices, and Birkhoff polytopes. CRHMC is scalable up to 0.5 million dimension on hypercubes and simplices and up to 0.1 million dimension on Birkhoff polytopes. We note that on the 0.5 million dimensional Birkhoff polytope the ESS is only 16, which is not reliable compared to the ESS on the other instances.

To the best of our knowledge, this is the first demonstration that it is possible to sample such a large model. As seen in Figure 3.3, CRHMC can scale smoothly up to half-million dimension on hypercubes and simplices and up to dimension $10^5$ for Birkhoff polytopes (we could not obtain a reliable estimate of mixing rate and sampling time on the half-million dimensional Birkhoff

polytope, as the ESS is only 16 after 2 days). However, we believe that one can find room for further improvement of CRHMC by tuning parameters or leveraging engineering techniques. We also expect that CRHMC enables us to estimate the volume of $B_n$ for $n \geq 20$, going well beyond the previously best possible dimension.

### 3.4 Uniformity Test

We used the following uniformity test to check whether samples from CRHMC form the uniform distribution over a polytope $P$: check that the fraction of the samples in the scaled set $x \cdot P$ is proportional to $x^{\text{dim}}$. As seen in Figure 3.4, the empirical CDFs of the radial distribution to the power of $(1/\text{dim})$ are close to the CDFs of the uniform distribution over those polytopes.

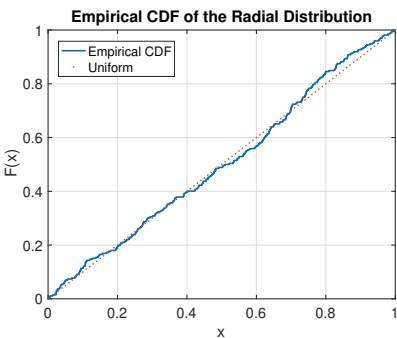 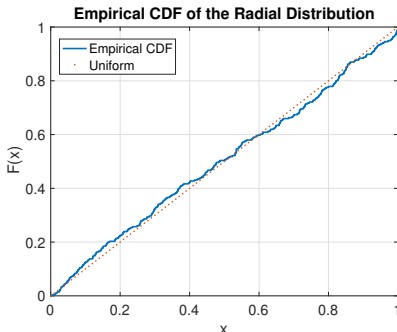

**Figure 3.4:** We plot the empirical cumulative distribution function of the radial distribution to the power of $(1/\text{dim})$ with 1000 ESS obtained by running CRHMC on *ATCC-49176* ($952 \times 1069$, left) and *Aci-PHEA* ($1319 \times 1561$, right), and in the plot $x$-axis is the scaling factor. We can observe the CDFs are very close to the CDFs of the uniform distribution over the polytopes defined by two instances.

**Acknowledgement.** The authors are grateful to Ben Cousins for helpful discussions, and to Ronan Fleming, Ines Thiele and their research groups for advice on metabolic models. This work was supported in part by NSF awards DMS-1839116, DMS-1839323, CCF-1909756, CCF-2007443 and CCF-2134105.

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
