| Aci_D21 | 851 | 1758 | 0.4257 | 0.6884 | 0.2974 |
| Aci_MR95 | 994 | 2859 | 0.9624 | 2.0668 | 0.5237 |
| Abi_49176 | 1069 | 2951 | 0.9608 | 1.9395 | 0.9622 |
| Aci_20731 | 1090 | 2946 | 0.1540 | 2.3014 | 1.1086 |
| Aci_PHEA | 1561 | 4640 | 0.3701 | 12.06 | - |
| iAF1260 | 2382 | 6368 | 4.4355 | 3687.2 | - |
| iJO1366 | 2583 | 7284 | 4.1608 | 70.5 | 35.556 |
| Recon1 | 3742 | 8717 | 0.7184 | 208.5 | - |
| Recon2 | 7440 | 19791 | 2.6116 | 10445* | - |
| Recon3 | 13543 | 48187 | 31.114 | 29211* | - |

| LP Model | Vars ($n$) | nnz | CRHMC | CHRR | CDHR |
|---|---|---|---|---|---|
| israel | 316 | 2519 | 0.1186 | 1.2224 | 0.4426 |
| gfrd_pnc | 1160 | 2393 | 0.2199 | 40.988 | 18.468 |
| 25fv47 | 1876 | 10566 | 0.8159 | 199.9 | - |
| pilot_ja | 2267 | 11886 | 1.3490 | 5059* | - |
| sctap2 | 2500 | 7334 | 0.6752 | 520.2 | - |
| ship08l | 4363 | 9434 | 0.6258 | 6512 | - |
| cre_a | 7248 | 17368 | 2.2205 | 30455* | - |
| woodw | 8418 | 23158 | 2.0689 | 30307* | - |
| 80bau3b | 12061 | 22341 | 11.881 | 47432* | - |
| ken_18 | 154699 | 295946 | 1616.3 | - | - |

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

# A    Additional Experiment Details

**Dataset.**    We summarize in Table 2 the dataset used in experiments. If a model is unbounded, we make it bounded by setting $l = \max(l, -10^7)$ and $u = \min(u, 10^7)$. As existing packages require full-dimensional representations of polytopes (i.e., $\{x : A'x \leq b'\}$), we transformed all constraint-based models to prepare instances for them as follows: (1) first preprocess each model by removing redundant constraints and appropriately scaling it, (2) find its corresponding full-dimensional description, and (3) round it via the maximum volume ellipsoid (MVE) algorithm making the polytope more amenable to sampling. We note that a full-dimensional polytope can be transformed into a constraint-based polytope and vice versa, so CRHMC can be run on either representation.

| Bio Model | Full-dim | Consts ($m$) | Vars ($n$) | nnz | | LP Model | Full-dim | Consts ($m$) | Vars ($n$) | nnz |
|---|---|---|---|---|---|---|---|---|---|---|
| ecoli | 24 | 72 | 95 | 291 | | LP Model | Full-dim | Consts ($m$) | Vars ($n$) | nnz |
| cardiac_mit | 12 | 230 | 220 | 228 | | israel | 142 | 174 | 316 | 2519 |
| Aci_D21 | 103 | 856 | 851 | 1758 | | gfrd_pnc | 544 | 616 | 1160 | 2393 |
| Aci_MR95 | 123 | 917 | 994 | 2859 | | 25fv47 | 1056 | 821 | 1876 | 10566 |
| Abi_49176 | 157 | 952 | 1069 | 2951 | | pilot_ja | 1002 | 940 | 2267 | 11886 |
| Aci_20731 | 164 | 1009 | 1090 | 2946 | | sctap2 | 1410 | 1090 | 2500 | 7334 |
| Aci_PHEA | 328 | 1319 | 1561 | 4640 | | ship08l | 2700 | 778 | 4363 | 9434 |
| iAF1260 | 572 | 1668 | 2382 | 6368 | | cre_a | 3703 | 3516 | 7248 | 17368 |
| iJO1366 | 590 | 1805 | 2583 | 7284 | | woodw | 4656 | 1098 | 8418 | 23158 |
| Recon1 | 932 | 2766 | 3742 | 8717 | | 80bau3b | 9233 | 2262 | 12061 | 22341 |
| Recon2 | 2430 | 5063 | 7440 | 19791 | | ken_18 | 49896 | 105127 | 154699 | 295946 |
| Recon3 | 5335 | 8399 | 13543 | 48187 | | | | | | |

**Table 2:** Constraint-based models. Each constraint-based model has a form of $\{x \in \mathbb{R}^n : Ax = b, l \leq x \leq u\}$ for $A \in \mathbb{R}^{m \times n}, b \in \mathbb{R}^m$ and $l, u \in \mathbb{R}^n$, where the rows and columns correspond to constraints and variables respectively. The full-dimension of each model is obtained by transforming its degenerate subspace to a full dimensional representation (i.e., $A'x \leq b'$), and we count the number of nonzero (nnz) entries of a preprocessed matrix $A$.

**Preprocessing.**    We preprocessed each constrained-based model prior to sampling. This preprocessing consists mainly of simplifying polytopes, scaling properly for numerical stability, and finding a feasible starting point. To simplify a given polytope, we check if $l_i = u_i$ for each $i \in [n]$ and then incorporate such variables $x_i$ into $Ax = b$. Any dense column is split into several columns with less non-zero entries by introducing additional variables. Then we remove dependent rows of $A$ by the Cholesky decomposition. Then we find the Dikin ellipsoid of the polytope. If the width along some axis is smaller than a preset tolerance, then we fix variables in such directions, reducing columns of $A$. Lastly, we run the primal-dual interior-point method with the log-barrier to find an analytic center of the polytope, which will be used as a starting point in sampling. When finding the analytic center of the simplified polytope, if a coordinate of the analytic center is too close to a boundary (to be

precise, smaller than a preset tolerance boundary $10^{-8}$), then we assume that the inequality constraint (either $x_i \leq u_i$ or $l_i \leq x_i$) is tight, and we collapse such a variable by moving it into the constraints $Ax = b$. We go back to the step for removing dependent rows and repeat until no more changes are made to $A$. Along with simplification, we keep rescaling $A, b, l, u$ for numerical stability.

**Coordinate Hit-and-Run (CDHR).**    We briefly explain how CHRR works. First, rounding via the MVE algorithm finds the maximum volume ellipsoid inscribed in the polytope and applies, to the polytope, an affine transformation that makes this ellipsoid a unit ball. This procedure puts a possibly highly-skewed polytope into John's position, which guarantees that the polytope contains a unit ball and is contained in a ball of radius $n$. This position still has a beneficial effect on sampling in practice in the sense that the random walk can converge in fewer steps. After the transformation, the random walk based on Coordinate Hit-and-Run (CHAR) chooses a random coordinate and moves to a random point on the line through the current point along the chosen coordinate.

When running CHRR and CDHR, we recorded a sample every $n^2$ steps. The mixing rate (i.e., the number of steps required to get a sample from a target distribution) of Hit-and-Run (HAR), a general version of CHAR choosing a random direction (unit vector) instead of a random coordinate, is $O^*(n^2 R^2)$ for a polytope $P$ with $B_n \subseteq P \subseteq R \cdot B_n$, where $B_n$ is the unit ball in $\mathbb{R}^n$ [34]. It was proved only recently that CHAR mixes in $O^*(n^9 R^2)$ steps on such a polytope [29, 35]. Even though this bound is not as tight as the mixing-rate bound for HAR, it was reported in [20] that CHRR mixes in the same number of steps as HAR empirically. Moreover, the per-step complexity of CHAR can be $n$ times faster than that of HAR, so CHAR brings a significant speed-up in practice.

**Comparison Setup.**    We set the parameters of CRHMC to values in `default_options.m` in the experiments. For the competitors, we proceeded with the following additional steps for fair comparison. First, as the VolEsti package does not support the MVE rounding, we rounded each polytope by the MVE algorithm in the CHRR package and then transformed the rounded polytope so that the R interface can read the data file. Next, we limited all algorithms to a single core, since the R interface uses a single core as a default whereas MATLAB uses as many available cores as possible.

## A.1   Polytope Definition

**Hypercube.**    The $n$-dimensional hypercube is defined by $\{x \in \mathbb{R}^n : -\frac{1}{2} \leq x_i \leq \frac{1}{2} \text{ for all } i \in [n]\}$. Note that it has no equality constraint and its full-dimension is $n$.

**Simplex.**    The $n$-dimensional simplex is defined by $\{x \in \mathbb{R}^n : 0 \leq x_i \text{ for all } i \in [n], \sum_{i=1}^{n} x_i = 1\}$. Note that its full-dimension is $n - 1$.

**Birkhoff Polytope.**    The $n^{th}$ Birkhoff polytope $B_n$ is the set of all doubly stochastic $n \times n$ matrices (or the convex hull of all permutation matrices), which is defined as

$$B_n = \{(X_{ij})_{i,j \in [n]} : \sum_j X_{ij} = 1 \text{ for all } i \in [n], \sum_i X_{ij} = 1 \text{ for all } j \in [n], \text{ and } X_{ij} \geq 0\}.$$

Namely, $B_n$ is defined in a constrained $\mathbb{R}^{n^2}$-dimensional space, and its full-dimension is $n^2 - (2n - 1) = (n-1)^2$. We ran CRHMC on $B_{\sqrt{n}}$ to examine its efficiency on (roughly) $n$-dimensional Birkhoff polytope.

## B   Deferred details of CRHMC

In this section, we present all technical details behind an *idealized* version of our algorithm, CRHMC, together with correctness of CRHMC. Subsequently in Appendix C, we provide details on a *discretized* version of CRHMC.

## B.1 Deferred details of Section 2.1

Recall that in Section 2.1 we mention that the following constrained Hamiltonian satisfies the Hamiltonian ODE $\left( \frac{dx}{dt} = \frac{\partial H(x,v)}{\partial v}, \frac{dv}{dt} = -\frac{\partial H(x,v)}{\partial x} \right)$:

$$H(x,v) = \overline{H}(x,v) + \lambda(x,v)^\top c(x) \quad \text{with} \quad \overline{H}(x,v) = f(x) + \frac{1}{2}v^\top M(x)^\dagger v + \log \mathrm{pdet}(M(x))$$

where

$$\lambda(x,v) = (Dc(x)Dc(x)^\top)^{-1}\left( D^2c(x)[v, \frac{dx}{dt}] - Dc(x)\frac{\partial \overline{H}(x,v)}{\partial x} \right).$$

**Lemma 1.** *Consider the constrained Hamiltonian defined by (2.5) with* $\mathrm{Range}(M(x)) = \mathrm{Null}(Dc(x))$ *and*

$$\lambda(x_t,v_t) = (Dc(x_t)Dc(x_t)^\top)^{-1}\left( D^2c(x_t)[v_t, \frac{dx_t}{dt}] - Dc(x_t)\frac{\partial \overline{H}(x_t,v_t)}{\partial x} \right).$$

*When the initial point satisfies* $c(x_0) = 0$, *the ODE solution of (2.2) satisfies* $c(x_t) = 0$ *and* $Dc(x_t)v_t = Dc(x_0)v_0$ *for all t.*

*Proof.* First we compute

$$\frac{d}{dt}c(x_t) = Dc(x_t) \cdot \frac{dx_t}{dt} = Dc(x_t) \cdot \frac{\partial H(x_t,v_t)}{\partial v_t}$$

$$= Dc(x_t)M(x_t)^\dagger v + Dc(x_t)D_v\lambda(x_t,v_t)^\top c(x_t)$$

$$= Dc(x_t)D_v\lambda(x_t,v_t)^\top c(x_t)$$

where we used $\mathrm{Range}(M(x)^\dagger) = \mathrm{Range}(M(x)) = \mathrm{Null}(Dc(x))$. Since $c(x_0) = 0$, by the uniqueness of the ODE solution, we have that $c(x_t) = 0$ for all $t$. Next we compute

$$\frac{dv_t}{dt} = -\frac{\partial H(x_t,v_t)}{\partial x}$$

$$= -\frac{\partial \overline{H}(x_t,v_t)}{\partial x} - Dc(x_t)^\top \lambda(x_t,v_t) - D_x\lambda(x_t,v_t)^\top c(x_t)$$

$$= -\frac{\partial \overline{H}(x_t,v_t)}{\partial x} - Dc(x_t)^\top \lambda(x_t,v_t)$$

where we used $c(x_t) = 0$. Hence, we have

$$\frac{d}{dt}Dc(x_t)v_t = D^2c(x_t)[v_t, \frac{dx_t}{dt}] + Dc(x_t)\frac{dv_t}{dt}$$

$$= D^2c(x_t)[v_t, \frac{dx_t}{dt}] - Dc(x_t)\frac{\partial \overline{H}(x_t,v_t)}{\partial x} - Dc(x_t)Dc(x_t)^\top\lambda(x_t,v_t).$$

By setting $\lambda(x_t,v_t) = (Dc(x_t)Dc(x_t)^\top)^{-1}(D^2c(x_t)[v_t, \frac{dx_t}{dt}] - Dc(x_t)\frac{\partial \overline{H}(x_t,v_t)}{\partial x})$, we have $\frac{d}{dt}Dc(x_t)v_t = 0$ and $Dc(x_t)v_t = Dc(x_0)v_0$ for all $t$ (i.e., $v_t \in \mathrm{Null}(Dc(x_t))$ during Step 2). $\square$

## B.2 Deferred details of Section 2.2

In Section 2.2, we mention that a naive algorithm computing $\partial H/\partial x$ and $\partial H/\partial v$ is bound to face the following challenges, especially in high-dimensional regime, and briefly explain how we address each of them. In this section, we give full details on our computational tricks.

> 1. Computation of the pseudo-inverse and its derivatives takes $O(n^3)$, except for very special matrices $\Longrightarrow$ Find equivalent formulas (Appendix B.2.1).
>
> 2. The Lagrangian term in the constrained Hamiltonian entails extra computation such as $D^2c(x) \Longrightarrow$ Simplify the constrained Hamiltonian (Appendix B.2.2).
>
> 3. A naive approach to computing leverage scores in $\partial H/\partial x$ results in a very dense matrix $\Longrightarrow$ Track sparsity pattern (Appendix B.2.3).

### B.2.1 Avoiding pseudo-inverse and pseudo-determinant

We start with a formula for $M(x)^\dagger$.

**Lemma 2.** *Let $M(x) = Q(x) \cdot g(x) \cdot Q(x)$ where $Q(x) = I - Dc(x)^\top (Dc(x) \cdot Dc(x)^\top)^{-1} Dc(x)$ is the orthogonal projection to the null space of $Dc(x)$. Then, $Dc(x) \cdot M(x)^\dagger = 0$ and $M(x)^\dagger = g(x)^{-\frac{1}{2}} \cdot (I - P(x)) \cdot g(x)^{-\frac{1}{2}}$ with*

$$P(x) = g(x)^{-\frac{1}{2}} \cdot Dc(x)^\top (Dc(x) \cdot g(x)^{-1} \cdot Dc(x)^\top)^{-1} Dc(x) \cdot g(x)^{-\frac{1}{2}}.$$

*Proof.* Recall that $\text{Range}(M(x)^\dagger) = \text{Range}(M(x))$. Hence, for any $u \in \mathbb{R}^n$, we have that $M(x)^\dagger u \in \text{Range}(M(x))$. Since $\text{Range}(M(x)) \subseteq \text{Range}(Q(x))$ and $\text{Range}(Q(x)) = \text{Null}(Dc(x))$ due to the definition of the orthogonal projection $Q(x)$, it follows that $Dc(x) \cdot M(x)^\dagger u = 0$ for all $u$.

For the formula of $M(x)^\dagger$, we simplify the notation by ignoring the parameter $x$. Let $N = g^{-\frac{1}{2}} P g^{-\frac{1}{2}}$ and $J = Dc(x)$. The goal is to prove that $M^\dagger = N$. First, we show some basic identities about $Q$ and $N$:

$$\begin{aligned}
QN &= Qg^{-\frac{1}{2}}(I - g^{-\frac{1}{2}} J^\top (Jg^{-1}J^\top)^{-1} Jg^{-\frac{1}{2}})g^{-\frac{1}{2}} \\
&= (I - J^\top (JJ^\top)^{-1} J)(g^{-1} - g^{-1}J^\top (Jg^{-1}J^\top)^{-1} Jg^{-1}) \\
&= g^{-1} - J^\top (JJ^\top)^{-1} Jg^{-1} \\
&\quad - (g^{-1}J^\top (Jg^{-1}J^\top)^{-1} Jg^{-1} - J^\top (JJ^\top)^{-1} Jg^{-1}J^\top (Jg^{-1}J^\top)^{-1} Jg^{-1}) \\
&= N. \tag{B.1}
\end{aligned}$$

Similarly, we have $NQ = N$, $QgN = Q$, and $NgQ = Q$. To prove that $M^\dagger = N$, we need to check that $MN$ and $NM$ are symmetric, $MNM = M$, and $NMN = N$.

For symmetry of $MN$ and $NM$, we note that $MN = QgQN = QgN = Q$ and $NM = NQgQ = NgQ = Q$. For the formula of $MNM$ and $NMN$, we note that that $Q$ is a projection matrix and hence

$$\begin{aligned}
MNM &= QM = QQgQ = QgQ = M, \\
NMN &= QN = N.
\end{aligned}$$

Therefore, we have $M^\dagger = N$. $\qquad\square$

Another bottleneck of the algorithm is to compute $\log \text{pdet} M(x)$. The next lemma shows a simpler formula that can take advantage of sparse Cholesky decomposition.

**Lemma 3.** *We have that*

$$\log \text{pdet}(M(x)) = \log \det g(x) + \log \det (Dc(x) \cdot g(x)^{-1} \cdot Dc(x)^\top) - \log \det (Dc(x) \cdot Dc(x)^\top).$$

*Proof.* We simplify the notation by ignoring the parameter $x$ and letting $J = Dc(x)$. Let

$$\begin{aligned}
f_1(g) &= \log \text{pdet}(Q \cdot g \cdot Q), \\
f_2(g) &= \log \det g + \log \det Jg^{-1}J^\top - \log \det JJ^\top.
\end{aligned}$$

Clearly, $f_1(I) = f_2(I) = 0$, and hence it suffices to prove that their derivatives are the same.

Note that $\text{Range}(Q \cdot g \cdot Q) = \text{Null}(J)$ and $\text{Range}(J^\top)$ is the orthogonal complement of $\text{Null}(J)$. Since $J^\top (JJ^\top)^{-1} J$ is the orthogonal projection to $\text{Range}(J^\top)$, all of its eigenvectors in $\text{Range}(J^\top)$ have eigenvalue 1 and all the rest in $\text{Null}(J)$ have eigenvalue 0. Therefore, by padding eigenvalue 1 on $\text{Range}(J^\top) = \text{Null}(J)^\perp = \text{Range}(QgQ)^\perp$, we have

$$\begin{aligned}
\text{pdet}(Q \cdot g \cdot Q) &= \det(Q \cdot g \cdot Q + J^\top (JJ^\top)^{-1} J) \\
&= \det(Q \cdot g \cdot Q + (I - Q)).
\end{aligned}$$

Using $D \log \det A(g)[u] = \text{Tr}(A(g)^{-1} DA(g)[u])$, the directional derivative of $f_1$ on direction $u$ is

$$Df_1(g)[u] = \text{Tr}((Q \cdot g \cdot Q + (I - Q))^{-1} Q \cdot u \cdot Q).$$

Let $N = (Q \cdot g \cdot Q)^\dagger$. As shown in the proof of Lemma 2, we have $NQ = QN = N$ and $QgN = Q$. By using these identities, we can manually check that $(Q \cdot g \cdot Q + (I - Q))^{-1} = N + (I - Q)$. Hence,

$$Df_1(g)[u] = \mathrm{Tr}\,((N + (I - Q))Q \cdot u \cdot Q) = \mathrm{Tr}(NuQ)$$
$$= \mathrm{Tr}(QNu) = \mathrm{Tr}(Nu)$$

where we used idempotence of the projection matrix $Q$ (i.e., $Q^2 = Q$).

On the other hand, we have

$$Df_2(g)[u] = \mathrm{Tr}(g^{-1}u) - \mathrm{Tr}\left((Jg^{-1}J^\top)^{-1}(Jg^{-1}ug^{-1}J^\top)\right)$$
$$= \mathrm{Tr}\left((g^{-1} - g^{-1}J^\top(Jg^{-1}J^\top)^{-1}Jg^{-1})u\right)$$
$$= \mathrm{Tr}(Nu)$$

where we used the alternative formula of $N$ in Lemma 2. This shows that the derivative of $f_1$ equals to that of $f_2$ at any point $g \succ 0$. Since the set of positive definite matrices is connected and $f_1(I) = f_2(I)$, this implies that $f_1(g) = f_2(g)$ for all $g \succ 0$. $\qquad\square$

Combining Lemma 2 and Lemma 3, we have the following formula of the Hamiltonian.

$$\overline{H}(x,v) = H_0(x,v) + \lambda(x,v)^\top c(x),$$
$$H_0(x,v) = f(x) + \frac{1}{2}v^\top g(x)^{-\frac{1}{2}}\left(I - g(x)^{-\frac{1}{2}} \cdot Dc(x)^\top(Dc(x) \cdot g(x)^{-1} \cdot Dc(x)^\top)^{-1}Dc(x) \cdot g(x)^{-\frac{1}{2}}\right)g(x)^{-\frac{1}{2}}v$$
$$+ \frac{1}{2}\left(\log\det g(x) + \log\det\left(Dc(x) \cdot g(x)^{-1} \cdot Dc(x)^\top\right) - \log\det\left(Dc(x) \cdot Dc(x)^\top\right)\right).$$

### B.2.2 Simplification for subspace constraints

For the case $c(x) = Ax - b$, the constrained Hamiltonian is

$$\overline{H}(x,v) = f(x) + \frac{1}{2}v^\top g^{-\frac{1}{2}}(I - P)g^{-\frac{1}{2}}v + \frac{1}{2}\left(\log\det g + \log\det Ag^{-1}A^\top - \log\det AA^\top\right) + \lambda^\top c \tag{B.2}$$

where $P = g^{-\frac{1}{2}}A^\top(Ag^{-1}A^\top)^{-1}Ag^{-\frac{1}{2}}$. The following lemma shows that the dynamics corresponding to $\overline{H}$ above is equivalent to a simpler Hamiltonian. The key observation is that the algorithm only needs to know $x(h)$ in the HMC dynamics, and not $v(h)$. Thus we can replace $\overline{H}$ by any other $H$ that produces the same $x(h)$.

**Lemma 4.** *The Hamiltonian dynamics of $x$ corresponding to (B.2) is same as the dynamics of $x$ corresponding to*

$$H(x,v) = f(x) + \frac{1}{2}v^\top g^{-\frac{1}{2}}(I - P)g^{-\frac{1}{2}}v + \frac{1}{2}\left(\log\det g + \log\det Ag^{-1}A^\top\right) \tag{B.3}$$

*where $P = g^{-\frac{1}{2}}A^\top(Ag^{-1}A^\top)^{-1}Ag^{-\frac{1}{2}}$. Furthermore, we have*

$$\frac{dx}{dt} = g^{-\frac{1}{2}}(I - P)g^{-\frac{1}{2}}v, \tag{B.4}$$
$$\frac{dv}{dt} = -\nabla f(x) + \frac{1}{2}Dg\left[\frac{dx}{dt}, \frac{dx}{dt}\right] - \frac{1}{2}\mathrm{Tr}(g^{-\frac{1}{2}}(I - P)g^{-\frac{1}{2}}Dg). \tag{B.5}$$

*Proof.* Note that the dynamics of $x$ corresponding to (B.2) is given by

$$\frac{dx}{dt} = \frac{\partial\overline{H}}{\partial v} = g^{-\frac{1}{2}}(I - P)g^{-\frac{1}{2}}v + (D_v\lambda)^\top c$$
$$= g^{-\frac{1}{2}}(I - P)g^{-\frac{1}{2}}v \tag{B.6}$$

where we used that $c(x) = 0$ (Lemma 1).

Now let us compute the dynamics of $v$. Note that

$$v^\top g^{-\frac{1}{2}}(I - P)g^{-\frac{1}{2}}v = v^\top g^{-1}v - v^\top g^{-1}A^\top(A \cdot g^{-1} \cdot A^\top)^{-1}Ag^{-1}v.$$

Hence, we have

$$D_x \left( \frac{1}{2} v^\top g^{-\frac{1}{2}} (I - P) g^{-\frac{1}{2}} v \right)$$

$$= -\frac{1}{2} v^\top g^{-1} \cdot Dg \cdot g^{-1} v + v^\top g^{-1} \cdot Dg \cdot g^{-1} A^\top (A \cdot g^{-1} \cdot A^\top)^{-1} A g^{-1} v$$

$$- \frac{1}{2} v^\top g^{-1} A^\top (A \cdot g^{-1} \cdot A^\top)^{-1} A \cdot g^{-1} \cdot Dg \cdot g^{-1} \cdot A^\top (A \cdot g^{-1} \cdot A^\top)^{-1} A g^{-1} v$$

$$= -\frac{1}{2} v^\top g^{-\frac{1}{2}} (I - P) g^{-\frac{1}{2}} \cdot Dg \cdot g^{-\frac{1}{2}} (I - P) g^{-\frac{1}{2}} v$$

$$= -\frac{1}{2} Dg \left[ \frac{dx}{dt}, \frac{dx}{dt} \right],$$

where we used $\frac{dx}{dt} = g^{-\frac{1}{2}} (I - P) g^{-\frac{1}{2}} v$ in (B.6). Therefore, it follows that

$$\frac{dv}{dt} = -\frac{\partial \overline{H}}{\partial x} - (D_v \lambda)^\top c - A^\top \lambda \tag{B.7}$$

$$= -\nabla f(x) + \frac{1}{2} Dg \left[ \frac{dx}{dt}, \frac{dx}{dt} \right] - \frac{1}{2} \mathrm{Tr}(g^{-1} Dg) \tag{B.8}$$

$$+ \frac{1}{2} \mathrm{Tr} \left( (Ag(x)^{-1} A^\top)^{-1} Ag(x)^{-1} \cdot Dg \cdot g(x)^{-1} A^\top \right) - A^\top \lambda$$

$$= -\nabla f(x) + \frac{1}{2} Dg \left[ \frac{dx}{dt}, \frac{dx}{dt} \right] - \frac{1}{2} \mathrm{Tr}(g^{-\frac{1}{2}} (I - P) g^{-\frac{1}{2}} Dg) - A^\top \lambda$$

where we used that $c = 0$ again in the second equality.

Recall that $\frac{dx}{dt} = g^{-\frac{1}{2}} (I - P) g^{-\frac{1}{2}} v$. In this formula, let us perturb $v$ by $A^\top y$ for any $y$ as follows.

$$(I - P) g^{-\frac{1}{2}} (v + A^\top y) = (I - P) g^{-\frac{1}{2}} v + \left( I - g^{-\frac{1}{2}} A^\top (Ag^{-1} A^\top)^{-1} Ag^{-\frac{1}{2}} \right) g^{-\frac{1}{2}} A^\top y$$

$$= (I - P) g^{-\frac{1}{2}} v + (g^{-\frac{1}{2}} A^\top y - g^{-\frac{1}{2}} A^\top (Ag^{-1} A^\top)^{-1} (Ag^{-1} A^\top) y)$$

$$= (I - P) g^{-\frac{1}{2}} v + (g^{-\frac{1}{2}} A^\top y - g^{-\frac{1}{2}} A^\top y)$$

$$= (I - P) g^{-\frac{1}{2}} v.$$

Hence, removing $A^\top \lambda$ from $\frac{dv}{dt}$ in (B.7) does not change the dynamics of $x$, and thus we have the new dynamics given simply by (B.4) and (B.5). By repeating this proof, one can check that the simplified Hamiltonian (B.3) also yields (B.4) and (B.5). $\qquad \square$

### B.2.3 Efficient Computation of Leverage Score

In this section, we discuss how we efficiently compute the diagonal entries of $A^\top (Ag^{-1} A^\top)^{-1} A$. Our idea is based on the fact that certain entries of $(Ag^{-1} A^\top)^{-1}$ can be computed as fast as computing sparse Cholesky decomposition of $Ag^{-1} A^\top$ [45, 5], which can be $O(n)$ time faster than computing $(Ag^{-1} A^\top)^{-1}$ in many settings.

For simplicity, we focus on the case $g(x)$ as a diagonal matrix, since we use the log-barrier $\phi(x) = -\sum_{i=1}^m (\log(x_i - l_i) + \log(u_i - x_i))$ in implementation. We first note that we maintain a "sparsity pattern" $\mathrm{sp}(M)$ of a sparse matrix $M$ so that we handle only these entries in downstream tasks. The sparsity pattern indicates "candidates" of nonzero entries of a matrix (i.e., $\mathrm{sp}(M) \supseteq nnz(M) = \{(i,j) : M_{ij} \neq 0\}$). For instance, it is obvious that $\mathrm{sp}(cc^\top) = \{(i,j) : c_i c_j \neq 0\} = nnz(cc^\top)$ for a column vector $c$ and that $\mathrm{sp}(Ag^{-1} A^\top) = \bigcup_{i \in [n]} \mathrm{sp}(A_i A_i^\top)$ follows from the equality $Ag^{-1} A^\top = \sum_{i=1}^n (Ag^{-\frac{1}{2}})_i (Ag^{-\frac{1}{2}})_i^\top$, where $M_i$ denote the $i^{th}$ column of $M$ (See Theorem 2.1 in [12]). Then we compute the Cholesky decomposition to obtain a sparse triangular matrix $L$ such that $LL^\top = Ag^{-1} A^\top$ with a property $\mathrm{sp}(Ag^{-1} A^\top) \subseteq \mathrm{sp}(L^\top) \cup \mathrm{sp}(L)$ (See Theorem 4.2 in [12]).

Once the sparsity pattern of $L$ is identified, we compute $S := (Ag^{-1} A^\top)^{-1}|_{\mathrm{sp}(L)}$, the restriction of $S$ to $\mathrm{sp}(L)$, that is, the inverse matrix $S$ is computed only for entries in $\mathrm{sp}(L)$. [45, 5] showed that this matrix $S$ can be computed as fast as the Cholesky decomposition of $Ag^{-1} A^\top$.

For completeness, we explain how they compute $S$ efficiently. Let $L_0 D L_0^\top$ be the LDL decomposition of $Ag^{-1}A^\top$ such that the diagonals of $L_0$ is one and so $L = L_0 D^{\frac{1}{2}}$, and it easily follows that

$$S = D^{-1}L_0^{-1} + (I - L_0^\top)S = D^{-\frac{1}{2}}L^{-1} + (I - L^\top D^{-\frac{1}{2}})^{-1}S.$$

Since $D^{-1}L_0^{-1}$ is lower triangular and $I - L_0^\top$ is strictly upper triangular, symmetry of $S$ implies that $S$ can be computed from the bottom row to the top row one by one. We note that the computation of $S$ on any entry in $\mathrm{sp}(L)$ only requires previously computed $S$ on entries in $\mathrm{sp}(L)$, due to the sparsity pattern of $I - L^\top D^{-\frac{1}{2}}$. [45, 5] showed that the total cost of computing $S$ is $O(\sum_{i=1}^n n_i^2)$ for backward substitution, where $n_i$ is the number of nonzeros in the $i^{th}$ column of $L$. This exactly matches the cost of computing $L$. In our experiments, for many sparse matrices $A$, we found that $O(\sum_{i=1}^n n_i^2)$ is roughly $O(n^{1.5})$ and it is much faster than dense matrix inverse.

We have presented methods to save computational cost, avoiding full computation of the inverse $(Ag^{-1}A^\top)^{-1}$. This attempt is justified by the fact that only entries of $Ag^{-1}A^\top$ in $\mathrm{sp}(L) \cup \mathrm{sp}(L^\top)$ matter in computing $\mathrm{diag}(A^\top SA) = \mathrm{diag}(A^\top(Ag^{-1}A^\top)^{-1}A)$.

**Lemma 5.** *Computation of* $\mathrm{diag}(A^\top(Ag^{-1}A^\top)^{-1}A)$ *involves accessing only entries of* $(Ag^{-1}A^\top)^{-1}$ *in* $\mathrm{sp}(Ag^{-1}A^\top)$.

*Proof.* Let $M := (Ag^{-1}A^\top)^{-1} \in \mathbb{R}^{m \times m}$, $\sigma_i := (A^\top(Ag^{-1}A^\top)^{-1}A)_{ii}$ for $i \in [n]$, and $a_i$ be the $i^{th}$ column of $A$. Observe that

$$\sigma_i = a_i^\top(Ag^{-1}A^\top)^{-1}a_i = \mathrm{Tr}(a_i^\top Ma_i) = \mathrm{Tr}(Ma_ia_i^\top).$$

As the entries of $M$ only in $\mathrm{sp}(a_ia_i^\top)$ matter when computing the trace, we have that all the entries of $M$ used for computing $\sigma_i$ for all $i \in [n]$ are included in $\bigcup_{i=1}^n \mathrm{sp}(a_ia_i^\top) = \mathrm{sp}(Ag^{-1}A^\top)$. $\square$

Now let us divide the diagonals of $S$ by 2. Then we have $(Ag^{-1}A^\top)^{-1}|_{\mathrm{sp}(L)\cup\mathrm{sp}(L^\top)} = S + S^\top$ and thus

$$\mathrm{diag}(A^\top(Ag^{-1}A^\top)^{-1}A) = \mathrm{diag}(A^\top(Ag^{-1}A^\top)^{-1}|_{\mathrm{sp}(L)\cup\mathrm{sp}(L^\top)}A)$$
$$= \mathrm{diag}(A^\top SA + A^\top S^\top A) = 2 \cdot \mathrm{diag}(A^\top SA)$$

and the last term can be computed efficiently using $S$. In our experiment, the cost of computing leverage score is roughly twice the cost of computing Cholesky decomposition in all datasets.

Finally, we discuss another approach to compute leverage score with the same asymptotic complexity. We consider the function

$$V(g) = \log\det Ag^{-1}A^\top$$

where $g$ is a sparse matrix $g \in \mathbb{R}^{\mathrm{sp}(g)}$ and $V$ is defined only on $\mathbb{R}^{\mathrm{sp}(g)}$. Note that $V(g)$ can be computed using Cholesky decomposition of $A^\top g^{-1}A^\top$ and multiplying the diagonal of the decomposition. Next, we note that

$$\nabla V(g) = -(g^{-1}A^\top(Ag^{-1}A^\top)^{-1}Ag^{-1})|_{\mathrm{sp}(g)}.$$

Hence, we can compute leverage score by first computing $\nabla V(g)$ via automatic differentiation, and the time complexity of computing $\nabla V$ is only a small constant factor more than the time complexity of computing $V$ [18]. The only problem with this approach is that the Cholesky decomposition algorithm is an algorithm involving a large loop and sparse operations and existing automatic differentiation packages are not efficient to differentiate such functions.

### B.3 Stationarity of CRHMC

Now, the ideal CRHMC (or the continuous CRHMC) is the same as Algorithm 1 with the simplified constrained Hamiltonian $H$ in place of the unconstrained Hamiltonian. In this section, we prove that the Markov chain defined by the ideal CRHMC projected to $x$ satisfies detailed balance with respect to its target distribution proportional to $e^{-f(x)}$ subject to $c(x) = 0$, leading to the target distribution being stationary.

To this end, we introduce a few notations here. Let $\mathcal{M} = \{x \in \mathbb{R}^n : c(x) = 0\}$ be a manifold in $\mathbb{R}^n$ and $\pi(x)$ be a desired distribution on $\mathcal{M}$ proportional to $e^{-f(x)}$ satisfying $\int_{\mathcal{M}} \pi(x)dx = 1$

(to be precise, the Radon-Nikodym derivative of $\pi$ w.r.t. the Hausdorff measure on the manifold $\mathcal{M}$ is proportional to $e^{-f(x)}$). We denote the set of velocity $v$ at $x \in \mathcal{M}$ (i.e., cotangent space) by $\mathcal{T}_x\mathcal{M} = \mathrm{Null}(Dc(x)) = \{v \in \mathbb{R}^n : Dc(x)M(x)^{\dagger}v = 0\}$. Let $T_h$ be the map sending $(x, v)$ to $(x', v') = (x(h), y(h))$ in the Hamiltonian ODE (Step 2 of Algorithm 1) and define $F_{x,h}(v) := (\pi_1 \circ T_h)(x, v) := x'$, where $\pi_1(x, v) := x$ is the projection to the position space $x$. For a matrix $A$, we denote by $|A|$ the absolute value of its determinant $|\det(A)|$.

Note that we check the detailed balance of the induced chain on the "original $(x)$" space without moving to the "phase $(x, v)$" space, unlike Brubaker's proof [4].

**Theorem 6.** *For $x, x' \in \mathcal{M}$, let $\mathbf{P}_x(x')$ be the probability density of the one-step distribution to $x'$ starting at $x$ in CRHMC (i.e., transition kernel from $x$ to $x'$). It satisfies detailed balance with respect to the desired distribution $\pi$ (i.e., $\pi(x)\mathbf{P}_x(x') = \pi(x')\mathbf{P}_{x'}(x)$).*

*Proof.* Fix $x$ and $x'$ in $\mathcal{M}$. Let $C_1$ be the normalization constant of $e^{-f(x)}$ (i.e., $\pi(x) = C_1 e^{-f(x)}$). The transition kernel $\mathbf{P}_x(x')$ is characterized as the pushforward by $F_{x,h}$ of the probability measure $v \sim \mathcal{N}(0, M(x))$ on $\mathcal{T}_x\mathcal{M}$, so it follows that

$$\mathbf{P}_x(x') = C_2 \int_{V_x} \frac{e^{-\frac{1}{2}\log\mathrm{pdet}(M(x)) - \frac{1}{2}v^{\top}M(x)^{\dagger}v}}{|DF_{x,h}(v)|} dv,$$

where $C_2$ is the normalization constant of $e^{-\frac{1}{2}\log\mathrm{pdet}(M(x)) - \frac{1}{2}v^{\top}M(x)^{\dagger}v}$ and $V_x = \{v \in \mathcal{T}_x\mathcal{M} : F_{x,h}(v) = x'\}$ is the set of velocity in cotangent space at $x$ such that the Hamiltonian ODE with step size $h$ sends $(x, v)$ to $(x', v')$. (Further details for deducing the 1-step distribution can be found in Lemma 10 of [31]) As $c(x) = 0$ for $x \in \mathcal{M}$, it follows that

$$\pi(x)\mathbf{P}_x(x')$$
$$= C_1 C_2 \int_{V_x} \frac{e^{-f(x) - \frac{1}{2}\log\mathrm{pdet}(M(x)) - \frac{1}{2}v^{\top}M(x)^{\dagger}v - \lambda(x,v)^{\top}c(x)}}{|DF_{x,h}(v)|} dv = C_1 C_2 \int_{V_x} \frac{e^{-H(x,v)}}{|DF_{x,h}(v)|} dv.$$

Going forward, we use three important properties of the Hamiltonian dynamics including reversibility, Hamiltonian preservation, and volume preservation, which still hold for the constrained Hamiltonian $H$. Due to reversibility $T_{-h}(x', v') = (x, v)$, we can write

$$\pi(x')\mathbf{P}_{x'}(x) = C_1 C_2 \int_{V_{x'}} \frac{e^{-H(x',v')}}{|DF_{x',-h}(v')|} dv',$$

where $V_{x'} = \{v' \in \mathcal{T}_{x'}\mathcal{M} : F_{x',-h}(v') = x\}$ is the counterpart of $V_x$. From reversibility $T_{-h} \circ T_h = I$, the inverse function theorem implies $DT_{-h} = (DT_h)^{-1}$. Now let us denote

$$DT_h(x, v) = \begin{bmatrix} A & B \\ C & D \end{bmatrix} \quad \& \quad DT_{-h}(x', v') = \begin{bmatrix} A' & B' \\ C' & D' \end{bmatrix},$$

where each entry is a block matrix with the same size. Note that $DF_{x,h}(v) = B$ and $DF_{x',-h}(v') = B'$ hold by the definition of Jacobian. Together with $DT_{-h} = (DT_h)^{-1}$, a formula for the inverse of a block matrix results in

$$|DF_{x',-h}(v')| = |B'| = \frac{|B|}{|D||A - BD^{-1}C|} = \frac{|B|}{|DT_h(x,v)|} = |B| = |DF_{x,h}(v)|,$$

where we use the property of volume preservation in the fourth equality (i.e., $|DT_h(x, v)| = 1$). Finally, the property of Hamiltonian preservation implies $H(x, v) = H(x', v')$ and thus

$$\int_{V_{x'}} \frac{e^{-H(x',v')}}{|DF_{x',-h}(v')|} dv' = \int_{V_x} \frac{e^{-H(x,v)}}{|DF_{x,h}(v)|} dv.$$

Therefore, $\pi(x)\mathbf{P}_x(x') = \pi(x')\mathbf{P}_{x'}(x)$ holds. $\qquad\square$

Similar reasoning as Theorem 3 and Lemma 1 in [4] gives $\pi$-irreducibility and aperiodicity of the process, so CRHMC converges to the unique stationary distribution $\pi \propto e^{-f(x)}$.

## C  Discretization

We discuss how to implement our Hamiltonian dynamics using the implicit midpoint method in Section C.1 and present theoretical guarantees of correctness and efficiency of the discretized CRHMC in Section C.2.

### C.1  Discretized CRHMC based on Implicit Midpoint Integrator

In our algorithm, we discretize the Hamiltonian process into steps of step size $h$ and run the process for $T$ iterations (see Algorithm 2). Rather than resampling the velocity at every step, we may change the velocity more gradually, using the following update:

$$v' \leftarrow \sqrt{\beta}v + \sqrt{1-\beta}z$$

where $z \sim \mathcal{N}(0, M(x))$ and $\beta$ is a parameter. We note that this step is time-reversible, i.e., $\mathbf{P}(v|x)\mathbf{P}(v \to v') = \mathbf{P}(v'|x)\mathbf{P}(v' \to v)$ (see Theorem 8). Starting from $(x^{(0)}, v^{(0)})$, let $(x^{(t)}, v^{(t)})$ be the point obtained after iteration $t$. In the beginning of each iteration, we compute the Cholesky decomposition of $Ag(x)^{-1}A^\top$ for later use and resample the velocity with momentum. As noted previously in Lemma 4, for $c(x) = Ax - b$ we can just use the simplified Hamiltonian in (B.3),

$$H(x,v) = f(x) + \frac{1}{2}v^\top g(x)^{-\frac{1}{2}}\left(I - P(x)\right)g(x)^{-\frac{1}{2}}v + \frac{1}{2}\left(\log\det g(x) + \log\det Ag(x)^{-1}A^\top\right)$$

instead of the constrained Hamiltonian $H + \lambda^\top c$. We solve the Hamiltonian dynamics for $H$ by the implicit midpoint method, which we will discuss below, and then use a Metropolis filter on $H$ to ensure the distribution is correct.

**Implicit Midpoint Method.**  For general Riemannian manifolds, explicit integrators such as the leapfrog method (LM) are not symplectic, unlike IMM. LM is symplectic when the Hamiltonian equations are separable (i.e., each of $dx/dt$ and $dv/dt$ is a function of either x or v only). However, in the general Riemannian manifold setting, where $dx/dt$ depends on position $x$ due to mass matrices (which is $g(x)$ in our paper) as well as velocity $v$, the Hamiltonian is no longer separable, which prevents us from using LM. We refer interested readers to Section 3 and Section 4.1 in [10].

We now elaborate on how the implicit midpoint integrator works (see Algorithm 3), which is symplectic (so measure-preserving) and reversible [19]. Let us write $H(x,v) = \overline{H}_1(x,v) + \overline{H}_2(x,v)$, where

$$\overline{H}_1(x,v) = f(x) + \frac{1}{2}\left(\log\det g(x) + \log\det Ag(x)^{-1}A^\top\right),$$

$$\overline{H}_2(x,v) = \frac{1}{2}v^\top g(x)^{-\frac{1}{2}}\left(I - P(x)\right)g(x)^{-\frac{1}{2}}v.$$

Starting from $(x_0, v_0)$, in the first step of the integrator, we run the process on the Hamiltonian $\overline{H}_1$ with step size $\frac{h}{2}$ to get $(x_{1/3}, v_{1/3})$, and this discretization leads to $x_{1/3} = x_0 + \frac{h}{2}\frac{\partial\overline{H}_1}{\partial v}(x_0, v_0)$ and $v_{1/3} = v_0 - \frac{h}{2}\frac{\partial\overline{H}_1}{\partial x}(x_0, v_0)$. Note that $x_{1/3} = x_0$ due to $\frac{\partial\overline{H}_1}{\partial v} = 0$. In the second step of the integrator, we run the process on $\overline{H}_2$ with step size $h$ by solving

$$x_{\frac{2}{3}} = x_{\frac{1}{3}} + h\frac{\partial\overline{H}_2}{\partial v}\left(\frac{x_{\frac{1}{3}} + x_{\frac{2}{3}}}{2}, \frac{v_{\frac{1}{3}} + v_{\frac{2}{3}}}{2}\right),$$

$$v_{\frac{2}{3}} = v_{\frac{1}{3}} - h\frac{\partial\overline{H}_2}{\partial x}\left(\frac{x_{\frac{1}{3}} + x_{\frac{2}{3}}}{2}, \frac{v_{\frac{1}{3}} + v_{\frac{2}{3}}}{2}\right).$$

To this end, starting from $x_{2/3} = x_{1/3}$ and $v_{2/3} = v_{1/3}$, we apply $x_{2/3} \leftarrow x_{1/3} + h\frac{\partial\overline{H}_2}{\partial v}\left(\frac{x_{1/3}+x_{2/3}}{2}, \frac{v_{1/3}+v_{2/3}}{2}\right)$ and $v_{2/3} \leftarrow v_{1/3} - h\frac{\partial\overline{H}_2}{\partial x}\left(\frac{x_{1/3}+x_{2/3}}{2}, \frac{v_{1/3}+v_{2/3}}{2}\right)$ iteratively with the following subroutine for computing $\frac{\partial\overline{H}_2}{\partial v}$ and $\frac{\partial\overline{H}_2}{\partial x}$. According to Lemma 4, this computation involves solving $g(x)^{-1}A^\top\left(Ag(x)^{-1}A^\top\right)^{-1}Ag(x)^{-1}v$ for some $v$ and $x$. To compute $\left(Ag(x)^{-1}A^\top\right)^{-1}Ag(x)^{-1}v$, we use the Newton's method, which iteratively computes $\nu \leftarrow \nu + M^{-1}Ag(x)^{-1}\left(v - A^\top\nu\right)$ for some $M$. Note that the Newton's method guarantees that $\nu$

converges to $M^{-1}Ag(x)^{-1}v$ if $M$ is invertible. Here, we choose $M = Ag(x^{(t)})^{-1}A^\top$ to ensure fast convergence. Since we have already computed the Cholesky decomposition of $M$ in the beginning, $M^{-1}Ag(x)^{-1}\left(v - A^\top\nu\right)$ can be computed efficiently by backward and forward substitution. In the third step of the integrator, we run the process on the Hamiltonian $\overline{H}_1$ with step size $\frac{h}{2}$ again to get $(x_1, v_1)$, which results in $x_1 = x_{2/3}$ and $v_1 = v_{2/3} - \frac{h}{2}\frac{\partial \overline{H}_1}{\partial x}(x_1, v_{2/3})$.

We note that CRHMC is affine-invariant and provably independent of condition number (Theorem 15), and thus the step size and momentum only need to depend on the dimension. In practice, we set the momentum to roughly $1 - h$, and for the step size $h$, we decrease it until the acceptance probability is close enough to 1 during the warm-up phase. Empirically, we found that the step size stays between 0.05 and 0.2 in practice even for high dimensional ill-conditioned polytopes. This step size is remarkable, given that for these instances a standard package like STAN ends up selecting a small step size like $10^{-8}$ and thus fails to converge.

Putting Algorithm 2 and Algorithm 3 together, we obtain discretization of constrained Riemannian Hamiltonian Monte Carlo algorithm.

---

**Algorithm 2:** `Discretized Constrained Riemannian Hamiltonian Monte Carlo with Momentum`

**Input:** Initial point $x^{(0)}$, velocity $v^{(0)}$, record frequency $T$, step size $h$, ODE steps $K$
**for** $t = 1, 2, \cdots, T$ **do**

    Let $\overline{v} = v^{(t-1)}$ and $x = x^{(t-1)}$.
    // Step 1: Resample $v$ with momentum
    Let $z \sim \mathcal{N}(0, M(x))$. Update $\overline{v}$:

$$v \leftarrow \sqrt{\beta}\overline{v} + \sqrt{1 - \beta}z.$$

    // Step 2: Solve $\frac{dx}{dt} = \frac{\partial H(x,v)}{\partial v}$, $\frac{dv}{dt} = -\frac{\partial H(x,v)}{\partial x}$ via the implicit midpoint method
    Use `Implicit Midpoint Method`$(x, v, h, K)$ to find $(x', v')$ such that

$$v_{\frac{1}{3}} = v - \frac{h}{2}\frac{\partial \overline{H}_1(x, v)}{\partial x}, \tag{C.1}$$

$$x' = x + h\frac{\partial \overline{H}_2\left(\frac{x+x'}{2}, \frac{v_{1/3}+v_{2/3}}{2}\right)}{\partial v}, \quad v_{\frac{2}{3}} = v_{\frac{1}{3}} - h\frac{\partial \overline{H}_2\left(\frac{x+x'}{2}, \frac{v_{1/3}+v_{2/3}}{2}\right)}{\partial x},$$

$$v' = v_{\frac{2}{3}} - \frac{h}{2}\frac{\partial \overline{H}_1(x', v_{\frac{2}{3}})}{\partial x}.$$

    // Step 3: Filter
    With probability $\min\left\{1, \frac{e^{-H(x',v')}}{e^{-H(x,v)}}\right\}$, set $x^{(t)} \leftarrow x'$ and $v^{(t)} \leftarrow v'$.
    Otherwise, set $x^{(t)} \leftarrow x$ and $v^{(t)} \leftarrow -v$.
**end**
**Output:** $x^{(T)}$

---

### C.2 Theoretical Guarantees

In terms of efficiency, we first show that one iteration of Algorithm 2 incurs the cost of solving a few Cholesky decomposition and $O(K)$ sparse triangular systems. We also show in Lemma 9 that the implicit midpoint integrator converges to the solution of Eq. (C.1) in logarithmically many iterations. Regarding correctness, Theorem 8 and Lemma 9 together show that the discretized CRHMC (Algorithm 2) converges to the stationary distribution indeed (see Remark 10).

**Theorem 7.** *The cost of each iteration of Algorithm 2 is solving $O(1)$ Cholesky decomposition and $O(K)$ triangular systems, where $K$ is the number of iterations in Algorithm 3.*

---

**Algorithm 3:** `Implicit Midpoint Method`

---

**Input:** Initial point $x$, velocity $v$, step size $h$, ODE steps $K$

`// Step 1: Solve` $\frac{dx}{dt} = \frac{\partial \overline{H}_1(x,v)}{\partial v}$, $\frac{dv}{dt} = -\frac{\partial \overline{H}_1(x,v)}{\partial x}$

Set $x_{\frac{1}{3}} \leftarrow x$ and $v_{\frac{1}{3}} \leftarrow v - \frac{h}{2}\frac{\partial \overline{H}_1(x,v)}{\partial x}$.

`// Step 2: Solve` $\frac{dx}{dt} = \frac{\partial \overline{H}_2(x,v)}{\partial v}$, $\frac{dv}{dt} = -\frac{\partial \overline{H}_2(x,v)}{\partial x}$ `via implicit midpoint`

Set $\nu \leftarrow 0$.

**for** $k = 1, 2, \cdots, K$ **do**

> Let $x_{\mathrm{mid}} \leftarrow \frac{1}{2}\left(x_{\frac{1}{3}} + x_{\frac{2}{3}}\right)$ and $v_{\mathrm{mid}} \leftarrow \frac{1}{2}\left(v_{\frac{1}{3}} + v_{\frac{2}{3}}\right)$
>
> Set $\nu \leftarrow \nu + \left(LL^\top\right)^{-1} Ag(x_{\mathrm{mid}})^{-1}\left(v_{\mathrm{mid}} - A^\top \nu\right)$
>
> Set $x_{\frac{2}{3}} \leftarrow x_{\frac{1}{3}} + hg(x_{\mathrm{mid}})^{-1}\left(v_{\mathrm{mid}} - A^\top \nu\right)$
>
> and $v_{\frac{2}{3}} \leftarrow v_{\frac{1}{3}} + \frac{h}{2}Dg(x_{\mathrm{mid}})\left[g(x_{\mathrm{mid}})^{-1}\left(v_{\mathrm{mid}} - A^\top \nu\right), g(x_{\mathrm{mid}})^{-1}\left(v_{\mathrm{mid}} - A^\top \nu\right)\right]$

**end**

`// Step 3: Solve` $\frac{dx}{dt} = \frac{\partial \overline{H}_1(x,v)}{\partial v}$, $\frac{dv}{dt} = -\frac{\partial \overline{H}_1(x,v)}{\partial x}$

Set $x_1 \leftarrow x_{\frac{2}{3}}$ and $v_1 \leftarrow v_{\frac{2}{3}} - \frac{h}{2}\frac{\partial \overline{H}_1}{\partial x}(x_{\frac{2}{3}}, v_{\frac{2}{3}})$.

**Output:** $x_1, v_1$

---

*Proof.* We first solve the Cholesky decomposition to get $L_{t-1}L_{t-1}^\top = Ag(x^{(t-1)})^{-1}A^\top$ at the beginning of iteration. Recall that

$$
H(x,v) = \overline{H}_1(x,v) + \overline{H}_2(x,v)
$$
$$
= \left(f(x) + \frac{1}{2}(\log \det g(x) + \log \det Ag(x)^{-1}A^\top)\right)
$$
$$
+ \left(\frac{1}{2}v^\top g(x)^{-\frac{1}{2}}\left(I - g(x)^{-\frac{1}{2}}A^\top (Ag(x)^{-1}A^\top)^{-1} Ag(x)^{-\frac{1}{2}}\right) g(x)^{-\frac{1}{2}}v\right).
$$

The value of $H(x^{(t-1)}, v^{(t-1)})$ should be computed later for the filter step and can be efficiently computed by the given $L_{t-1}L_{t-1}^\top = Ag(x^{(t-1)})^{-1}A^\top$ and solving two sparse triangular systems (i.e., $L_{t-1}^{-\top}(L_{t-1}^{-1}(Ag(x)^{-\frac{1}{2}}))$). We need the same cost (i.e., Cholesky decomposition and solving two triangular systems) for the value of $H(x', v')$, where $(x', v')$ is the output of Algorithm 3. We note that $L$ inherits sparsity of $A$ and thus each triangular system can be solved efficiently by backward and forward substitution.

In the implicit midpoint method, one main component is computation of $\frac{\partial \overline{H}_1(x,v)}{\partial x}$ in Step 1 and $\frac{\partial \overline{H}_1}{\partial x}(x_{\frac{2}{3}}, v_{\frac{2}{3}})$ in Step 3 due to leverage scores. As seen in Section B.2.3, the cost for these computations is within a constant factor of solving the Cholesky decomposition for $Ag(x^{(t-1)})^{-1}A^\top$ and $Ag(x_{\frac{2}{3}})^{-1}A^\top$. Another component is solving $O(K)$ triangular systems to update $\nu$ in Step 2.

Adding up all these costs, each iteration of Algorithm 2 only requires solving $O(1)$ Cholesky decomposition and $O(K)$ sparse triangular systems. $\qquad \square$

**Theorem 8.** *The Markov chain defined by Algorithm 2 projected to $x$ has a stationary density proportional to $\exp(-f(x))$, and is irreducible and aperiodic. Therefore, this Markov chain converges to the stationary distribution.*

*Proof.* Each iteration consists of two stages: resampling velocity with momentum in Step 1 (i.e., $(x, \overline{v})$ to $(x, v)$) and solving ODE followed by the filter in Step 2 and 3 (i.e., $(x, v)$ to $(x', v')$). To prove the claim, we show that Step 1 is time-reversible with respect to the conditional distribution $\pi(v|x)$ and that Step 2 followed by Step 3 is also time-reversible with respect to $\pi(x, v)$.

We begin with the first part. We have $\pi(\overline{v}|x) = \mathcal{N}(0, M(x))$ due to the definition of $H$. Since $\overline{v}|x \sim \mathcal{N}(0, M(x))$ and $z \sim \mathcal{N}(0, M(x))$ are independent Gaussians, the update rule $v = \sqrt{\beta}\overline{v} + \sqrt{1 - \beta}z$

implies $\pi(v|x) = \mathcal{N}(0, M(x))$. Let $\mathbf{P}(z)$ be the probability density and $C$ be the normalization constant for Gaussian $\mathcal{N}(0, M(x))$. Then, the time-reversibility w.r.t. $\pi(v|x)$ is immediate from the following computation:

$$\pi(\overline{v}|x)\mathbf{P}(\overline{v} \to v) = C^2 \exp(-\frac{1}{2}\overline{v}^\top M^\dagger \overline{v}) \cdot \exp(-\frac{1}{2}\frac{(v - \sqrt{\beta}\overline{v})^\top M^\dagger (v - \sqrt{\beta}\overline{v})}{1 - \beta})$$

$$= C^2 \exp\left(-\frac{1}{2}\left(\overline{v}^\top M^\dagger \overline{v} + \frac{v^\top M^\dagger v}{1 - \beta} + \frac{\beta \overline{v}^\top M^\dagger \overline{v}}{1 - \beta} - \frac{\sqrt{\beta}}{1 - \beta}(\overline{v}^\top M^\dagger v + v^\top M^\dagger \overline{v})\right)\right)$$

$$= C^2 \exp\left(-\frac{1}{2}\left(\frac{v^\top M^\dagger v}{1 - \beta} + \frac{\overline{v}^\top M^\dagger \overline{v}}{1 - \beta} - \frac{\sqrt{\beta}}{1 - \beta}(\overline{v}^\top M^\dagger v + v^\top M^\dagger \overline{v})\right)\right),$$

$$\pi(v|x)\mathbf{P}(v \to \overline{v}) = C^2 \exp(-\frac{1}{2}v^\top M^\dagger v) \cdot \exp(-\frac{1}{2}\frac{(\overline{v} - \sqrt{\beta}v)^\top M^\dagger (\overline{v} - \sqrt{\beta}v)}{1 - \beta})$$

$$= C^2 \exp\left(-\frac{1}{2}\left(v^\top M^\dagger v + \frac{\overline{v}^\top M^\dagger \overline{v}}{1 - \beta} + \frac{\beta v^\top M^\dagger v}{1 - \beta} - \frac{\sqrt{\beta}}{1 - \beta}(\overline{v}^\top M^\dagger v + v^\top M^\dagger \overline{v})\right)\right)$$

$$= C^2 \exp\left(-\frac{1}{2}\left(\frac{v^\top M^\dagger v}{1 - \beta} + \frac{\overline{v}^\top M^\dagger \overline{v}}{1 - \beta} - \frac{\sqrt{\beta}}{1 - \beta}(\overline{v}^\top M^\dagger v + v^\top M^\dagger \overline{v})\right)\right)$$

$$\implies \quad \pi(\overline{v}|x)\mathbf{P}(\overline{v} \to v) = \pi(v|x)\mathbf{P}(v \to \overline{v}).$$

The second part follows from a stronger statement due to symmetry of $v$ in $H(x, v)$: In the space where $(x, v)$ and $(x, -v)$ are identified, the Markov chain defined by Step 2 and 3 satisfies detailed balance with respect the density $\pi([x, v])$ proportional to $\exp(-H(x, v))$, where $[x, v]$ denotes the identified point for $(x, v)$ and $(x, -v)$. Consider the pairs $[x, v] = \{(x, v), (x, -v)\}$ and $[x', v'] = \{(x', v'), (x', -v')\}$ where in Step 2 $(x, v)$ goes to $(x', v')$ and $(x', -v')$ goes to $(x, -v)$ due to reversibility of the implicit midpoint method. We now verify that the filtering probability is the same in either direction, using the measure-preserving property of Step 2

$$\pi(x, v)\mathbf{P}\left((x, v) \to (x', v')\right) = \pi(x, v)\min\left\{1, \frac{\pi(x', v')}{\pi(x, v)}\right\}$$

$$= \min\left\{\pi(x, v), \pi(x', v')\right\}$$

$$= \min\left\{\pi(x, -v), \pi(x', -v')\right\}$$

$$= \pi(x', -v')\min\left\{1, \frac{\pi(x, -v)}{\pi(x', -v')}\right\}$$

$$= \pi(x', -v')\mathbf{P}\left((x', -v') \to (x, -v)\right).$$

Therefore, for any two pairs $[x, v]$ and $[x', v']$, we have $\pi([x, v])\mathbf{P}\left([x, v] \to [x', v']\right) = \pi([x', v']\mathbf{P}\left([x', v'] \to [x, v]\right)$, and thus this detailed balance implies that the target density is stationary.

Its irreducibility is implied by the non-zero lower bound on the conductance of the discretized CRHMC (Theorem 15). To see this, let $A$ and $B$ be two subsets of positive measure such that one subset is not reachable from another in infinitely many steps. Take the set $R$ of reachable points from $A$ via running the Markov chain, and note that $R$ and $R^c(\supseteq B)$ have non-zero measures. However, the non-zero conductance, meaning that there must be a positive probability of stepping out of $R$, which contradicts the definition of $R$. Now for aperiodicity, as assumed at the beginning of the mixing rate proof (Appendix D), we consider a lazy version of the discretized CRHMC instead, which makes the chain stay where it is at with probability $1/2$ at each iteration, which prevents potential periodicity of the process. Note that this modification worsens the mixing rate only by a factor of 2.

Putting these three together, we can show that the discretized CRHMC converges to the target distribution. $\qquad \square$

Now we show in Lemma 9 that the implicit midpoint method (Algorithm 3) converges to the solution of (C.1) in logarithmically many iterations. To show the convergence of Algorithm 3, we denote by $\mathcal{T}$ the map induced by one iteration of Step 2.

**Definition 1.** *Let*

$$\mathcal{T}(x,v,\nu) = \begin{pmatrix} x_{\frac{1}{3}} + hg(x_{mid})^{-1}(v_{mid} - A^\top\lambda_1) \\ v_{\frac{1}{3}} + \frac{h}{2}Dg(x_{mid})[g(x_{mid})^{-1}(v_{mid} - A^\top\lambda_1), g(x_{mid})^{-1}(v_{mid} - A^\top\lambda_1)] \\ \lambda_1 \end{pmatrix},$$

*where* $x_{mid} = \frac{1}{2}(x_{\frac{1}{3}} + x)$, $v_{mid} = \frac{1}{2}(v_{\frac{1}{3}} + v)$, *and* $\lambda_1 = \nu + (LL^\top)^{-1}Ag(x_{mid})^{-1}\left(v_{mid} - A^\top\nu\right)$.
*Let* $(x^*_{\frac{2}{3}}, v^*_{\frac{2}{3}}, \nu^*)$ *be the fixed point of* $\mathcal{T}$.

We assume that $g$ is given by the Hessian of a highly self-concordant barrier $\phi$ (see E.2). Note that the log-barrier is highly self-concordant. We can show that for small enough step size $h$, Algorithm 3 can solve (C.1) to $\delta$-accuracy in logarithmically many iterations.

**Lemma 9.** *Suppose* $g(x) = \nabla^2\phi(x)$ *for some highly self-concordant barrier* $\phi$. *For any input* $(x_{\frac{1}{3}}, v_{\frac{1}{3}})$, *let* $(x^{(k)}_{\frac{2}{3}}, v^{(k)}_{\frac{2}{3}}, \nu^{(k)})$ *be points obtained after* $k$ *iterations in Step 2 of Algorithm 3. Let* $(\widetilde{x}_{\frac{2}{3}}, \widetilde{v}_{\frac{2}{3}})$ *be the solution for* $(x_{\frac{2}{3}}, v_{\frac{2}{3}})$ *in the following equation*

$$x_{\frac{2}{3}} = x_{\frac{1}{3}} + h\frac{\partial\overline{H}_2}{\partial v}\left(\frac{x_{\frac{1}{3}} + x_{\frac{2}{3}}}{2}, \frac{v_{\frac{1}{3}} + v_{\frac{2}{3}}}{2}\right), \quad v_{\frac{2}{3}} = v_{\frac{1}{3}} - h\frac{\partial\overline{H}_2}{\partial x}\left(\frac{x_{\frac{1}{3}} + x_{\frac{2}{3}}}{2}, \frac{v_{\frac{1}{3}} + v_{\frac{2}{3}}}{2}\right).$$

*Let* $\|x\|_A := \sqrt{x^\top A x}$ *for a matrix* $A$. *For any* $(x,v,\nu)$, *define the norm*

$$\|(x,v,\lambda)\| := \|x\|_{g(x_{\frac{1}{3}})} + \|v\|_{g(x_{\frac{1}{3}})^{-1}} + h\|A^\top\nu\|_{g(x_{\frac{1}{3}})^{-1}}.$$

*If* $\left\|(x^{(0)}_{\frac{2}{3}}, v^{(0)}_{\frac{2}{3}}, \nu^{(0)}) - (\widetilde{x}_{\frac{2}{3}}, \widetilde{v}_{\frac{2}{3}}, \nu^*)\right\| \leq r$ *with* $h \leq r \leq \min(\frac{1}{10}, \frac{\sqrt{h}}{4}, \frac{\|v^*\|_{g(x_0)^{-1}}}{4})$, *then*

$$\left\|(x^{(L)}_{\frac{2}{3}}, v^{(L)}_{\frac{2}{3}}, \nu^{(L)}) - (\widetilde{x}_{\frac{2}{3}}, \widetilde{v}_{\frac{2}{3}}, \nu^*)\right\| \leq \delta$$

*for some* $L = O\left(\log_{1/C}\frac{r}{\delta}\right)$, *where* $C = O_n(h)$ *is the Lipschitz constant of the map* $\mathcal{T}$.

*Proof.* Since $(x^*_{\frac{2}{3}}, v^*_{\frac{2}{3}}, \nu^*)$ is the fixed point of $\mathcal{T}$ (i.e., $\nu^* = \lambda_1$), we have

$$\nu^* = \nu^* + (LL^\top)^{-1}Ag(x_{\text{mid}})^{-1}\left(v_{\text{mid}} - A^\top\nu^*\right)$$

and thus $Ag(x_{\text{mid}})^{-1}v_{\text{mid}} = Ag(x_{\text{mid}})^{-1}A^\top\nu^*$. For invertible $Ag(x_{\text{mid}})^{-1}A^\top$, we have

$$\nu^* = \left(Ag(x_{\text{mid}})^{-1}A^\top\right)^{-1}Ag(x_{\text{mid}})^{-1}v_{\text{mid}}.$$

Similarly by using the definition of the fixed point and this new formula for $\nu^*$,

$$\begin{aligned}
x^*_{\frac{2}{3}} &= x_{\frac{1}{3}} + hg(x_{\text{mid}})^{-1}v_{\text{mid}} - hg(x_{\text{mid}})^{-1}A^\top\nu^* \\
&= x_{\frac{1}{3}} + hg(x_{\text{mid}})^{-1}v_{\text{mid}} - hg(x_{\text{mid}})^{-1}A^\top\left(Ag(x_{\text{mid}})^{-1}A^\top\right)^{-1}Ag(x_{\text{mid}})^{-1}v_{\text{mid}} \\
&= x_{\frac{1}{3}} + h\frac{\partial\overline{H}_2}{\partial v}(x_{\text{mid}}, v_{\text{mid}})
\end{aligned}$$

and

$$\begin{aligned}
v^*_{\frac{2}{3}} &= v_{\frac{1}{3}} + \frac{h}{2}Dg(x_{\text{mid}})[g(x_{\text{mid}})^{-1}(v_{\text{mid}} - A^\top\nu^*), g(x_{\text{mid}})^{-1}(v_{\text{mid}} - A^\top\nu^*)] \\
&= v_{\frac{1}{3}} - h\frac{\partial\overline{H}_2}{\partial x}(x_{\text{mid}}, v_{\text{mid}})
\end{aligned}$$

which shows that $(x^*_{\frac{2}{3}}, v^*_{\frac{2}{3}})$ is exactly the solution for $(x,v)$ in the equation

$$x = x_{\frac{1}{3}} + h\frac{\partial\overline{H}_2}{\partial v}\left(\frac{x_{\frac{1}{3}} + x}{2}, \frac{v_{\frac{1}{3}} + v}{2}\right), \quad v = v_{\frac{1}{3}} - h\frac{\partial\overline{H}_2}{\partial x}\left(\frac{x_{\frac{1}{3}} + x}{2}, \frac{v_{\frac{1}{3}} + v}{2}\right).$$

Next, we show that the iterations in Step 2 converges to $(x_{\frac{2}{3}}^*, v_{\frac{2}{3}}^*, \nu^*)$. If $\left\|(x_{\frac{2}{3}}^{(0)}, v_{\frac{2}{3}}^{(0)}, \nu^{(0)}) - (x_{\frac{2}{3}}^*, v_{\frac{2}{3}}^*, \nu^*)\right\| \le r$ for some $C = O_n(h)$, we have

$$
\begin{aligned}
\left\|(x_{\frac{2}{3}}^{(\ell)}, v_{\frac{2}{3}}^{(\ell)}, \nu^{(\ell)}) - (x_{\frac{2}{3}}^*, v_{\frac{2}{3}}^*, \nu^*)\right\| &= \left\|\mathcal{T}(x_{\frac{2}{3}}^{(\ell-1)}, v_{\frac{2}{3}}^{(\ell-1)}, \nu^{(\ell)}) - \mathcal{T}(x_{\frac{2}{3}}^*, v_{\frac{2}{3}}^*, \nu^*)\right\| \\
&\le C \left\|(x_{\frac{2}{3}}^{(\ell-1)}, v_{\frac{2}{3}}^{(\ell-1)}, \nu^{(\ell-1)}) - (x_{\frac{2}{3}}^*, v_{\frac{2}{3}}^*, \nu^*)\right\| \\
&\le C^\ell \left\|(x_{\frac{2}{3}}^{(0)}, v_{\frac{2}{3}}^{(0)}, \nu^{(0)}) - (x_{\frac{2}{3}}^*, v_{\frac{2}{3}}^*, \nu^*)\right\|,
\end{aligned}
$$

where the first equality follows from $(x_{\frac{2}{3}}^*, v_{\frac{2}{3}}^*, \nu^*)$ is the fixed point of $\mathcal{T}$ and the second inequality follows from Lemma 12. Therefore, we have $\left\|(x_{\frac{2}{3}}^{(L)}, v_{\frac{2}{3}}^{(L)}, \nu^{(L)}) - (x_{\frac{2}{3}}^*, v_{\frac{2}{3}}^*, \nu^*)\right\| \le \delta$ for $L = O\left(\log_C \frac{r}{\delta}\right)$. $\qquad\square$

*Remark* 10. Lemma 9 shows that Algorithm 3 converges to the solution of (C.1) in logarithmically many iterations for small enough step size $h$. In Step 1 of Algorithm 2, $v$ is resampled so that every iteration of Algorithm 2 is a non-degenerate map. Then, the total variation distance between the distributions generated by solving (C.1) using Algorithm 3 and solving (C.1) exactly in one iteration of Algorithm 2 can be bounded by error due to Algorithm 3. Theorem 8 shows that the process will converge to the exact stationary distribution. Therefore, in order for the accumulated error of Algorithm 2 to remain bounded for polynomially many steps, it suffices to run logarithmically many iterations in Algorithm 3. Any small bias due to the numerical error in the ODE computation is corrected by the filter, and maintaining as small error as possible is important to keep the acceptance probability high.

## C.3 Deferred Proof

**Lemma 11** ([28], Lemma 28). *Suppose $g(x) = \nabla^2\phi(x)$ for some highly self-concordance barrier $\phi$. Then, we have that*

- $(1 - \|y - x\|_{g(x)})^2 g(x) \preceq g(y) \preceq \frac{1}{(1 - \|y - x\|_{g(x)})^2} g(x)$.

- $\|Dg(x)[v, v]\|_{g(x)^{-1}} \le 2\|v\|_{g(x)}^2$.

- $\|Dg(x)[v, v] - Dg(y)[v, v]\|_{g(x)^{-1}} \le \frac{6}{(1 - \|y - x\|_{g(x)})^3} \|v\|_{g(x)}^2 \|y - x\|_{g(x)}$.

- $\|Dg(x)[v, v] - Dg(x)[w, w]\|_{g(x)^{-1}} \le 2 \|v - w\|_{g(x)} \|v + w\|_{g(x)}$.

**Lemma 12.** *Let $g(x) = \nabla^2\phi(x)$ for some highly self-concordance barrier $\phi$. Given $x_0, v_0$ and $L$ such that $LL^\top = Ag(x_0)^{-1}A^\top$, consider the map*

$$
\mathcal{T}(x, v, \lambda) = \begin{pmatrix} x_0 + hg(x_{1/2})^{-1}(v_{1/2} - A^\top\lambda_1) \\ v_0 + \frac{h}{2}Dg(x_{1/2})[g(x_{1/2})^{-1}(v_{1/2} - A^\top\lambda_1), g(x_{1/2})^{-1}(v_{1/2} - A^\top\lambda_1)] \\ \lambda_1 \end{pmatrix}
$$

*where $x_{1/2} = (x_0 + x)/2$, $v_{1/2} = (v_0 + v)/2$ and $\lambda_1 = \lambda + (LL^\top)^{-1}Ag(x_{1/2})^{-1}\left(v_{1/2} - A^\top\lambda\right)$. Let $(x^*, v^*, \lambda^*)$ be a fixed point of $\mathcal{T}$. For any $x, v, \lambda$, we define the norm*

$$
\|(x, v, \lambda)\| = \|x\|_{g(x_0)} + \|v\|_{g(x_0)^{-1}} + h\|A^\top\lambda\|_{g(x_0)^{-1}}.
$$

*Let $\Omega = \{(x, v, \lambda) : \|(x, v, \lambda) - (x^*, v^*, \lambda^*)\| \le r\}$ with $h \le r \le \min(\frac{1}{10}, \frac{\sqrt{h}}{4}, \frac{\|v^*\|_{g(x_0)^{-1}}}{4})$. Suppose that $(x_0, v_0, 0) \in \Omega$. Then, for any $(x, v, \lambda), (\overline{x}, \overline{v}, \overline{\lambda}) \in \Omega$, we have*

$$
\|\mathcal{T}(x, v, \lambda) - \mathcal{T}(\overline{x}, \overline{v}, \overline{\lambda})\| \le C\|(x, v, \lambda) - (\overline{x}, \overline{v}, \overline{\lambda})\|
$$

*where $C = (\frac{3r}{h} + \|v^*\|_{g(x_0)^{-1}})(400r + 18h\|v^*\|_{g(x_0)^{-1}})$.*

*Remark* 13. Note that we should think $r = \Theta_n(h)$ because that is the distance between $(x_0, v_0, 0)$ and $(x^*, v^*, \lambda^*)$. In that case, the Lipschitz constant of $\mathcal{T}$ is $O_n(h\|v^*\|_{g(x_0)^{-1}}^2) = O_n(h)$. Hence, if the step size $h$ is small enough, then $\mathcal{T}$ is a contractive mapping. In practice, we can take $h$ close to a constant because $g$ is decomposable into barriers in each dimension and the bound can be improved using this.

*Proof.* We use $\mathcal{T}(x,v,\lambda)_x$ to denote the $x$ component of $\mathcal{T}(x,v,\lambda)$ and similarly for $\mathcal{T}(x,v,\lambda)_v$ and $\mathcal{T}(x,v,\lambda)_\lambda$. For simplicity, we write $g_0 = g(x_0)$, $g_{1/2} = g(x_{1/2})$ and $\overline{g}_{1/2} = g(\overline{x_{1/2}})$. By the assumption, we have that

$$\|x - x_0\|_{g_0} \leq \|x - x^*\|_{g_0} + \|x^* - x_0\|_{g_0} \leq 2r.$$

Similarly, $\|\overline{x} - x_0\|_{g_0} \leq 2r$.

We first bound $\mathcal{T}(x,v,\lambda)_\lambda$. Note that

$$\mathcal{T}(\overline{x},\overline{v},\overline{\lambda})_\lambda - \mathcal{T}(x,v,\lambda)_\lambda = \alpha_1 + \alpha_2 + \alpha_3 + \alpha_4$$

where

$$\alpha_1 = (I - (LL^\top)^{-1}Ag_0^{-1}A^\top)(\overline{\lambda} - \lambda),$$
$$\alpha_2 = (LL^\top)^{-1}Ag_0^{-1}(\overline{v_{1/2}} - v_{1/2}),$$
$$\alpha_3 = (LL^\top)^{-1}A(g_{1/2}^{-1} - g_0^{-1})((\overline{v_{1/2}} - A^\top\overline{\lambda}) - (v_{1/2} - A^\top\lambda)),$$
$$\alpha_4 = (LL^\top)^{-1}A(\overline{g}_{1/2}^{-1} - g_{1/2}^{-1})(\overline{v_{1/2}} - A^\top\overline{\lambda}).$$

Using that $LL^\top = Ag(x_0)^{-1}A^\top$, we have $\alpha_1 = 0$. For $\alpha_2$, we have

$$\|A^\top\alpha_2\|_{g_0^{-1}}^2 = (\overline{v_{1/2}} - v_{1/2})^\top g_0^{-1}A^\top(LL^\top)^{-1}Ag_0^{-1}A^\top(L^\top L)^{-1}Ag_0^{-1}(\overline{v_{1/2}} - v_{1/2})$$
$$= (\overline{v_{1/2}} - v_{1/2})^\top g_0^{-1}A^\top(Ag_0^{-1}A^\top)^{-1}Ag_0^{-1}(\overline{v_{1/2}} - v_{1/2})$$
$$\leq (\overline{v_{1/2}} - v_{1/2})^\top g_0^{-1}(\overline{v_{1/2}} - v_{1/2})$$
$$= \frac{1}{4}\|\overline{v} - v\|_{g_0^{-1}}^2$$

where we use $LL^\top = Ag(x_0)^{-1}A^\top$ and $g_0^{-1/2}A^\top(Ag_0^{-1}A^\top)^{-1}Ag_0^{-1/2} = B^\top(BB^\top)^{-1}B \preceq I$ for $B = Ag_0^{-1/2}$. For $\alpha_3$, by self-concordance of $g$ (Lemma 11) and $\|x - x_0\|_{g_0} \leq 2r$, we have

$$(1-r)^2g_0 \preceq g_{1/2} \preceq \frac{1}{(1-r)^2}g_0 \tag{C.2}$$

and hence $(g_0^{1/2}(g_{1/2}^{-1} - g_0^{-1})g_0^{1/2})^2 \preceq ((1-r)^{-2} - 1)^2 I$. Using this and $P = g_0^{-1/2}A^\top(Ag_0^{-1}A^\top)^{-1}Ag_0^{-1/2} \preceq I$, we have

$$\|A^\top\alpha_3\|_{g_0^{-1}} = \|g_0^{1/2}(g_{1/2}^{-1} - g_0^{-1})((\overline{v_{1/2}} - A^\top\overline{\lambda}) - (v_{1/2} - A^\top\lambda))\|_P$$
$$\leq \|g_0^{1/2}(g_{1/2}^{-1} - g_0^{-1})((\overline{v_{1/2}} - A^\top\overline{\lambda}) - (v_{1/2} - A^\top\lambda))\|_2$$
$$\leq ((1-r)^{-2} - 1)\|g_0^{-1/2}((\overline{v_{1/2}} - A^\top\overline{\lambda}) - (v_{1/2} - A^\top\lambda))\|_2$$
$$\leq ((1-r)^{-2} - 1)(\frac{1}{2}\|\overline{v} - v\|_{g_0^{-1}} + \|A^\top(\overline{\lambda} - \lambda)\|_{g_0^{-1}}).$$

Using $r \leq 1/10$, we have

$$\|A^\top\alpha_3\|_{g_0^{-1}} \leq 1.2r\|\overline{v} - v\|_{g_0^{-1}} + 2.4r\|A^\top(\overline{\lambda} - \lambda)\|_{g_0^{-1}}.$$

For $\alpha_4$, similarly, we have

$$\|A^\top\alpha_4\|_{g_0^{-1}} \leq ((1 - 0.5\|\overline{x} - x\|_{g_{1/2}})^{-2} - 1)\|\overline{v_{1/2}} - A^\top\overline{\lambda}\|_{g_0^{-1}}$$
$$\leq ((1 - 0.6\|\overline{x} - x\|_{g_0})^{-2} - 1)\|\overline{v_{1/2}} - A^\top\overline{\lambda}\|_{g_0^{-1}}$$
$$\leq 1.5\|\overline{x} - x\|_{g_0}\|\overline{v_{1/2}} - A^\top\overline{\lambda}\|_{g_0^{-1}}$$

where we used $g_{1/2} \preceq 1.2g_0$ (by (C.2)) in the second inequality and $\|\overline{x} - x\|_{g_0} \leq \|\overline{x} - x^*\|_{g_0} + \|x - x^*\|_{g_0} \leq \frac{1}{5}$ at the end. Combining everything, we have

$$\|A^\top(\mathcal{T}(\overline{x},\overline{v},\overline{\lambda})_\lambda - \mathcal{T}(x,v,\lambda)_\lambda)\|_{g_0^{-1}} = \|A^\top(\overline{\lambda}_1 - \lambda_1)\|_{g_0^{-1}}$$

$$\leq 0.7\|\overline{v} - v\|_{g_0^{-1}} + 2.4r\|A^\top(\overline{\lambda} - \lambda)\|_{g_0^{-1}} + 1.5\|\overline{x} - x\|_{g_0}\|\overline{v_{1/2}} - A^\top\overline{\lambda}\|_{g_0^{-1}}. \qquad \text{(C.3)}$$

Now we bound $\mathcal{T}(x, v, \lambda)_x$. Note that

$$\mathcal{T}(\overline{x}, \overline{v}, \overline{\lambda})_x - \mathcal{T}(x, v, \lambda)_x = h\beta_1 + h\beta_2$$

where

$$\beta_1 = g_{1/2}^{-1}((\overline{v}_{1/2} - A^\top\overline{\lambda}_1) - (v_{1/2} - A^\top\lambda_1)),$$
$$\beta_2 = (\overline{g}_{1/2}^{-1} - g_{1/2}^{-1})(\overline{v}_{1/2} - A^\top\overline{\lambda}_1).$$

By a proof similar to above, we have

$$\|\beta_1\|_{g_0} \leq 1.2(\|\overline{v}_{1/2} - v_{1/2}\|_{g_0^{-1}} + \|A^\top(\overline{\lambda}_1 - \lambda_1)\|_{g_0^{-1}}),$$
$$\|\beta_2\|_{g_0} \leq 0.6\|\overline{x} - x\|_{g_0}\|\overline{v}_{1/2} - A^\top\overline{\lambda}_1\|_{g_0^{-1}}.$$

and thus

$$\|\mathcal{T}(\overline{x}, \overline{v}, \overline{\lambda})_x - \mathcal{T}(x, v, \lambda)_x\|_{g_0}$$
$$\leq 0.6h\|\overline{v} - v\|_{g_0^{-1}} + 1.2h\|A^\top(\overline{\lambda}_1 - \lambda_1)\|_{g_0^{-1}} + 0.6h\|\overline{x} - x\|_{g_0}\|\overline{v}_{1/2} - A^\top\overline{\lambda}_1\|_{g_0^{-1}}.$$

Finally, we bound $\mathcal{T}(x, v, \lambda)_v$. We split the term

$$\mathcal{T}(\overline{x}, \overline{v}, \overline{\lambda})_v - \mathcal{T}(x, v, \lambda)_v = \frac{h}{2}\gamma_1 + \frac{h}{2}\gamma_2$$

where

$$\gamma_1 = Dg(x_{1/2})[\overline{g}_{1/2}^{-1}(\overline{v}_{1/2} - A^\top\overline{\lambda}_1), \overline{g}_{1/2}^{-1}(\overline{v}_{1/2} - A^\top\overline{\lambda}_1)]$$
$$- Dg(x_{1/2})[g_{1/2}^{-1}(v_{1/2} - A^\top\lambda_1), g_{1/2}^{-1}(v_{1/2} - A^\top\lambda_1)],$$
$$\gamma_2 = Dg(\overline{x}_{1/2})[\overline{g}_{1/2}^{-1}(\overline{v}_{1/2} - A^\top\overline{\lambda}_1), \overline{g}_{1/2}^{-1}(\overline{v}_{1/2} - A^\top\overline{\lambda}_1)]$$
$$- Dg(x_{1/2})[\overline{g}_{1/2}^{-1}(\overline{v}_{1/2} - A^\top\overline{\lambda}_1), \overline{g}_{1/2}^{-1}(\overline{v}_{1/2} - A^\top\overline{\lambda}_1)].$$

Let $\overline{\eta} = \overline{g}_{1/2}^{-1}(\overline{v}_{1/2} - A^\top\overline{\lambda}_1)$ and $\eta = g_{1/2}^{-1}(v_{1/2} - A^\top\lambda_1)$. For $\gamma_1$, we have that

$$\|Dg(x_{1/2})[\overline{\eta}, \overline{\eta}] - Dg(x_{1/2})[\eta, \eta]\|_{g_{1/2}^{-1}}$$
$$\leq 2\|Dg(x_{1/2})[\overline{\eta} - \eta, \overline{\eta}]\|_{g_{1/2}^{-1}} + \|Dg(x_{1/2})[\overline{\eta} - \eta, \overline{\eta} - \eta]\|_{g_{1/2}^{-1}}$$
$$\leq 4\|\overline{\eta} - \eta\|_{g_{1/2}}\|\overline{\eta}\|_{g_{1/2}} + 2\|\overline{\eta} - \eta\|_{g_{1/2}}^2$$

where we use Lemma 11. Using $g_{1/2} \preceq 1.2g_0$ (by (C.2)),

$$\|\gamma_1\|_{g_0^{-1}} \leq 4\|(\overline{v}_{1/2} - A^\top\overline{\lambda}_1) - (v_{1/2} - A^\top\lambda_1)\|_{g_0^{-1}}\|\overline{v}_{1/2} - A^\top\overline{\lambda}_1\|_{g_0^{-1}}$$
$$+ 2\|(\overline{v}_{1/2} - A^\top\overline{\lambda}_1) - (v_{1/2} - A^\top\lambda_1)\|_{g_0^{-1}}^2.$$

For $\gamma_2$, we use Lemma 11 and get

$$\|\gamma_2\|_{g_0^{-1}} \leq \frac{4}{(1 - 0.6\|\overline{x} - x\|_{g_0})^3}\|\overline{v}_{1/2} - A^\top\overline{\lambda}_1\|_{g_0^{-1}}^2\|\overline{x} - x\|_{g_0}$$
$$\leq 6\|\overline{v}_{1/2} - A^\top\overline{\lambda}_1\|_{g_0^{-1}}^2\|\overline{x} - x\|_{g_0}.$$

Combining everything, we have

$$\|\mathcal{T}(\overline{x}, \overline{v}, \overline{\lambda})_v - \mathcal{T}(x, v, \lambda)_v\|_{g_0^{-1}}$$
$$\leq 2h\|(\overline{v}_{1/2} - A^\top\overline{\lambda}_1) - (v_{1/2} - A^\top\lambda_1)\|_{g_0^{-1}}\|\overline{v}_{1/2} - A^\top\overline{\lambda}_1\|_{g_0^{-1}}$$
$$+ h\|(\overline{v}_{1/2} - A^\top\overline{\lambda}_1) - (v_{1/2} - A^\top\lambda_1)\|_{g_0^{-1}}^2$$
$$+ 3h\|\overline{v}_{1/2} - A^\top\overline{\lambda}_1\|_{g_0^{-1}}^2\|\overline{x} - x\|_{g_0}$$

$\square$

Combining the bounds for $\mathcal{T}_\lambda, \mathcal{T}_x, \mathcal{T}_v$, we have

$$\|\mathcal{T}(x,v,\lambda) - \mathcal{T}(\overline{x}, \overline{v}, \overline{\lambda})\|$$
$$\leq 0.7h\|\overline{v} - v\|_{g_0^{-1}} + 2.4rh\|A^\top(\overline{\lambda} - \lambda)\|_{g_0^{-1}} + 1.5h\|\overline{x} - x\|_{g_0}\|\overline{v_{1/2}} - A^\top\overline{\lambda}\|_{g_0^{-1}}$$
$$+ 0.6h\|\overline{v} - v\|_{g_0^{-1}} + 1.2h\|A^\top(\overline{\lambda}_1 - \lambda_1)\|_{g_0^{-1}} + 0.6h\|\overline{x} - x\|_{g_0}\|\overline{v}_{1/2} - A^\top\overline{\lambda}_1\|_{g_0^{-1}}$$
$$+ 2h\|(\overline{v}_{1/2} - A^\top\overline{\lambda}_1) - (v_{1/2} - A^\top\lambda_1)\|_{g_0^{-1}}\|\overline{v}_{1/2} - A^\top\overline{\lambda}_1\|_{g_0^{-1}}$$
$$+ h\|(\overline{v}_{1/2} - A^\top\overline{\lambda}_1) - (v_{1/2} - A^\top\lambda_1)\|_{g_0^{-1}}^2$$
$$+ 3h\|\overline{v}_{1/2} - A^\top\overline{\lambda}_1\|_{g_0^{-1}}^2\|\overline{x} - x\|_{g_0}.$$

To simplify the terms, we note that

$$\|\overline{v}_{1/2} - A^\top\overline{\lambda}_1\|_{g_0^{-1}} \leq \|\overline{v}_{1/2} - A^\top\overline{\lambda}\|_{g_0^{-1}} + \|A^\top(LL^\top)^{-1}A\overline{g}_{1/2}^{-1}\left(\overline{v}_{1/2} - A^\top\overline{\lambda}\right)\|_{g_0^{-1}}$$
$$= \|\overline{v}_{1/2} - A^\top\overline{\lambda}\|_{g_0^{-1}} + \|g_0^{1/2}\overline{g}_{1/2}^{-1}\left(\overline{v}_{1/2} - A^\top\overline{\lambda}\right)\|_P$$
$$\leq \|\overline{v}_{1/2} - A^\top\overline{\lambda}\|_{g_0^{-1}} + \|g_0^{1/2}\overline{g}_{1/2}^{-1}\left(\overline{v}_{1/2} - A^\top\overline{\lambda}\right)\|_2$$
$$\leq 3\|\overline{v}_{1/2} - A^\top\overline{\lambda}\|_{g_0^{-1}}.$$

Using this and simplifying, we have

$$\|\mathcal{T}(x,v,\lambda) - \mathcal{T}(\overline{x}, \overline{v}, \overline{\lambda})\|$$
$$\leq 1.3h\|\overline{v} - v\|_{g_0^{-1}} + 2.4rh\|A^\top(\overline{\lambda} - \lambda)\|_{g_0^{-1}} + 3.3h\|\overline{x} - x\|_{g_0}\|\overline{v_{1/2}} - A^\top\overline{\lambda}\|_{g_0^{-1}}$$
$$+ 1.2h\|A^\top(\overline{\lambda}_1 - \lambda_1)\|_{g_0^{-1}}$$
$$+ 6h(\frac{1}{2}\|\overline{v} - v\|_{g_0^{-1}} + \|A^\top(\overline{\lambda}_1 - \lambda_1)\|_{g_0^{-1}})\|\overline{v}_{1/2} - A^\top\overline{\lambda}\|_{g_0^{-1}}$$
$$+ h\|\overline{v} - v\|_{g_0^{-1}}^2 + 2h\|A^\top(\overline{\lambda}_1 - \lambda_1)\|_{g_0^{-1}}^2$$
$$+ 27h\|\overline{v}_{1/2} - A^\top\overline{\lambda}\|_{g_0^{-1}}^2\|\overline{x} - x\|_{g_0}.$$

Next, we note that

$$\|\overline{v_{1/2}} - A^\top\overline{\lambda}\|_{g_0^{-1}} \leq \frac{1}{2}\|\overline{v} - v^*\|_{g_0^{-1}} + \frac{1}{2}\|v_0 - v^*\|_{g_0^{-1}} + \|v^*\|_{g_0^{-1}}$$
$$+ \frac{1}{2}\|A^\top\overline{\lambda} - A^\top\lambda^*\|_{g_0^{-1}} + \frac{1}{2}\|A^\top\overline{\lambda} - A^\top\lambda^*\|_{g_0^{-1}} + \|A^\top\lambda^*\|_{g_0^{-1}}$$
$$\leq \frac{1}{2}r + \frac{1}{2}r + \|v^*\|_{g_0^{-1}} + \frac{r}{2h} + \frac{r}{2h} + \frac{r}{h} \leq \frac{3r}{h} + \|v^*\|_{g_0^{-1}}$$

Using this, (C.3), $h \leq r$, $r^2 \leq \frac{h}{16}$, $r \leq \|v^*\|_{g_0^{-1}}/4$, we have

$$\|A^\top(\overline{\lambda}_1 - \lambda_1)\|_{g_0^{-1}} \leq \|\overline{v} - v\|_{g_0^{-1}} + 3r\|A^\top(\overline{\lambda} - \lambda)\|_{g_0^{-1}} + (\frac{5r}{h} + 2\|v^*\|_{g_0^{-1}})\|\overline{x} - x\|_{g_0}$$
$$\leq r + \frac{3r^2}{h} + \frac{5r^2}{h} + 2r\|v^*\|_{g_0^{-1}} \leq \frac{8r^2}{h} + 2r\|v^*\|_{g_0^{-1}} \leq 1$$

Hence, we can further simplify it to

$$\|\mathcal{T}(x,v,\lambda) - \mathcal{T}(\overline{x}, \overline{v}, \overline{\lambda})\|$$
$$\leq 2.3h\|\overline{v} - v\|_{g_0^{-1}} + 2.4rh\|A^\top(\overline{\lambda} - \lambda)\|_{g_0^{-1}} + 3.3h\|\overline{x} - x\|_{g_0}\|\overline{v_{1/2}} - A^\top\overline{\lambda}\|_{g_0^{-1}}$$
$$+ 3.2h\|A^\top(\overline{\lambda}_1 - \lambda_1)\|_{g_0^{-1}}$$
$$+ 6h(\frac{1}{2}\|\overline{v} - v\|_{g_0^{-1}} + \|A^\top(\overline{\lambda}_1 - \lambda_1)\|_{g_0^{-1}})\|\overline{v}_{1/2} - A^\top\overline{\lambda}\|_{g_0^{-1}}$$
$$+ 27h\|\overline{v}_{1/2} - A^\top\overline{\lambda}\|_{g_0^{-1}}^2\|\overline{x} - x\|_{g_0}$$
$$\leq (\frac{3r}{h} + \|v^*\|_{g_0^{-1}})(6h\|\overline{v} - v\|_{g_0^{-1}} + 9h\|A^\top(\overline{\lambda}_1 - \lambda_1)\|_{g_0^{-1}} + 31h\|\overline{x} - x\|_{g_0})$$

**Figure D.1:** Proof outline for the mixing rates of CRHMC

$$+ 2.4rh\|A^\top(\overline{\lambda} - \lambda)\|_{g_0^{-1}}$$

where we used $\|\overline{v}_{1/2} - A^\top\overline{\lambda}\|_{g_0^{-1}} \leq \frac{3r}{h} + \|v^*\|_{g_0^{-1}}$ and $r \geq h$. Using the bound on $\|A^\top(\overline{\lambda}_1 - \lambda_1)\|_{g_0^{-1}}$, we have

$$\|\mathcal{T}(x, v, \lambda) - \mathcal{T}(\overline{x}, \overline{v}, \overline{\lambda})\|$$
$$\leq (\frac{3r}{h} + \|v^*\|_{g_0^{-1}})(15h\|\overline{v} - v\|_{g_0^{-1}} + 27rh\|A^\top(\overline{\lambda} - \lambda)\|_{g_0^{-1}} + 9h(\frac{36r}{h} + 2\|v^*\|_{g_0^{-1}})\|\overline{x} - x\|_{g_0})$$
$$+ 2.4rh\|A^\top(\overline{\lambda} - \lambda)\|_{g_0^{-1}}$$
$$\leq (\frac{3r}{h} + \frac{1}{4r})(15h\|\overline{v} - v\|_{g_0^{-1}} + 30rh\|A^\top(\overline{\lambda} - \lambda)\|_{g_0^{-1}} + 9h(\frac{36r}{h} + 2\|v^*\|_{g_0^{-1}})\|\overline{x} - x\|_{g_0})$$
$$\leq (\frac{3r}{h} + \|v^*\|_{g_0^{-1}})(400r + 18h\|v^*\|_{g_0^{-1}})\|(x, v, \lambda) - (\overline{x}, \overline{v}, \overline{\lambda})\|.$$

## D   Condition Number Independence via Self-concordant Barrier

In this section, we analyze the convergence rates of the ideal CRHMC and discretized RHMC in our setting respectively, showing that both are independent of condition numbers. We only show the case when $f$ **is linear**,

$$\pi(x) \propto e^{-f(x)} = e^{-\alpha^\top x}, \text{ for some } \alpha \in \mathbb{R}^n.$$

However, recall that **all logconcave densities** can be reduced to this linear case (see (1.2)). We also focus on when a manifold $\mathcal{M}$ is a polytope in the form of $\{x \in \mathbb{R}^n : A'x \geq b', Ax = 0\}$ for full-rank $A' \in \mathbb{R}^{m \times n}, A \in \mathbb{R}^{p \times n}$ and $b' \in \mathbb{R}^m$, with the Riemannian metric induced by the Hessian of the logarithmic barrier of the polytope. For simplicity, we consider the case when there is no momentum (i.e., $\beta = 0$) in Algorithm 2. In addition, we consider a *lazy* version of Algorithm 2 to avoid a uniqueness issue of the stationary distribution of the Markov chain. The lazy version of the Markov chain, at each step, does nothing with probability $\frac{1}{2}$ (in other words, stays at where it is and does not move). Note that this change for the purpose of proof worsens the mixing rate only by a factor of 2.

In this setting, we show that the mixing rates of the ideal CRHMC and the discretized CRHMC (Algorithm 2) are $O\left(mk^{7/6} \log^2 \frac{\Lambda}{\epsilon}\right)$ and $O\left(mk^3 \log^3 \frac{\Lambda}{\epsilon}\right)$, where $\Lambda$ is a warmness parameter and $k$ is the dimension of the constrained space defined by $\{x \in \mathbb{R}^n : Ax = 0\}$. We remark that our algorithm is actually independent of condition number (i.e., no dependency on $\|\alpha\|_2$ and the geometry of polytope). This is the key reason that our sampler is much more efficient for skewed instances than previous samplers.

We first shed light on how RHMC and CRHMC can be related (see Figure D.1), establishing a correspondence between RHMC (full-dimensional space) and CRHMC (constrained space). This connection enables us to refer to the mixing rates of the ideal RHMC and discretized RHMC proven in [28]. To be precise, we prove in Section D.1 the following theorem on the mixing rate of the ideal CRHMC, which can solve the Hamiltonian equations accurately without any error.

**Theorem 14** (Mixing rate of ideal CRHMC). *Let $\pi_T$ be the distribution obtained after $T$ iterations of the ideal CRHMC on a convex body $\mathcal{M} = \{x \in \mathbb{R}^n : A'x \geq b', Ax = 0\}$. Let $\Lambda = \sup_{S \subseteq \mathcal{M}} \frac{\pi_0(S)}{\pi(S)}$ be the warmness of the initial distribution $\pi_0$. For any $\epsilon > 0$, there exists $T = O\left(mk^{7/6} \log^2 \frac{\Lambda}{\epsilon}\right)$ such that $\|\pi_T - \pi\|_{\mathrm{TV}} \leq \epsilon$, where $k$ is the dimension of the constrained space defined by $\{x \in \mathbb{R}^n : Ax = 0\}$.*

We then prove in Section D.2 the convergence rate of the discretized RHMC (Algorithm 2).

**Theorem 15** (Mixing rate of discretized CRHMC). *Let $\pi_T$ be the distribution obtained after $T$ iterations of Algorithm 2 on a convex body $\mathcal{M} = \{x \in \mathbb{R}^n : A'x \geq b', Ax = 0\}$. Let $\Lambda = \sup_{S \subseteq \mathcal{M}} \frac{\pi_0(S)}{\pi(S)}$ be the warmness of the initial distribution $\pi_0$. For any $\epsilon > 0$, there exists $T = O\left(mk^3 \log^3 \frac{\Lambda}{\epsilon}\right)$ such that $\|\pi_T - \pi\|_{\mathrm{TV}} \leq \epsilon$, where $k$ is the dimension of the constrained space defined by $\{x \in \mathbb{R}^n : Ax = 0\}$.*

We believe there is room for improvement on the $n$-dependence via a more careful analysis.

## D.1 Convergence rate of ideal CRHMC

Lee and Vempala [31] first analyzed Riemannian Hamiltonian Monte Carlo (RHMC) on $n$-dimensional polytopes embedded in $\mathbb{R}^n$, with an invertible metric induced by the Hessian of the logarithmic barrier of the polytopes. They bounded the mixing rate in terms of *smoothness* parameters that depend on the manifold. In particular for uniform sampling, they showed that the convergence rate of RHMC is $O(mn^{2/3})$. Subsequently, [28] extended their analysis to exponential densities and further analyzed the convergence rate of RHMC discretized by the implicit midpoint method, showing that the mixing rates are $O(mn^{7/6})$ and $O(mn^3)$, respectively.

However, our metric $M(x)$ defined for the constrained space could be singular in the underlying space $\mathbb{R}^n$, so we cannot directly refer to any theoretical results from [31, 28]. To address this challenge, we establish a formalism that allows us to reduce the ideal CRHMC to the ideal RHMC, obtaining the mixing rate through this reduction.

Even though our convex body $\mathcal{M} = \{x \in \mathbb{R}^n : A'x \geq b', Ax = 0\}$ of dimension $k$ is embedded in $\mathbb{R}^n$, we can handle it with an invertible metric $\overline{g}$ on $\mathcal{M}$ properly defined as if it is embedded in $\mathbb{R}^k$. To this end, we use $\{u_1, ..., u_k\}$ to denote an orthonormal basis of the constrained space (which is the null space of $A$) and extend it to an orthonormal basis of $\mathbb{R}^n$ denoted by $\{u_1, ..., u_k, ..., u_n\}$. We also define two matrices $U_k \in \mathbb{R}^{n \times k}$ and $U \in \mathbb{R}^{n \times n}$ by

$$U_k = \begin{bmatrix} u_1 & \cdots & u_k \end{bmatrix} \quad \& \quad U = \begin{bmatrix} U_k & u_{k+1} & \cdots & u_n \end{bmatrix}.$$

Using this orthonormal basis $\{u_1, ..., u_k\}$, we can consider a new coordinate system $y = (y_1, ..., y_k) \in \mathbb{R}^k$ on the $k$-dimensional manifold $\mathcal{M}$. Moreover, there exists one-to-one correspondence between $y$ and $x$; for any $x \in \mathcal{M}$ there is a unique $y$ such that $x = U_k y$, and we can recover this $y$ by multiplying $U_k^\top$ (i.e., $y = U_k^\top x$).

Let us define the invertible local metric $\overline{g}$ at $y \in \mathcal{M}$ by

$$\overline{g}(y)(u_i, u_j) \stackrel{\text{def}}{=} g(x)(u_i, u_j) \quad \text{for } i, j \leq k.$$

With abuse of notations, we also use $\overline{g}(y)$ to denote the $k \times k$ matrix with its $(i, j)$-entry being $\overline{g}(y)(u_i, u_j)$. We first establish relationships between $\overline{g}(y)$ (and its inverse $\overline{g}^{-1}$) and $M(x)$ (and its pseudoinverse $W \stackrel{\text{def}}{=} M(x)^\dagger$). We recall that for the orthogonal projection $Q$ to the null space of $A$

$$M(x) = Q^\top g(x)Q, \quad W(x) = g(x)^{-\frac{1}{2}}(I - P(x))g(x)^{-\frac{1}{2}}.$$

**Lemma 16.** *We have $\overline{g}(y) = U_k^\top M(x)U_k = U_k^\top g(x)U_k$ and $\overline{g}(y)^{-1} = U_k^\top W(x)U_k$.*

*Proof.* It is immediate from the definition of $\overline{g}$ that $\overline{g}(y) = U_k^\top g(x)U_k$. Since the quadratic forms of $M(x)$ and $g(x)$ agree on the constrained space, we also have $\overline{g}(y) = U_k^\top M(x)U_k$.

For $\overline{g}^{-1}$, we define two matrices $P_k \in \mathbb{R}^{n \times k}$ and $P_r \in \mathbb{R}^{n \times (n-k)}$ by

$$P_k = \begin{bmatrix} I_k \\ 0_{(n-k) \times k} \end{bmatrix}, \quad P_r = \begin{bmatrix} 0_{k \times (n-k)} \\ I_{n-k} \end{bmatrix}$$

where $0_{(n-k)\times k}$ is the zero matrix of size $(n-k) \times k$, $I_k$ is the identity matrix of size $k \times k$ and so on. Due to $U_k = UP_k$, the upper-left $k \times k$ submatrix of $g'(x) := U^\top g(x)U$ is exactly $\overline{g}(y)$. Let us represent the inverse of $g'$ in the form of block matrix:

$$g'(x)^{-1} = \begin{bmatrix} B_1 & B_2 \\ B_2^\top & B_3 \end{bmatrix},$$

for $B_1 \in \mathbb{R}^{k \times k}$, $B_2 \in \mathbb{R}^{k \times (n-k)}$ and $B_3 \in \mathbb{R}^{(n-k) \times (n-k)}$. Using the formula of the inverse of block matrices (see App. E.3),

$$\overline{g}(y)^{-1} = B_1 - B_2 B_3^{-1} B_2^\top.$$

It is straightforward to check

$$\begin{aligned}
B_1 &= P_k^\top g'(x)^{-1} P_k = P_k^\top U^\top g(x)^{-1} U P_k \\
&= U_k^\top g(x)^{-1} U_k, \\
B_2 &= P_k^\top g'(x)^{-1} P_r = U_k^\top g(x)^{-1} U_r, \\
B_3 &= P_r^\top g'(x)^{-1} P_r = U_r^\top g(x)^{-1} U_r,
\end{aligned}$$

for $U_r = \begin{bmatrix} u_{k+1} & \cdots & u_n \end{bmatrix} \in \mathbb{R}^{n \times (n-k)}$. Therefore,

$$\begin{aligned}
\overline{g}(y)^{-1} &= U_k^\top g(x)^{-1} U_k - U_k^\top g(x)^{-1} U_r (U_r^\top g(x)^{-1} U_r)^{-1} U_r^\top g(x)^{-1} U_k \\
&= U_k^\top \left( g(x)^{-1} - g(x)^{-1} U_r (U_r^\top g(x)^{-1} U_r)^{-1} U_r^\top g(x)^{-1} \right) U_k.
\end{aligned}$$

Since $g(x)^{-\frac{1}{2}} U_r (U_r^\top g(x)^{-1} U_r)^{-1} U_r^\top g(x)^{-\frac{1}{2}}$ is the orthogonal projection to the row space of $U_r^\top g(x)^{-\frac{1}{2}}$ and this row space is the same with the row space of $Ag(x)^{-\frac{1}{2}}$, the uniqueness of orthogonal projection matrices implies

$$g(x)^{-\frac{1}{2}} A^\top (Ag(x)^{-1}A^\top)^{-1} Ag(x)^{-\frac{1}{2}} = g(x)^{-\frac{1}{2}} U_r (U_r^\top g(x)^{-1} U_r)^{-1} U_r^\top g(x)^{-\frac{1}{2}}.$$

Therefore,

$$\begin{aligned}
\overline{g}(y)^{-1} &= U_k^\top \left( g(x)^{-1} - g(x)^{-1} A^\top (Ag(x)^{-1}A^\top)^{-1} Ag(x)^{-1} \right) U_k \\
&= U_k^\top \left( g(x)^{-\frac{1}{2}} \left( I - g(x)^{-\frac{1}{2}} A^\top (Ag(x)^{-1}A^\top)^{-1} Ag(x)^{-\frac{1}{2}} \right) g(x)^{-\frac{1}{2}} \right) U_k \\
&= U_k^\top \left( g(x)^{-\frac{1}{2}} (I - P(x)) g(x)^{-\frac{1}{2}} \right) U_k \\
&= U_k^\top W(x) U_k.
\end{aligned}$$

$\square$

We can now view the ideal CRHMC with the metric $M(x)$ as the ideal RHMC with the metric $\overline{g}$ on the $k$-dimensional manifold. Note that we have to ensure that the local metric $\overline{g}$ is also induced by the Hessian of a logarithmic barrier, in order to refer to results from it.

**Lemma 17.** *Let $\overline{A'} = A'U_k$ and $\psi(y)$ be the logarithmic barrier of the $k$-dimensional polytope defined by $\{y \in \mathbb{R}^k : \overline{A'}y \le b'\}$. Then $\nabla_y^2 \psi(y) = \overline{g}(y)$.*

*Proof.* Observe that $\{y \in \mathbb{R}^k : \overline{A'}y \ge b'\}$ is the new representation of $\mathcal{M} = \{x \in \mathbb{R}^n : A'x \ge b', Ax = 0\}$ in the $y$-coordinate system. Due to $\overline{A'}y = Ax$, we have $S_x = \mathrm{Diag}(A'x - b') = \mathrm{Diag}(\overline{A'}y - b) = S_y$. For the logarithmic barrier $\phi(x)$ of $\{x \in \mathbb{R}^n : A'x \ge b\}$, direct computation results in

$$\begin{aligned}
\nabla_y^2 \psi(y) &= \overline{A'}^\top S_y^{-2} \overline{A'} = U_k^\top A'^\top S_x^{-2} A' U_k = U_k^\top \nabla_x^2 \phi(x) U_k \\
&= U_k^\top g(x) U_k = \overline{g}(y),
\end{aligned}$$

where we used $\nabla_x^2 \phi(x) = A'^\top S_x^{-2} A'$ in the third equality and Lemma 16 in the last equality. $\square$

Most importantly, we prove that this ideal RHMC on the $k$-dimensional manifold with the metric $\overline{g}(y)$ is equivalent to the ideal CRHMC with the metric $M(x)$.

**Lemma 18.** *The dynamics $(x, v)$ and $(y, u)$ of the ideal CRHMC in $\mathbb{R}^n$ and the ideal RHMC in $\mathbb{R}^k$ are equivalent in a sense that the Hamiltonian equations for $(x, v)$ can be obtained by lifting up the Hamiltonian equations for $(y, u)$ from $\mathbb{R}^k$ to $\mathbb{R}^n$ via multiplying $U_k$. That is, when we lift up the dynamics $(y, u)$ in $\mathbb{R}^k$ to the dynamics $(\overline{x}, \overline{v})$ in $\mathbb{R}^n$ defined by $\overline{x} = U_k y$ and $\overline{v} = U_k u$, it follows that*

$$\frac{dx}{dt} = \frac{d\overline{x}}{dt}, \frac{dv}{dt} = \frac{d\overline{v}}{dt} \text{ and } v, \overline{v} \sim \mathcal{N}(0, M(x)).$$

*Proof.* We first recall from the proof of Lemma 4 that the Hamiltonian equations of $(x, v)$ are

$$\frac{dx}{dt} = W(x)v,$$

$$\begin{aligned}
\frac{dv}{dt} &= -\nabla_x f(x) - \frac{1}{2}\text{Tr}\left[W(x)Dg(x)\right] + \frac{1}{2}Dg(x)\left[\frac{dx}{dt}, \frac{dx}{dt}\right] - A^\top \lambda(x, v) \\
&= -\nabla_x f(x) - \frac{1}{2}\text{Tr}\left[W(x)Dg(x)\right] + \frac{1}{2}Dg(x)\left[\frac{dx}{dt}, \frac{dx}{dt}\right] \\
&\quad - A^\top \left(AA^\top\right)^{-1} A\left(-\nabla_x f(x) - \frac{1}{2}\text{Tr}\left[W(x)Dg(x)\right] + \frac{1}{2}Dg(x)\left[\frac{dx}{dt}, \frac{dx}{dt}\right]\right) \\
&= (I - A^\top\left(AA^\top\right)^{-1} A)\left(-\nabla_x f(x) - \frac{1}{2}\text{Tr}\left[W(x)Dg(x)\right] + \frac{1}{2}Dg(x)\left[\frac{dx}{dt}, \frac{dx}{dt}\right]\right) \\
&= U_k U_k^\top\left(-\nabla_x f(x) - \frac{1}{2}\text{Tr}\left[W(x)Dg(x)\right] + \frac{1}{2}Dg(x)\left[\frac{dx}{dt}, \frac{dx}{dt}\right]\right),
\end{aligned}$$

where the last equality follows from that $U_k U_k^\top$ is the orthogonal projection to the null space of $A$.

From Lemma 7 of [31], the Hamiltonian equations of $(y, u)$ are

$$\frac{dy}{dt} = \overline{g}(y)^{-1}u,$$

$$\frac{du}{dt} = -\nabla_y f(U_k y) - \frac{1}{2}\text{Tr}\left[\overline{g}(y)^{-1}D\overline{g}(y)\right] + \frac{1}{2}D\overline{g}(y)\left[\frac{dy}{dt}, \frac{dy}{dt}\right].$$

From the definitions of $(\overline{x}, \overline{v})$, we have

$$\begin{aligned}
\frac{d\overline{x}}{dt} &= U_k \overline{g}(y)^{-1}u = U_k U_k^\top W(\overline{x})U_k u = U_k U_k^\top W(\overline{x})\overline{v} \\
&\underset{(*)}{=} W(\overline{x})\overline{v}, \\
\frac{d\overline{v}}{dt} &= U_k\left(-\nabla_y f(U_k y) - \frac{1}{2}\text{Tr}\left[\overline{g}(y)^{-1}D\overline{g}(y)\right] + \frac{1}{2}D\overline{g}(y)\left[\frac{dy}{dt}, \frac{dy}{dt}\right]\right),
\end{aligned}$$

where $(*)$ follows from that $W(x)\overline{v}$ is already in the constrained space (i.e., the null space of $A$). Let us examine each term in $d\overline{v}/dt$ separately.

$$\begin{aligned}
\nabla_y f(U_k y) &= U_k^\top \nabla_{\overline{x}} f(\overline{x}), \\
\text{Tr}\left[\overline{g}(y)^{-1}D_y \overline{g}(y)\right] &= \text{Tr}\left[U_k^\top W(\overline{x})U_k \cdot D_y\left(U_k^\top g(U_k y)U_k\right)\right] \\
&= \text{Tr}\left[U_k U_k^\top W(\overline{x})U_k U_k^\top (U_k^\top D_{\overline{x}} g(\overline{x}))\right] \\
&= \text{Tr}\left[W(\overline{x})U_k U_k^\top (U_k^\top D_{\overline{x}} g(\overline{x}))\right] \\
&= \text{Tr}\left[(U_k U_k^\top W(\overline{x}))^\top (U_k^\top D_{\overline{x}} g(\overline{x}))\right] \\
&= \text{Tr}\left[W(\overline{x})(U_k^\top D_{\overline{x}} g(\overline{x}))\right] \\
&= U_k^\top \text{Tr}\left[W(\overline{x})D_{\overline{x}} g(\overline{x})\right], \\
D_y \overline{g}(y)\left[\frac{dy}{dt}, \frac{dy}{dt}\right] &= U_k^\top (U_k^\top D_{\overline{x}} g(\overline{x}))U_k\left[\frac{dy}{dt}, \frac{dy}{dt}\right] \\
&= \left(\overline{v}^\top W(\overline{x})U_k\right) U_k^\top (U_k^\top D_{\overline{x}} g(\overline{x}))U_k\left(U_k^\top W(\overline{x})\overline{v}\right)
\end{aligned}$$

$$= \overline{v}^{\top} W(\overline{x})(U_k^{\top} D_{\overline{x}} g(\overline{x})) W(\overline{x}) \overline{v}$$

$$= U_k^{\top} D_{\overline{x}} g(\overline{x}) \left[ \frac{d\overline{x}}{dt}, \frac{d\overline{x}}{dt} \right].$$

Putting all these together, the Hamiltonian equations of $(\overline{x}, \overline{v})$ can be written as

$$\frac{d\overline{x}}{dt} = W(\overline{x})\overline{v},$$

$$\frac{d\overline{v}}{dt} = U_k U_k^{\top} \left( -\nabla f(\overline{x}) - \frac{1}{2} \mathrm{Tr}\left[ W(\overline{x}) Dg(\overline{x}) \right] + \frac{1}{2} Dg(\overline{x}) \left[ \frac{d\overline{x}}{dt}, \frac{d\overline{x}}{dt} \right] \right).$$

Therefore, the Hamiltonian equations of $(x, v)$ and $(\overline{x}, \overline{v})$ are exactly the same. In addition, $\overline{v} = U_k u$ leads to $\overline{v} \sim \mathcal{N}(0, U_k \overline{g}(y) U_k^{\top}) = \mathcal{N}(0, M(x))$. $\qquad\square$

Using these three lemmas, we conclude that the dynamics of the ideal CRHMC on the constrained space is equivalent to that of the ideal RHMC on the corresponding $k$-dimensional polytope. Therefore, Theorem 14 immediately follows from Corollary 3 in [28].

**Corollary.** *Let $\pi$ be a target distribution on a polytope with $m$ constraints in $\mathbb{R}^n$ such that $\frac{d\pi}{dx} \sim e^{-\alpha^{\top} x}$ for $\alpha \in \mathbb{R}^n$. Let $\mathcal{M}$ be the Hessian manifold of the polytope induced by the logarithmic barrier of the polytope. Let $\Lambda = \sup_{S \subset \mathcal{M}} \frac{\pi_0(S)}{\pi(S)}$ be the warmness of the initial distribution $\pi_0$. Let $\pi_T$ be the distribution obtained after $T$ iterations of the ideal RHMC on $\mathcal{M}$. For any $\varepsilon > 0$ and step size $h = O\left( \frac{1}{n^{7/12} \log^{1/2} \frac{\Lambda}{\varepsilon}} \right)$, there exists $T = O\left( mn^{7/6} \log^2 \frac{\Lambda}{\varepsilon} \right)$ such that $\|\pi_T - \pi\|_{TV} \leq \varepsilon$.*

### D.2 Convergence rate of discretized CRHMC

We attempt to demonstrate a similar reduction of the discretized CRHMC. However, it is trickier than that of the ideal RHMC, since Algorithm 2 uses the simplified Hamiltonian, which omits the Lagrangian term $c(x)^{\top} \lambda(x, v)$, in place of the full Hamiltonian.

We look into the reduction in two steps. First of all, we show in Section D.2.1 that the dynamics of $x$ is the same under the discretized CRHMC via IMM with the simplified Hamiltonian, and the discretized CRHMC via IMM with the full Hamiltonian, and that the acceptance probabilities are the same as well. Next in Section D.2.2, we show a correspondence between the discretized CRHMC via IMM and the discretized RHMC via IMM, just as we did for the ideal case in Section D.1.

#### D.2.1 Simplified Hamiltonian and full Hamiltonian in constrained space

We recall that the simplified Hamiltonian $\overline{H}(x, v)$ is the sum of two parts defined by

$$\overline{H}_1(x, v) = f(x) + \frac{1}{2} \log \mathrm{pdet} M(x)$$

$$= f(x) + \frac{1}{2} \left( \log \det g(x) + \log \det Ag(x)^{-1} A^{\top} - \log \det AA^{\top} \right),$$

$$\overline{H}_2(x, v) = \frac{1}{2} v^{\top} W(x) v.$$

The full Hamiltonian $H(x, v)$ with the Lagrangian term $c(x)^{\top} \lambda(x, v)$ can be also written as the sum of two parts defined by

$$H_1(x, v) = f(x) + \frac{1}{2} \left( \log \det g(x) + \log \det Ag(x)^{-1} A^{\top} - \log \det AA^{\top} \right)$$

$$- x^{\top} A^{\top} (AA^{\top})^{-1} A \left( \nabla f(x) + \frac{1}{2} \mathrm{Tr}\left[ W(x) Dg(x) \right] \right),$$

$$H_2(x, v) = \frac{1}{2} v^{\top} W(x) v + \frac{1}{2} x^{\top} A^{\top} (AA^{\top})^{-1} A Dg(x) \left[ \frac{dx}{dt}, \frac{dx}{dt} \right],$$

and IMM with this Hamiltonian is implemented as in (C.1). We note from the proof of Lemma 18 that

$$\frac{\partial H_1}{\partial x}(x, v) = U_k U_k^{\top} \frac{\partial \overline{H}_1}{\partial x}(x, v),$$

$$\frac{\partial H_2}{\partial x}(x, v) = U_k U_k^\top \frac{\partial \overline{H}_2}{\partial x}(x, v),$$

$$\frac{\partial H_2}{\partial v}(x, v) = U_k U_k^\top \frac{\partial \overline{H}_2}{\partial v}(x, v).$$

**Lemma 19.** *For step size $h$, let $(\overline{x}, \overline{v})$ and $(x, v)$ be the outputs of IMM with the simplified Hamiltonian and with the full Hamiltonian starting from $(x_0, v_0)$, respectively. Then $\overline{x} = x$, and $\frac{e^{-\overline{H}(\overline{x}, \overline{v})}}{e^{-\overline{H}(x_0, v_0)}} = \frac{e^{-H(x, v)}}{e^{-H(x_0, v_0)}}.$*

*Proof.* We use $x_{\frac{1}{3}}(= x_0), x_{\frac{2}{3}}(= x)$ and $v_{\frac{1}{3}}, v_{\frac{2}{3}}, v$ to denote the points obtained during one step of IMM with the full Hamiltonian. We similarly define $\overline{x}_i$ and $\overline{v}_i$ for $i = \frac{1}{3}, \frac{2}{3}$. As $(x_0, v_0)$ is a starting point, $U_k U_k^\top x_0 = x_0$ and $U_k U_k^\top v_0 = v_0$. Due to $\mathrm{Null}(W(x)) = \mathrm{row}(A)$, we have that $W(z) U_k U_k^\top w = W(z) w$. By comparing the first step of IMM for each Hamiltonian,

$$U_k U_k^\top \overline{v}_{\frac{1}{3}} = v_0 - \frac{h}{2} U_k U_k^\top \frac{\partial \overline{H}_1}{\partial x}(x_0, v_0) = v_0 - \frac{h}{2} \frac{\partial H_1}{\partial x}(x_0, v_0) = v_{\frac{1}{3}},$$

and thus $U_k U_k^\top \overline{v}_{\frac{1}{3}} = v_{\frac{1}{3}}$.

From the second step of IMM, $\overline{x}_{\frac{2}{3}}$ is already in the null space of $A$. For $x_{\frac{1}{3}} = \overline{x}_{\frac{1}{3}} = x_0, \overline{x}_{\mathrm{mid}} = (\overline{x}_{\frac{1}{3}} + \overline{x}_{\frac{2}{3}})/2$ and $\overline{v}_{\mathrm{mid}} = (\overline{v}_{\frac{1}{3}} + \overline{v}_{\frac{2}{3}})/2$, the second step of IMM with the simplified Hamiltonian is

$$\overline{x}_{\frac{2}{3}} = x_{\frac{1}{3}} + h U_k U_k^\top \frac{\partial \overline{H}_2}{\partial v}(\overline{x}_{\mathrm{mid}}, \overline{v}_{\mathrm{mid}}) = x_{\frac{1}{3}} + h U_k U_k^\top W(\overline{x}_{\mathrm{mid}}) \overline{v}_{\mathrm{mid}}$$

$$= x_{\frac{1}{3}} + h U_k U_k^\top W(\overline{x}_{\mathrm{mid}}) U_k U_k^\top \overline{v}_{\mathrm{mid}} = x_{\frac{1}{3}} + h U_k U_k^\top \frac{\partial \overline{H}_2}{\partial v}(\overline{x}_{\mathrm{mid}}, U_k U_k^\top \overline{v}_{\mathrm{mid}})$$

$$= x_{\frac{1}{3}} + h \frac{\partial H_2}{\partial v}(\overline{x}_{\mathrm{mid}}, U_k U_k^\top \overline{v}_{\mathrm{mid}}),$$

$$U_k U_k^\top \overline{v}_{\frac{2}{3}} = v_{\frac{1}{3}} + h U_k U_k^\top \frac{\partial \overline{H}_2}{\partial x}(\overline{x}_{\mathrm{mid}}, \overline{v}_{\mathrm{mid}})$$

$$= v_{\frac{1}{3}} + h U_k U_k^\top \left( -\frac{1}{2} Dg(\overline{x}_{\mathrm{mid}}) \left[ W(\overline{x}_{\mathrm{mid}}) \overline{v}_{\mathrm{mid}}, W(\overline{x}_{\mathrm{mid}}) \overline{v}_{\mathrm{mid}} \right] \right)$$

$$= v_{\frac{1}{3}} + h U_k U_k^\top \left( -\frac{1}{2} Dg(\overline{x}_{\mathrm{mid}}) \left[ W(\overline{x}_{\mathrm{mid}}) U_k U_k^\top \overline{v}_{\mathrm{mid}}, W(\overline{x}_{\mathrm{mid}}) U_k U_k^\top \overline{v}_{\mathrm{mid}} \right] \right)$$

$$= v_{\frac{1}{3}} + h U_k U_k^\top \frac{\partial \overline{H}_2}{\partial x}(\overline{x}_{\mathrm{mid}}, U_k U_k^\top \overline{v}_{\mathrm{mid}})$$

$$= v_{\frac{1}{3}} + h \frac{\partial H_2}{\partial x}(\overline{x}_{\mathrm{mid}}, U_k U_k^\top \overline{v}_{\mathrm{mid}}).$$

Note that $U_k U_k^\top \overline{v}_{\mathrm{mid}} = (U_k U_k^\top \overline{v}_{\frac{2}{3}} + v_{\frac{1}{3}})/2$ and $\overline{x}_{\mathrm{mid}} = (\overline{x}_{\frac{2}{3}} + x_{\frac{1}{3}})/2$. Since the solution of this second step is characterized as a unique fixed-point, it follows that $(\overline{x}_{\frac{2}{3}}, U_k U_k^\top \overline{v}_{\frac{2}{3}}) = (x_{\frac{2}{3}}, v_{\frac{2}{3}})$ and so $\overline{x} = x$. In the same way we analyzed $\overline{v}_{\frac{2}{3}}$, we can obtain that $U_k U_k^\top \overline{v} = v$.

We now compare the acceptance probabilities. We clearly have $\overline{H}(x_0, v_0) = H(x_0, v_0)$ due to $c(x_0) = 0$ and have $\overline{H}_1(\overline{x}, \overline{v}) = H_1(x, v)$ due to $\overline{x} = x$. For $\overline{H}_2$,

$$\overline{v}^\top W(\overline{x}) \overline{v} = \overline{v}^\top U_k U_k^\top W(x) U_k U_k^\top \overline{v} = v^\top W(x) v,$$

and so $\overline{H}_2(\overline{x}, \overline{v}) = H_2(x, v)$. $\qquad \square$

### D.2.2 CRHMC and RHMC discretized by IMM

In this section, we show that there is a correspondence between the dynamics of CRHMC discretized by IMM and that of RHMC discretized by IMM.

**Lemma 20.** *The discretized CRHMC via IMM in $\mathbb{R}^n$ and the discretized RHMC via IMM in $\mathbb{R}^k$ are equivalent. That is, the output $(x_1, v_1)$ given by the discretized CRHMC starting from $(x, v)$ is the*

*same with $(U_k y_1, U_k u_1)$, where $(y_1, u_1)$ is the output of the discretized RHMC starting from $(y, u)$ satisfying $(x, v) = (U_k y, U_k u)$. Moreover, the acceptance probabilities are the same due to*

$$\frac{e^{-H^c(x_1, v_1)}}{e^{-H^c(x, v)}} = \frac{e^{-H^r(y_1, u_1)}}{e^{-H^r(y, u)}},$$

*where $H^c(x, v)$ and $H^r(y, u)$ are the Hamiltonians of CRHMC and RHMC respectively.*

*Proof.* We first recall that $H^c(x, v)$ can be rewritten as the sum of two parts defined by

$$H_1^c(x, v) = f(x) + \frac{1}{2} \log \mathrm{pdet} M(x)$$
$$- x^\top A^\top (AA^\top)^{-1} A \left( \nabla f(x) + \frac{1}{2} \mathrm{Tr}\left[ W(x) Dg(x) \right] \right),$$
$$H_2^c(x, v) = \frac{1}{2} v^\top W(x) v + \frac{1}{2} x^\top A^\top (AA^\top)^{-1} A Dg(x) \left[ \frac{dx}{dt}, \frac{dx}{dt} \right].$$

Similarly for $H^r$, we can represent it by the sum of two parts defined by

$$H_1^r(y, u) = f(U_k y) + \frac{1}{2} \log \det \overline{g}(y),$$
$$H_2^r(y, u) = \frac{1}{2} u^\top \overline{g}(y)^{-1} u.$$

For the first claim, we need to show that each step of IMM for RHMC and CRHMC is equivalent, thus it suffices to check that for any $(y, u) \in \mathbb{R}^k \times \mathbb{R}^k$

$$\frac{\partial H_1^c(U_k y, U_k u)}{\partial x} = U_k \frac{\partial H_1^r(y, u)}{\partial y},$$
$$\frac{\partial H_2^c(U_k y, U_k u)}{\partial x} = U_k \frac{\partial H_2^r(y, u)}{\partial y},$$
$$\frac{\partial H_2^c(U_k y, U_k u)}{\partial v} = U_k \frac{\partial H_2^r(y, u)}{\partial u}.$$

These computations were already checked in the proof of Lemma 18.

For the second claim, we note that the Lagrangian term vanishes due to $c(x) = c(U_k y) = 0$. Then the second claim follows from

$$\log \det \overline{g}(y') = \log \det U_k^\top M(U_k y') U_k \quad \text{(Lemma 16)}$$
$$= \log \mathrm{pdet} M(U_k y'),$$
$$u'^\top \overline{g}(y')^{-1} u' = u'^\top U_k^\top W(U_k y') U_k u'. \quad \text{(Lemma 16)}$$

$\square$

The previous two lemmas imply that the dynamics of the discretized CRHMC via IMM on the constrained space is equivalent to that of the discretized RHMC via IMM on the corresponding k-dimensional polytope. Therefore, Theorem 15 follows from Corollary 4 in [28].

**Corollary.** *Let $\pi$ be a target distribution on a polytope with $m$ constraints in $\mathbb{R}^n$ such that $\frac{d\pi}{dx} \sim e^{-\alpha^\top x}$ for $\alpha \in \mathbb{R}^n$. Let $\mathcal{M}$ be the Hessian manifold of the polytope induced by the logarithmic barrier of the polytope. Let $\Lambda = \sup_{S \subset \mathcal{M}} \frac{\pi_0(S)}{\pi(S)}$ be the warmness of the initial distribution $\pi_0$. Let $\pi_T$ be the distribution obtained after $T$ iterations of RHMC discretized by IMM on $\mathcal{M}$. For any $\varepsilon > 0$ and step size $h = O\left( \frac{1}{n^{3/2} \log \frac{\Lambda}{\varepsilon}} \right)$, there exists $T = O\left( mn^3 \log^3 \frac{\Lambda}{\varepsilon} \right)$ such that $\| \pi_T - \pi \|_{TV} \le \varepsilon$.*

# E  Missing Notations and Definitions

## E.1  Notations

- We use $\mathcal{N}(\mu, \Sigma)$ to denote Gaussian distribution with mean $\mu$ and covariance $\Sigma$.

- We use $\text{Null}(A)$ and $\text{Range}(A)$ to denote the null space and image space of a matrix or linear operator $A$.
- We use $\nabla^2 f \in \mathbb{R}^{n \times n}$ to denote the Hessian of a function $f : \mathbb{R}^n \to \mathbb{R}$.
- We use $\|\cdot\|$ to denote $\ell_2$-norm unless specified otherwise, and define $\|x\|_A := \sqrt{x^\top A x}$ for a vector $x \in \mathbb{R}^n$ and a matrix $A \in \mathbb{R}^{n \times n}$.
- We use $\partial K$ to denote the boundary of the set $K$.
- For a matrix $g(x)$ with $x \in \mathbb{R}^n$, we use $Dg(x)$ to denote the derivative of $g(x)$ with respect to $x$. This can be thought of as the $n \times n \times n$ tensor such that $(Dg(x))(i,j,k) = \frac{\partial (g(x))_{ij}}{\partial x_k}$. In other words, $(Dg(x))(\cdot,\cdot,k)$ is the matrix, each of entries is the derivative of $g(x)$ with respect to $x_k$. In addition, for a vector $v \in \mathbb{R}^n$, $Dg(x)[v,v]$ is a vector in $\mathbb{R}^n$ such that $(Dg(x)[v,v])_i = v^\top Dg(x)(\cdot,\cdot,i)v$.
- For a matrix $A$ of size $n \times n$, we use $A \cdot Dg(x)$ to denote a $n \times n \times n$ tensor such that $(A \cdot Dg(x))(\cdot,\cdot,i) = A \cdot (Dg(x))(\cdot,\cdot,i)$. We use $\text{Tr}(A \cdot Dg(x))$ to denote a vector in $\mathbb{R}^n$ such that $(\text{Tr}(A \cdot Dg(x)))_i = \text{Tr}((A \cdot Dg(x))(\cdot,\cdot,i))$.

## E.2 Definitions

**Convex body.** A convex body is a compact and convex set.

**Isotropy.** A random variable $X$ is said to be in isotropic position if $\mathbb{E}X = 0$ and $\mathbb{E}XX^\top = I$.

**Pseudo-inverse.** For a matrix $A \in \mathbb{R}^{m \times n}$, it is well known that there always exists the unique pseudo-inverse matrix $A^\dagger$ that satisfies the following conditions:

1. $A^\dagger A A^\dagger = A^\dagger$.
2. $AA^\dagger A = A$.
3. $AA^\dagger$ and $A^\dagger A$ are symmetric.

It is also well known that $\text{Null}(A^\dagger) = \text{Null}(A^\top)$ and $\text{Range}(A^\dagger) = \text{Range}(A^\top)$.

**Pseudo-determinant.** For a square matrix $A$, its pseudo-determinant $\text{pdet}(A)$ is defined as the product of non-zero eigenvalues of $A$.

**Leverage score.** For a matrix $A \in \mathbb{R}^{m \times n}$, the leverage score of the $i^{th}$ row is $(A(A^\top A)^\dagger A^\top)_{ii}$ for $i \in [m]$. When $A$ is full-rank, it is simply $(A(A^\top A)^{-1} A^\top)_{ii}$.

**Log-barrier & Dikin ellipsoid.** For a polytope $P = \{x \in \mathbb{R}^n : Ax \leq b\}$ where $A \in \mathbb{R}^{m \times n}$ and $b \in \mathbb{R}^m$, let us denote the $i^{th}$ row of $A$ by $a_i$ and the $i^{th}$ row of $b$ by $b_i$. The log-barrier of $P$ is defined by

$$\phi(x) = -\sum_{i=1}^m \log(b_i - a_i^\top x).$$

For $x \in P$, the Dikin ellipsoid at $x$ is defined by $D(x) := \{y \in \mathbb{R}^n : (y-x)^\top \nabla^2 \phi(x)(y-x) \leq 1\}$. The Dikin ellipsoid is always contained in $P$.

**Analytic center.** The analytic center $x_{ac}$ of the polytope $P$ is the point minimizing the log-barrier (i.e., $x_{ac} = \arg\min \phi(x)$).

**Self-concordant function.** A function $f : \mathbb{R}^n \to \mathbb{R}$ is self-concordant if it satisfies $|D^3 f(x)[h,h,h]| \leq 2 (D^2 f(x)[h,h])^{3/2}$ for all $h \in \mathbb{R}^n$ and $x \in \mathbb{R}^n$.

**Highly self-concordant function.** A barrier $\phi$ is called *highly self-concordant* if it satisfies for all $h \in \mathbb{R}^d$ and $x \in \mathbb{R}^d$

$$|D^3 \phi(x)[h,h,h]| \leq 2 (D^2 \phi(x)[h,h])^{3/2} \quad \text{and} \quad |D^4 \phi(x)[h,h,h,h]| \leq 6 (D^2 \phi(x)[h,h])^2.$$

**Total variation.** For two probability distributions $P$ and $Q$ on support $K$, the total variation distance of $P$ and $Q$ is

$$\|P - Q\|_{\text{TV}} \overset{\text{def}}{=} \sup_{A \subseteq K} (P(A) - Q(A)).$$

### E.3 Details

**Inverse and Determinant of block matrix.** For a square matrix $M = \begin{bmatrix} A & B \\ C & D \end{bmatrix}$ with blocks $A, B, C, D$ of same size, if $D$ and $A - BD^{-1}C$ are invertible, then its inverse and determinant can be computed by

$$M^{-1} = \begin{bmatrix} (A - BD^{-1}C)^{-1} & -(A - BD^{-1}C)^{-1}BD^{-1} \\ -D^{-1}C(A - BD^{-1}C)^{-1} & D^{-1} + D^{-1}C(A - BD^{-1}C)^{-1}BD^{-1} \end{bmatrix},$$

$$\det(M) = \det(D)\det(A - BD^{-1}C).$$

**Orthogonal projection.** Let $S = \{x \in \mathbb{R}^n : Ax = b\}$ for $A \in \mathbb{R}^{m \times n}$ and $b \in \mathbb{R}^m$, and $x_0$ be a point in $S$. Thus $S - x_0$ is the null space of $A$, due to $A(x - x_0) = 0$. The orthogonal projection $P$ to this null space is

$$P = I - A^\top (AA^\top)^{-1}A.$$

Note that the range of $P$ always lies in the null space because

$$A(Pv) = A(I - A^\top(AA^\top)^{-1}A)v = Av - AA^\top(AA^\top)^{-1}Av = Av - Av = 0.$$

$I - P$ is also an orthogonal projection matrix, and eigenvalues of orthogonal projection matrices are either 0 or 1.

**Matrix calculus.** Let $U(x)$ be a $n \times n$ matrix with a parameter $x \in \mathbb{R}^n$.

$$\frac{\partial U^{-1}(x)}{\partial x_i} = -U(x)\frac{\partial U(x)}{\partial x_i}U(x).$$

Hence using the notation $Dg$, we can write in a more compact way as

$$DU^{-1}(x) = -U(x)DU(x)U(x).$$

For $\log \det$,

$$\frac{\partial \log \det U(x)}{\partial x_i} = \text{Tr}\left(U^{-1}(x)\frac{\partial U(x)}{\partial x_i}\right).$$

In other words,

$$D(\log \det U(x)) = \text{Tr}(U^{-1}(x)DU(x)).$$

**Cholesky decomposition.** For a symmetric positive definite matrix $A$, there exists a lower triangular matrix $L$ such that $LL^\top = A$.

**Newton's method.** For $f$ convex and twice differentiable in $\mathbb{R}^n$, consider an unconstrained convex optimization $\min_x f(x)$. Given a starting point $x_0 \in \mathbb{R}^n$, the Newton's method repeats

$$x_i = x_{i-1} - (\nabla^2 f(x_{i-1}))^{-1}\nabla f(x_{i-1}) \quad \forall i \in \mathbb{N}$$

to solve the optimization problem.