# OpenReview forum: "Sampling with Riemannian Hamiltonian Monte Carlo in a Constrained Space"
_NeurIPS.cc/2022/Conference — NeurIPS 2022 Accept_

### Official Review · Reviewer_kCFo · 2022-07-05

**Rating:** 4
**Confidence:** 3
**Soundness:** 3 good
**Presentation:** 2 fair
**Contribution:** 2 fair

**Summary:**

This paper provides a variation of the Constraint Riemannian Hamiltonian Monte Carlo (RHMC). Constraint variations of HMC deal with  sampling from a target distribution that by means of some constraints, is (implicitly) defined on a manifold. In the present work it is assumed that the constraint has the form c(x) = Ax − b. Such assumptions has led to fast computations and as it is mentioned, the proposed algorithm can be run on target spaces with dimensionality as high as 100K.

**Questions:**

* The proposed work is only compared against (2 implementations of) CHRR algorithm. I suggest that the comparison be with other variations of constraint HMC (e.g. Brubaker's constraint baseline HMC, constraint Metropolis-Hastings) and/or more recent implementations of CHMC algorithms such as Graham's "Manifold Markov chain Monte Carlo methods for Bayesian inference in diffusion models".

* I think CHRR is for sampling from (constraint) uniform distributions. Are the presented experimental target densities uniform? If no, how CHRR is used and if yes, what is the usage of Riemannian HMC (since gradients are always 0) ?



**Limitations:**


Not applicable. This work which is theoretical.

**Strengths And Weaknesses:**

* It seems to me that the contribution of the presented work is incremental. The idea of constraint HMC is already proposed in a more general setting by:
    Brubaker et al, "A Family of MCMC Methods on Implicitly Defined Manifolds"
where Constrained Riemann Manifold HMC (CRHMC) is mentioned as a specific case. The present work correctly gives this credit to Brubaker et al. (in Line 135) but then claims Brubaker's CRHMC "requires eigenvalue decomposition and is not efficient for large problems". As far as I understand, Brubaker's work does not necessarily require an eigenvalue decomposition and the present work is indeed an instance of Brubaker's work where extra assumptions/restrictions mentioned in Lines 192 to 220, has led to efficient computations.

* If I have understood correctly and the contribution is efficient computations under extra assumptions/restrictions, then the robustness of this work could possibly be demonstrated by extensive empirical experiments. Nonetheless, the provided experimental results seem minimal to me (the proposed algorithm is only compared against one benchmark) which is suitable for sampling from (constraint) uniform distributions only. I will return to this in the "Questions" section.

* In general the paper's writing could be improved and notations/terminology could be defined more rigorously.

---

> ### Author Response · Authors · 2022-07-30
> **Response to Reviewer kCFo**
>
> Thank you for your review. We are not sure why you put the ethics flag on our work. Could you please clarify?
>
> Response to Strengths and Weakness
> - Eigendecomposition of Brubaker’s CHMC: Brubaker’s implementation runs eigendecomposition in line 232 in the function computeLogNormConst in constrainedHMC.m in their package (https://www.cs.toronto.edu/~mbrubake/projects/cmcmc/). The authors further write in line 231 that “FIXME: there is almost certainly a better way to do this”.
> - Regarding our contributions: We disagree with the assessment that this work is incremental. It gives a 1,000-fold speed-up, allowing us to sample in previously inaccessible dimensions up to 100k with theoretical guarantees on benchmark data sets from computational biology.
> We would like to take this opportunity to clarify the contributions of the paper that you might have missed.
>   - Our CRHMC uses self-concordant barriers to bound the process inside the region defined by the `inequality’ constraints, in addition to the constraint $\{c(x)=0\}$, using the local metric induced by the Hessian of the barrier. Self-concordant barriers allow us to sample from such constraints efficiently, independent of the condition number.
>   - For constraints in the form $c(x)=Ax-b$, we develop several computational techniques and exploit the sparsity of $A$, allowing us to scale CRHMC to high dimensions. It takes several pages of non-trivial calculation, and all these computational techniques are effective on real-world constrained polytopes, as demonstrated in the experiment section. We believe these numerical tools are important in developing practically efficient algorithms.
>   - Our experiments are performed on metabolic models and real-world LP problems from benchmark data sets. For many instances in these datasets, our package is the first one that can sample in a reasonable time.
>   - Lastly, unlike other constrained HMC papers, we go the extra mile to analyze its convergence rate and to ensure that this work is more than an empirical work. Its convergence rate is independent of the condition number of the convex body (i.e., our CRHMC is guaranteed to work well for ill-conditioned problems).
>
> Response to questions
> - Other constrained HMC for the experiment: As we mention in Line 259, other constraint HMC implementations including Brubaker’s one do not support manifolds implicitly defined by the intersection of inequalities and $\{c(x)=0\}$ and other popular packages like STAN do not support constrained-based models. Therefore, we cannot compare our package with these implementations.
> - Target density in the experiment: Yes, the experiment for comparing against CHRR samples from the uniform distribution. For the uniform distribution, zero gradients have nothing to do with the effectiveness of Riemannian HMC. Since we work on a convex body $K$, the target density is proportional to the indicator function $1_{K}(x)$, and RHMC behaves differently from HMC in that RHMC takes into account the geometry of this target density; it locally stretches metric depending on how close a current position is to the boundary of the convex body.
> - Moreover, our CRHMC usually works as fast or faster for Gaussian sampling compared to uniform sampling. On five instances (cardiac_mit, ecoli, israel, Aci_D21, Aci_MR95), the sampling times per effective sample for Gaussian sampling are 0.012, 0.011, 0.097, 0.077, 0.15, while those for uniform sampling are 0.010, 0.010, 0.12, 0.43, 0.96.

---

> > ### Comment · Reviewer_kCFo · 2022-08-05
> > **One the code of Ethics and contribution**
> >
> > The ethics flag was for: "Under-representing an existing more general work".
> > I remove that flag however I still think that is the case! I would appreciate if the authors could either show that I am missing a point or would modify their claims.
> >
> > In the response, you correctly state that "Brubaker’s implementation runs eigendecomposition "
> > However, that does not address my concern. The theory that they present (regardless to the implementation) is general and does not *necessarily* require eigen value decomposition. Do you agree with this ?
> >
> > If you agree then do you also agree that your work is a special case of that work where extra assumptions/restrictions mentioned in Lines 192 to 220, has led to efficient computations ?

---

> > > ### Author Response · Authors · 2022-08-07
> > > **Response**
> > >
> > > “Do you also agree that your work is a special case of that work where extra assumptions/restrictions has led to efficient computations”
> > >
> > > - We disagree. Our setting in (1.1) has a convex set inequality constraint. For example, [0,1] interval is a convex set constraint. Brubaker’s CHMC focuses on a manifold without boundary while our paper is able to solve the boundary constraints in addition to their setting by introducing a new metric. Therefore, our paper is not a special case.
> > > - The classical way to extend sampling algorithms to the case with boundaries would simply reflect the walk around the boundary. Unfortunately, without using a proper Riemannian metric, such an algorithm needs to take a tiny step size to avoid stepping outside the domain and hence does not work for problems with million parameters. Handling the extra inequality constraint can in no way make the setting easier, for the simple reason that there are more constraints the algorithm needs to take care of. In order to handle boundary/inequality constraints efficiently, we had to use efficient self-concordant barriers. Therefore, our setting is harder than that of Brubaker’s and can be applied to more problems in practice. Moreover, Brubaker’s paper did not provide convergence rate proof while we are able to give a theoretical convergence rate guarantee.
> > >
> > >  “The theory that they present (regardless to the implementation) is general and does not necessarily require eigenvalue decomposition. ”
> > > - The reason they require a pseudo determinant is that the constraint matrix induces a low-rank matrix and therefore they need to compute a pseudo determinant for the Hamiltonian. Certainly, if the matrix is diagonal, then we can compute the pseudo determinant by only looking at the diagonal. Other than that case, it is difficult to say when we can have a simpler algorithm that can avoid eigenvalue decomposition. Therefore, unless for very simple cases like diagonal jacobian (of the constraint) with diagonal metric, before our paper, we believe in general SVD is required. This is the reason that the author already writes in their program that it is a problem that needs to be fixed. We do not think repeating their sentence in our paper is a misrepresentation. Please refer to the calculation on pages 17-20 in the appendix. The way we arrive at the formulas of our implementation is non-trivial.
> > >
> > > In addition to the above response, we would like to point out that, in general, result A is a variant of result B that works better than B in certain settings does not make A of any less value. As an analogy, one can argue all deep learning papers are special cases of the more general result of SGD that is known since the 50s because those papers are just coming up with new objective functions, in the same way as arguing we are just coming up with a better metric when there are boundary conditions to handle. The results of many deep learning papers are cool because of their applications. We have already demonstrated that our algorithm is of value by showing that our algorithm performs better on the benchmark problems by 3 orders of magnitude. Our package has been incorporated into a popular package used by the Bioinformatics library. Therefore, we believe we have sufficiently demonstrated the application of our algorithm.

---

### Official Review · Reviewer_jRWB · 2022-07-09

**Rating:** 7
**Confidence:** 4
**Soundness:** 3 good
**Presentation:** 3 good
**Contribution:** 2 fair

**Summary:**

The manuscript proposes a MCMC sampler based on the Constrained HMC method for sampling from a log log-concave density $\pi(x) \propto \exp[-f(x)]$ constrained on a convex subset $x \in K \subset \mathbb{R}^d$ and a constraint $c(x)=0$. The main innovations are computational tricks to scale the method to high-dimensional settings and exploit the sparsity pattern of constraints of the type $Ax=b$ for a sparse matrix $A$.

**Questions:**

The manuscript proposes a MCMC sampler based on the Constrained-HMC method for sampling from a log-concave density $\pi(x) \propto \exp[-f(x)]$ and constrained to stay within a convex subset $x \in K \subset \mathbb{R}^d$ and and an implicit manifold defined through the constraint $c(x)=0$. The main innovations are computational tricks to scale the method to high-dimensional settings and exploit the sparsity pattern of constraints of the type $Ax=b$ for a sparse matrix $A$.

The proposed method seems to deliver state-of-the-art results when evaluated in a range of challenging settings (the numerical studies are well executed). Furthermore, the authors have incorporated their method within a popular Bioinformatics library.


1. I am not finding the discussion of sparsity well-motivated or explained. The authors mention that sparsity is preserved, but it takes quite a bit of time for the reader to realize that (I think) it is related to the sparsity pattern of A within the constraint Ax=b.

2. The authors are not clear within the main paper (although it is somehow clarified in the Supp Material) whether they are considering MC methods with or without an accept reject. For example, several of the cited papers on (variation of) the Langevin dynamics do not have an accept-reject step (and are consequently only approximately sampling from the posterior).

3. It would be best to make it clearer whether the paper concentrates on constraints of the type Ax=b, or more general implicit manifolds. It seems like all the experiments are with linear constraints.

4. It was not clear to me whether the idea of using barrier functions within an HMC scheme is new or not. If it is not, it would be best to clarify. Something that is not very well explained is the following. Suppose that one runs a standard HMC algorithm (with explicit leapfrog integrator) in $\mathbb{R}^d$ constrained to a convex set $K \subset \mathbb{R}^d$ (i.e. without an equality constraint $c(x)=0$). For a barrier function $\phi$ and positive definite mass matrix $M(x) = \nabla^2 \phi(x)$, there does not seem anything that prevents the Markov chain to step out of the convex set $K$ because of discretization error. Is it different when using the implicit midpoint method for integrating the Hamiltonian dynamics? Can the authors clarify this?

5. Do the authors use the assumption that the target density is log-concave? where?

6. For general constraint $c(x)=0$, the CHMC requires a "projection" step to bring back the Markov chain back to the manifold $c(x)=0$. In practice (i.e. without further assumptions on the function $c$), this projection step may fail. It is the reason why the "reversibility checks"  have been developed (eg. see [1]).

7. There is a discussion that describes how having a position-dependent mass matrix can help take the geometry of the distribution into account for improved mixing properties (i.e. Riemaniann-HMC/etc..). But, if I am not mistaken, the authors are simply using as mass matrix the projection of $\nabla^2 \phi(x)$ on the tangent space to the manifold $c(x)=0$, which is *not* taking the geometry of the target distribution (eg. anisotropy of scales) into account (i.e. the target distribution $\pi(dx)$ is not used to design the mass matrix)

8. No discussion of how to tune the leapfrog step size and momentum refresh rate (i.e. coefficient $\beta$)?

9. The authors reports the ESS, but do not specify the ESS of what functional of the trajectory is considered



**Minor**:
1. if space is an issue, I do not think that it is necessary to spend so much time explaining to NeurIPS readers that sampling is important and difficult. The introduction can be shortened, if needed.


**[Answers by the authors acknowledged 4/8/2022]**
Thank you for the careful answers that have clarified my misunderstandings. It is indeed a very solid piece of work!

**References:**
1. "Monte Carlo on manifolds: sampling densities and integrating functions" by Emilio Zappa, Miranda Holmes-Cerfon, Jonathan Goodman

**Limitations:**

OK.

**Strengths And Weaknesses:**

The proposed method seems to deliver state-of-the-art results when evaluated on a range of challenging settings (the numerical studies are well executed). Furthermore, the authors have incorporated their method within a popular Bioinformatics library.

I have a number of questions, as well as remarks on the presentation of the paper, that are listed in the next section.

---

> ### Author Response · Authors · 2022-07-30
> **Response to Reviewer jRWB**
>
> Thank you for your careful reading of our paper and for your questions and reference suggestions.
>
> - Sparsity preservation: Your understanding of sparsity is correct, and we agree that we should elaborate on how sparsity is defined (number of nonzeros of A). Maintaining sparsity is key to computation and space efficiency. We will write a separate paragraph addressing this.
> - Accept-reject step: In theory, the accept-rejection step is not required if the Hamiltonian ODE is accurately solved, but this is impossible in practice. Since IMM does not preserve the Hamiltonian, to achieve high accuracy, we have to use the accept-rejection step in the implementation. We will mention this in the revised version.
> - Constraint form: Our paper focuses on two types of constraints: equality constraints Ax=b and inequality constraints Ax<=b.
> - To address your fourth question,
>   - Barrier functions for HMC: Even though barrier function is an existing tool for constrained HMC, we would like to emphasize that we use a special class of barrier functions, self-concordant barriers. In fact, as shown in Appendix D, the properties of self-concordance serve as the backbone to the proof of condition number independence of CRHMC. The usage of self-concordant barrier functions in handling inequality constraints was inspired by the interior point method in optimization theory.
>   - The rationale behind the choice of implicit midpoint method (IMM): For general Riemannian manifolds, the leapfrog method (LM) is not symplectic, unlike IMM. LM is symplectic when the Hamiltonian is separable (i.e., each of $dx/dt$ and $dv/dt$ is a function of either x or v only), which is the case for Hamiltonian Monte Carlo. However, in the general Riemannian manifold setting, where $dx/dt$ depends on position $x$ due to mass matrices (which is $g(x)$ in our paper) as well as velocity $v$, the Hamiltonian is no longer separable, which prevents us from using LM. You can find a more detailed explanation in Section 3 and Section 4.1 in https://arxiv.org/pdf/1910.06243.pdf. Also, there have been some attempts (https://arxiv.org/abs/1411.3367) to make LM work in the Riemannian setting, but they do not work for ill-conditioned problems, unlike our algorithm.
>   - Scenario for stepping out of the body: For the reason mentioned above, you cannot run HMC with the explicit leapfrog integrator. Back to your concern over stepping out of the body, CRHMC can prevent the chain from stepping outside. As the chain approaches the boundary, the local metric (i.e., the Hessian of self-concordant barriers) stretches accordingly so that you never pass the barrier (see https://player.slideplayer.com/90/14560948/slides/slide_26.jpg). To see this formally, suppose that the ideal CRHMC with step size $h$ brings $x$ to $x’$. In Line 1020, we show $||x-x’||_{\nabla^2 \phi(x)} < 1/4$ for step size $h = \tilde{O}(1/n)$. Hence, $x’$ is contained in the Dikin ellipsoid (Line 1093) at $x$, and it is well known that the Dikin ellipsoid is fully contained in the convex body. Next, as shown in Line 1044, for step size h, we can show that the distance between the ideal and discretized version is $< 1/4$ in the local metric, so again, the point obtained from the discretization is contained in the Dikin ellipsoid and cannot leave the body.
> - Role of log-concave density: Our mixing rate proof of discretized CRHMC uses an isoperimetric inequality for log-concave densities (see Appendix D.2).
> - Geometry of distribution in the design of mass matrices: Since we work in a constrained space (convex body in particular), a target distribution $\pi$ is basically some distribution $F$ truncated to the convex body $K$ (i.e., $d\pi/dx \propto dF/dx\cdot 1_K(x) $. Our point is that the target distribution has two parts, the convex body and the distribution $F$, so we can argue that the local metric induced by the Hessian of barriers for the convex body uses the geometry of the target distribution. Still, as you point out, an attempt to exploit the geometry of function $F$ in addition to that of the constraints is certainly an interesting research direction.
> - How to tune step size and momentum: We note that our CRHMC is affine-invariant and provably independent of the condition number (Theorem 14), and thus the step size and the momentum only need to depend on the dimension. In practice, we set the momentum to (1 - step-size), and for step size, we decrease it until the acceptance probability is close enough to 1 during a warm-up phase. Heuristically, we found that the step size stays between 0.05 and 0.2 in practice even for high dimensional ill-conditioned polytopes. This step size is remarkable, given that for the instances in our experiments, a standard package like STAN can end up selecting a small step size like $10^{-8}$ and thus fails to converge.
> - ESS: We report the minimum of the ESS of each coordinate.

---

### Official Review · Reviewer_Cyv8 · 2022-07-10

**Rating:** 5
**Confidence:** 3
**Soundness:** 3 good
**Presentation:** 2 fair
**Contribution:** 3 good

**Summary:**

The authors propose a method to sample from a distribution using Riemannian Hamiltonian Monte Carlo while respecting a set of constraints. The efficiency of the method relies on a combination of tricks from linear algebra, while the proposed Riemannian metric is based on the Jacobian of the constraints and a barrier function. In the experiments it is demonstrated the behavior of the sampler in different scenarios.

**Questions:**

In particular, it is not clear to me:
1. why the metric should not be invertible
2. how the associated null spaces are related to the problem
3. why the metric is then constructed as in Eq. 2.6
4. how the function $\phi$ is defined in practice.

In general, I find hard to access the Section 2.1. I think some discussion about the geometrical intuition/properties of these steps together with some figures could make the paper much more accessible. I believe that this part can be improved.

**Limitations:**

The authors discuss some limitations of the proposed method.

**Strengths And Weaknesses:**

- Originality: As the authors mention their approach is based on a previous work, which they extend such that to make it much more efficient computationaly. I think that the related work is cited and discussed very briefly in the intro. I am not an expert in the topic of sampling, but the proposed approach seems like a sufficient contribution.

- Quality: The technical part of the paper seems ok, but I have not checked it carefully. The claims are supported by theoretical analysis and experimentally validated.

- Clarity: I think that the paper is ok written, but the Section 2 seems a bit too technical without the necessary explanations to help the reader. For example, a lot of linear algebra results are used, but I have the feeling that the gist and intuition are missing. I believe that this makes the paper hard to access from non-expert readers like me. (Please see questions)

- Significance: From the claims and the empirical results it seems that the method indeed solves the problems better (more efficient) than previous approaches. It is quite likely that this work would be useful when sampling under constraints.

---

> ### Author Response · Authors · 2022-07-30
> **Response to Reviewer Cyv8**
>
> Thank you for your review and questions.
>
> Response to clarity
>
> Appendix E contains details missing in the main manuscript regarding our technical developments. We will supplement Appendix E by clarifying the linear algebraic facts (along with intuition) used in the paper.
>
> Response to questions
> - Singularity of metric matrix ($M(x)$): A main point is that the constrained space $\{x \in \mathbb{R}^n : c(x) = 0\}$ (which is the null space of $c(x)$) is a subspace of $\mathbb{R}^n$, so its dimension could be strictly smaller than that of the underlying space (i.e., $n$). For example, imagine that the constrained space is a two-dimensional plane in $\mathbb{R}^3$. Note that at each step the CRHMC moves in a direction $v$ drawn from the Gaussian distribution $N(0, M(x))$ and so $v$ is governed by $M(x)$. If $M(x)$ is invertible, then it has full rank, meaning that $v$ can be in any direction. CRHMC must always respect the constraint $c(x) = 0$ throughout all iterations, but some directions, say the direction perpendicular to the constrained space, would make CRHMC escape from the constrained space. Hence, $M(x)$ in general is not full rank, and not invertible. Actually, one main contribution of our work is that our implementation can handle such constrained space efficiently while the previous packages can’t.
> - Relevance of null space to formulation: As in our response above, the constrained space $c(x)=0$ is the null space of $c(x)$, In particular for our setting $c(x) = Ax - b$, the constrained space is a translation of the null space of $Dc(x) = A$.
> - Formula of metric (Eq 2.6): Since the random direction $v$ takes values from the image of the metric matrix, the orthogonal projection to $c(x)$ (or the null space of $A$) is a good candidate for the metric matrix. The orthogonal projection to $c(x)=0$ projects any point onto $c(x)=0$, so its image lies in the constrained space, $c(x)=0$. Then we simply use a well-known formula of orthogonal projections in Eq. 2.6.
> - Choice of barrier $\phi$: As we demonstrate in the Experiment section, the log-barrier $\phi(x) = - \sum_{i=1}^m (a_i^{\top}x-b_i)$ works well in practice.

---

> > ### Comment · Reviewer_Cyv8 · 2022-08-08
> > **After rebuttal**
> >
> > First of all, I would like to thank you for the reply.
> >
> > I think that your answers clarify my questions. I also believe that some discussions and comments of this style can improve the clarity of the paper and make it more accessible from (non-expert) readers. In particular, you can consider adding a few figures to show graphically all the relevant information. Perhaps, a comparison with previous approaches can be made using some figures and explaining the differences.
> >
> > I am considering improving my score from 5 to 6, after the discussion with the rest of the reviewers.

---

> > > ### Author Response · Authors · 2022-08-08
> > > **Response**
> > >
> > > Thanks again for your review. We are glad our responses clarified your questions and thank you for considering improving the score. We will add figures and more detailed explanations to make the relevant information clearer and more accessible.

---

### Official Review · Reviewer_CjP3 · 2022-07-11

**Rating:** 8
**Confidence:** 4
**Soundness:** 4 excellent
**Presentation:** 3 good
**Contribution:** 4 excellent

**Summary:**

The paper develops a constrained version of Riemannian Hamiltonian Monte Carlo, maintaining both sparsity and constraints. The paper shows that ill-conditioned, non-smooth, constrained distributions in very high dimensions, upwards of 100,000, can be sampled efficiently. The work is supported by several theoretical results on the correctness and computational efficiency, as well as bounds on the mixing time of the CRHMC and its discretizations.

**Questions:**

(see above)

**Limitations:**

None noted.

**Strengths And Weaknesses:**

### Strengths

- Significance:
    - The paper considers an important problem in machine learning that is of great interest to the broader machine learning community.
    - The proposed framework can address sampling problems that are deemed infeasible previously. The paper shows that ill-conditioned, non-smooth, constrained distributions in very high dimensions, upwards of 100,000, can be sampled efficiently.
    - The experiments also demonstrate the approach's superiority in comparison to existing methods in terms of rate of convergence and total sampling time.
- Quality and Clarity:
    - The paper is well-written and well-organized.
    - The theoretical analyses of the work are rigorous and of very high quality. The experiments are extensive and support the findings of the theoretical part.

### Weaknesses

- Quality:
    - I believe the proof of Theorem 8 (Appendix, on the correctness of the discretized version of CRHMC) is incorrect. In principle, the detailed-balance conditions need to be verified on two arbitrary measurable sets. This could be done by verifying the condition pointwise (as done in Theorem 8) if the state space is discrete or the proposal distribution is absolutely continuous. HMC proposals, however, contain point masses, and the detailed-balance condition can be verified pointwise only if the proposal is volume-preserving. This can be seen in the proof of Theorem 6 for the ideal version of RCHMC but is absent in Theorem 8.

      It is worth noting that this a not a minor point since the rest of the theoretical analyses (e.g., bounds on mixing time) relies on the convergence of discretized CRHMC obtained through this Theorem.
    - This brings a broader point about the choice of integrator for RCHMC, namely, why the implicit midpoint is preferred over the leapfrog integrator (which is known to be volume-preserving). The implicit midpoint is described in the paper with no context or references and very few details about its properties are spelled out. I think such an important component of the algorithm deserves some details treatment on its background and properties.
    - A minor point: the paper seems to be reluctant to state that RCHMC is ergodic or converges into the formal statement of the theorems: Theorem 6 states that RCHMC satisfies detailed balance; Theorem 8 states that RCHMC has a stationary distribution. Mathematically, those two statements are much weaker than ergodicity and are not strong enough to support later discussions such as Line 816.
- Presentation:
    - To continue on the point above, while the development of CRHMC follows well-founded rationales, I find it very hard to separate the contributions of the current construction from pre-existing methods in the field, simply because the paper doesn’t provide enough contexts. The construction of the constraint term in the Hamiltonian $\lambda(x, v)^T c(x)$ is rigorous, but is it an original concept, or does it draw inspiration from pre-existing works? The barrier function is a well-known tool in constrained HMC, but was introduced with no proper references. Similarly, several aspects of efficient computation and simplifications in Section 3 seems to base on are pre-existing ideas. I believe a more thorough review of the key ideas would strengthen the manuscript greatly.

---

> ### Author Response · Authors · 2022-07-29
> **Response to Reviewer CjP3**
>
> Thank you for your detailed review.
>
> Response to weaknesses
>
> Quality
> - Proof of convergence: We apologize for not directly mentioning in proof that the implicit midpoint method (IMM) is symplectic (so volume-preserving) in the Riemannian setting. We mention in Line 230-231, 723, and 781 that IMM is symplectic, with reference to literature on numerical integrators. Since IMM is volume-preserving, the argument of pointwise verification of detailed-balance condition is valid.
> - Rationale behind choice of IMM: Your second point brings to our attention that in our paper we did not explain the benefits of using IMM to discretize CRHMC. For general Riemannian manifolds, the leapfrog method (LM) is not symplectic, unlike IMM. LM is symplectic when the Hamiltonian is separable (i.e., each of $dx/dt$ and $dv/dt$ is a function of either x or v only), which is the case for Hamiltonian Monte Carlo. However, in the general Riemannian manifold setting, where $dx/dt$ depends on position $x$ due to mass matrices (which is $g(x)$ in our paper) as well as velocity $v$, the Hamiltonian is no longer separable, which prevents us from using LM. You can find a more detailed explanation in Section 3 and Section 4.1 in https://arxiv.org/pdf/1910.06243.pdf. There have been some attempts (https://arxiv.org/abs/1411.3367) to make LM work in the Riemannian setting, but they do not work for ill-conditioned problems unlike our algorithm.
> We will supplement Section 2.3 and Appendix C with our response above.
> - Non-rigorous statement of convergence: We will make them clearer in the new version, along with proofs for irreducibility and aperiodicity. As you mentioned, ergodicity and invariance of Markov chains imply that the process converges to that stationary distribution. What we show in Theorem 6 and 8 is detailed balance (so invariance). For the ideal CRHMC, its $\pi$-irreducibility and aperiodicity can be shown as proven in Theorem 3 and Lemma 1 respectively in Brubaker’s paper, and thus the ideal CRHCM actually converges to the target distribution.
> For the discretized CRHMC, we can show those properties as follows. For irreducibility, the non-zero lower bound on the conductance of the discretized CRHMC (Proposition 1) implies that the Markov chain is irreducible. To see this, let $A$ and $B$ be two subsets of positive measure such that one subset is not reachable from another infinitely many steps. Take the set $R$ of reachable points from $A$ via running the Markov chain, and note that $R$ and $R^c(\supseteq B)$ have non-zero measures. However, the non-zero conductance, meaning that there must be a positive probability of stepping out of R, contradicts the definition of $R$. Now for aperiodicity, as assumed at the beginning of the mixing rate proof (Line 887~), we may consider a lazy version of the discretized CRHMC instead, which makes the chain stay where it is at with probability $1/2$ at each iteration, to prevent potential periodicity of the process. Note that it worsens the mixing rate only by a factor of 2. The above argument (together with detailed balance in Theorem 8) shows that the discretized CRHMC converges to the target distribution.
>
> Presentation
> - We will add proper statements and citations to separate what are our new ideas and what are not.
>   - Constraint term: Adding such a Lagrangian term to handle constraints is common in optimization theory, which inspires our algorithm. We are not aware of its prior use in sampling.
>   - Barrier function for constrained HMC: Even though barrier function is an existing tool for constrained HMC, we would like to emphasize that we use a special class of barrier functions, self-concordant barriers. In fact, as shown in Appendix D, the properties of self-concordance serve as the backbone to the proof of condition number independence of CRHMC. The usage of self-concordant barrier functions in handling inequality constraints was inspired by the interior point method in optimization theory.
>   - Computational tricks for CRHMC: While some of the ideas (e.g., how to deal with sparsity) are not original, our contribution is how we put them together to make them meaningful and effective as a whole. Those techniques from several different areas together contribute to our computationally fast and efficient implementation of CRHMC, leading to significant speedup in practice.

---

> > ### Comment · Reviewer_CjP3 · 2022-08-03
> > **(comments addressed)**
> >
> > Thanks for your detailed responses; I think all of my comments are addressed. I intent to revise the rating up (This might have to wait until after consults with other reviewers.)

---

> > > ### Author Response · Authors · 2022-08-08
> > > **Response**
> > >
> > > Thanks again for your review. We are glad you find our responses helpful and thank you for intending to raise the score.

---

### Official Review · Reviewer_y2k1 · 2022-07-11

**Rating:** 8
**Confidence:** 3
**Soundness:** 3 good
**Presentation:** 4 excellent
**Contribution:** 4 excellent

**Summary:**

This paper gives a method for sampling log-concave distributions supported on compact convex sets, based on Riemannian Hamiltonian Monte Carlo (RHMC). Equality constraints are enforced via a Lagrange multiplier and inequality constraints are enforced via a carefully chosen Riemannian metric constructed from a self-concordant barrier function. Algorithms for exploiting sparsity for matrix inversion enable sampling from high-dimensional distributions. Tests show substantial improvement in sampling times in comparison to existing constrained samplers. Formal guarantees are provided for convergence of the associated integration scheme the convergence to the stationary distribution is bounded.

**Questions:**

Would it be possible to extend this to noncompact sets? How about non-convex sets and non-log-concave distributions? What are the challenges?

**Limitations:**

The authors address many technical limitations, but as described above, some fairly serious technical limitations are not adequately discussed without digging into the technicalities.

The authors indicate not potential negative societal impact, and I agree.

**Strengths And Weaknesses:**

Strengths:
This is a nice paper. It gives strong practical results for an important, but largely overlooked problem and gives a thorough theoretical analysis. The paper is very well-organized, with helpful guides to the structure of the appendix.

Weaknesses:
Probably the biggest weakness is that some of the assumptions are not clear from the problem setup. In particular, it is just stated that the constraint set, $\mathcal{K}$, must be convex. However, it really must be compact. Many of the arguments implicitly depend on this. These restrictions should definitely be clarified in the problem setup, and ideally also in the abstract.

If $\mathcal{K}$ is not compact, the cross-ratio distance from Definition 5 is not well-defined. It might be possible to extend this definition to the case in which $\mathcal{K}$ does not contain a line by using appropriate limits. However, in this case the constant $G$ from Lemma 17 could be zero. (Consider the case of the non-negative orthant in $\mathbb{R}^2$.)

If $\mathcal{K}$ is not compact and the $f$ is linear, then the corresponding density may not exist, rendering arguments about stationarity from B.3 irrelevant.

There are also some small typos and unclear arguments. Unfortunately, I did not have the time to step through every  argument in detail. But, I did catch a few issues:

- In line 575, it should read $MNM=M$
- In many places, "dynamic" is used where "dynamics" would be more common. For example, in 603 "the HMC dynamic" would normally be "the HMC dynamics"
- Similarly, "the dynamic on x" would normally be "the dynamics of x", in Lemma 4.
- $P$ gets redefined in line 601, and clashes with the definition from Lemma 2 in a confusing way.
- In line 798, we should have $\|x\|_A = \sqrt{x^\top A x}$
- Definition 5 implies that $\mathcal{K}$ must be compact. This should be stated more explicitly in the problem setup.
- I think there is a typo in the first inequality after line 967. I feel like this is based on a lower bound on the right side of the equation after line 964. So, the second side should have an integrand of $\mathcal{T}_x(S^c)$.
- I think the factor on the second inequality below line 967 should be $\frac{1}{40}$ and not $2/5$. This is because the bound seems to follow from lower bounding the integrands by $0.05=\frac{1}{20}$ for both terms. I don't think this really matters for the rest of the proof.
- The last inequality below line 968 should be strict.

---

> ### Author Response · Authors · 2022-07-29
> **Response to Reviewer y2k1**
>
> Thank you for your appreciation of our paper.
>
> Response to weaknesses
> - Compactness of K: Our work assumes that K is a convex body, i.e., it is convex and compact. We realized that we used both  ‘convex body’ and ‘convex set’, and we apologize for the confusion. We hope that this clarification relieves your concerns over the technical issues you mentioned in Limitations.
> - Thank you for pointing out the typos. We will fix them.
>
> Response to questions
> - Extension of CRHMC: Your questions raise important future research problems, solutions of which will certainly make CRHMC more impactful. In particular, the Hamiltonian Monte Carlo algorithm is known to work well for sampling from some non log-concave densities in practice, and there is nothing in our algorithm that prevents us from using CRHMC for non log-concave densities. However, the theoretical analysis still remains a challenge.

---

> > ### Comment · Reviewer_y2k1 · 2022-08-09
> > **Reply to rebuttal**
> >
> > Thanks for the response. I already had a high rating, and my plan is to keep it high.

---

### Author Response · Authors · 2022-08-02
**Rebuttal Revision**


Dear all,

We appreciate all the valuable feedback on our manuscript. We fixed the typos and made the following changes to our manuscript.
- L124 We added “convex body K” (convex and compact) to clarify the assumptions on the constraint set
- P3 footnote 1 We clarified sparsity.
- L133-135 We added the role of the self-concordant barrier to emphasize the difference between our algorithm and CHMC
- L169-176 We added more intuition on the self-concordant barrier.
- L220-223, L729-734 We added the motivation for using IMM instead of the commonly used integrators such as leapfrog.
- L753-759 We added how we chose the momentum and the step size in practice.
- L711-712, L806-814 We added the proof of irreducibility and aperiodicity to the proof that the Markov chain converges to the target distribution.
- L1140-1148, L1179-1186 We added more notations and details on matrix calculus.

---

### Meta-Review · Area_Chair_ka36 · 2022-08-20

**Recommendation:** Accept
**Confidence:** Certain

**Metareview:**

The focus of the submission is practically efficient sampling of non-smooth constrained log-concave distributions in high dimension as formulated in (1). In order to tackle this task, the authors design a constrained variant of the RHMC (Riemannian Hamiltonian Monte Carlo) technique, relying on self-concordant barrier functions. They demonstrate the practical efficiency of the proposed method (achieving significantly improved sampling time) compared to existing constrained samplers, with additional theoretical results in the supplement.

Sampling in high dimension is a fundamental problem of data science with various successful applications; designing new methods in the area is of clear interest for the NeurIPS community. As it was assessed by the reviewers, the authors deliver important new tools in this context. As it was also noted, the manuscript could be made slightly more accessible to wider audience.

**Award:**

No

---

### Decision · Program_Chairs · 2022-09-14

Accept